



# Carbon/nitrogen interactions in European forests and semi-natural vegetation. Part I: Fluxes and budgets of carbon, nitrogen and greenhouse gases from ecosystem monitoring and modelling

Chris R. Flechard[1], Andreas Ibrom[2], Ute M. Skiba[3], Wim de Vries[4], Marcel van Oijen[3], David R. Cameron[3], Nancy B. Dise[3], Janne F.J. Korhonen[5,6], Nina Buchmann[7], Arnaud Legout[8], David Simpson[9,10], Maria J. Sanz[11], Marc Aubinet[12], Denis Loustau[13], Leonardo Montagnani[14,15], Johan Neirynck[16], Ivan A. Janssens[17], Mari Pihlatie[5,6], Ralf Kiese[18], Jan Siemens[19], André-Jean Francez[20], Jürgen Augustin[21], Andrej Varlagin[22], Janusz Olejnik[23,24], Radosław Juszczak[25], Mika Aurela[26], Bogdan H. Chojnicki[25], Ulrich Dämmgen[27], Vesna Djuricic[28], Julia Drewer[3], Werner Eugster[7], Yannick Fauvel[1], David Fowler[3], Arnoud Frumau[29], André Granier[30], Patrick Gross[30], Yannick Hamon[1], Carole Helfter[3], Arjan Hensen[29], László Horváth[31], Barbara Kitzler[32], Bart Kruijt[33], Werner L. Kutsch[34], Raquel Lobo-do-Vale[35], Annalea Lohila[36,26], Bernard Longdoz[37], Michal V. Marek[38], Giorgio Matteucci[39], Marta Mitosinkova[40], Virginie Moreaux[13,41], Albrecht Neftel[42], Jean-Marc Ourcival[43], Kim Pilegaard[2], Gabriel Pita[44], Francisco Sanz[45], Jan K. Schjoerring[46], Maria-Teresa Sebastià[47,48], Y. Sim Tang[3], Hilde Uggerud[49], Marek Urbaniak[23], Netty van Dijk[3], Timo Vesala[36,6], Sonja Vidic[28], Caroline Vincke[50], Tamás Weidinger[51], Sophie Zechmeister-Boltenstern[52], Klaus Butterbach-Bahl[18], Eiko Nemitz[3] and Mark A. Sutton[3]

[1] Institut National de la Recherche Agronomique (INRA), UMR 1069 SAS, 65 rue de Saint-Brieuc, F-35042 Rennes, France
[2] Department of Environmental Engineering, Technical University of Denmark, Bygningstorvet, DK-2800 Kgs. Lyngby, Denmark
[3] Centre for Ecology and Hydrology (CEH), Bush Estate, Penicuik, EH26 0QB, UK
[4] Wageningen University and Research, Environmental Systems Analysis Group, PO Box 47, NL-6700 AA Wageningen, the Netherlands
[5] Environmental Soil Science, Department of Agricultural Sciences, Faculty of Agriculture and Forestry, PO. Box 56, FI-00014 University of Helsinki, Finland
[6] Institute for Atmospheric and Earth System Research/Forest Sciences, Faculty of Agriculture and Forestry, PO. Box 27, FI-00014 University of Helsinki, Finland
[7] Department of Environmental Systems Science, Institute of Agricultural Sciences, ETH Zurich, LFW C56, Universitatstr. 2, CH-8092 Zurich, Switzerland
[8] INRA, BEF, F-54000 Nancy, France
[9] EMEP MSC-W, Norwegian Meteorological Institute, Oslo, Norway
[10] Dept. Space, Earth & Environment, Chalmers University of Technology, Gothenburg, Sweden
[11] Basque Centre for Climate Change (BC3), Scientific Park, Sede Building, s/n Leioa, Bizkaia, Spain
[12] TERRA Teaching and Research Centre, Gembloux Agro-Bio Tech, University of Liège, Belgium
[13] Institut National de la Recherche Agronomique (INRA), UMR 1391 ISPA, F-33140 Villenave d'Ornon, France
[14] Forest Services, Autonomous Province of Bolzano, Via Brennero 6, I-39100 Bolzano, Italy
[15] Faculty of Science and Technology, Free University of Bolzano, Piazza Università 5, I-39100 Bolzano, Italy
[16] Research Institute for Nature and Forest (INBO), Gaverstraat 35, BE-9500 Geraardsbergen, Belgium
[17] Centre of Excellence PLECO (Plant and Vegetation Ecology), Department of Biology, University of Antwerp, BE-2610 Wilrijk, Belgium
[18] Karlsruhe Institute of Technology (KIT), Institute of Meteorology and Climate Research, Atmospheric Environmental Research (IMK-IFU), Kreuzeckbahnstr. 19, D-82467 Garmisch-Partenkirchen, Germany
[19] Institute of Soil Science and Soil Conservation, iFZ Research Centre for Biosystems, Land Use and Nutrition, Justus Liebig University Giessen, Heinrich-Buff-Ring 26-32, D-35392 Giessen, Germany
[20] University of Rennes, CNRS, UMR 6553 ECOBIO, Campus de Beaulieu, 263 avenue du Général Leclerc, F-35042 Rennes cedex, France
[21] Leibniz Centre for Agricultural Landscape Research (ZALF), Eberswalder Straße 84, D-15374, Müncheberg, Germany
[22] A.N. Severtsov Institute of Ecology and Evolution, Russian Academy of Sciences, 119071, Leninsky pr.33, Moscow, Russia
[23] Department of Meteorology, Poznań University of Life Sciences, Piątkowska 94, 60-649 Poznań, Poland
[24] Department of Matter and Energy Fluxes, Global Change Research Centre, AS CR, v.v.i. Belidla 986/4a, 603 00 Brno, Czech Republic
[25] Laboratory of Bioclimatology, Department of Ecology and Environmental Protection, Poznan University of Life Sciences, Piatkowska 94, 60-649 Poznan, Poland
[26] Finnish Meteorological Institute, Climate System Research, PL 503, FI-00101, Helsinki, Finland
[27] Weststrasse 5, D-38162 Weddel, Germany





[28] Air Quality Department, Meteorological and Hydrological Service, Gric 3, 10000 Zagreb, Croatia

[29] TNO, Environmental Modelling, Sensing & Analysis, Petten, The Netherlands

[30] Institut National de la Recherche Agronomique (INRA), UMR 1434 Silva, Site de Nancy, Rue d'Amance, F-54280 Champenoux, France

[31] Greengrass - Atmospheric Environment Expert Ltd. fellowship, Kornélia utca 14/a, 2030 Érd, Hungary

[32] Federal Research and Training Centre for Forests, Natural Hazards and Landscape, Seckendorff-Gudent-Weg 8, A-1131 Vienna, Austria

[33] Wageningen University and Research, PO Box 47, 6700AA Wageningen, The Netherlands

[34] Integrated Carbon Observation System (ICOS ERIC) Head Office, Erik Palménin aukio 1, FI-00560 Helsinki, Finland

[35] Centro de Estudos Florestais, Instituto Superior de Agronomia, Universidade de Lisboa, Tapada da Ajuda, 1349-017 Lisbon, Portugal

[36] Institute for Atmospheric and Earth System Research/Physics, Faculty of Science, POBox 68, FI-00014 University of Helsinki, Finland

[37] Gembloux Agro-Bio Tech, Axe Echanges Ecosystèmes Atmosphère, 8, Avenue de la Faculté, BE-5030 Gembloux, Belgium

[38] Global Change Research Institute, Academy of Sciences, Bělidla 4a, 603 00 Brno, Czech Republic

[39] National Research Council of Italy, Institute for Agriculture and Forestry Systems in the Mediterranean (CNR-ISAFOM), Via Patacca, 85 I-80056 Ercolano (NA), Italy

[40] Slovak Hydrometeorological Institute, Department of Air Quality, Jeseniova 17, 83315 Bratislava, Slovakia

[41] Université Grenoble Alpes, CNRS, IGE, F-38000 Grenoble, France

[42] NRE, Oberwohlenstrasse 27, CH-3033 Wohlen b. Bern, Switzerland

[43] CEFE, CNRS, Univ Montpellier, Univ Paul Valéry Montpellier 3, EPHE, IRD, Montpellier, France.

[44] Mechanical Engineering Department, Instituto Superior Técnico (Technical University of Lisbon), Ave. Rovisco Pais, IST, 1049-001 Lisboa, Portugal

[45] Fundacion CEAM, C/ Charles R. Darwin, 46980 Paterna (Valencia), Spain

[46] Department of Plant and Environmental Sciences, Faculty of Science, University of Copenhagen, Thorvaldsensvej 40, DK-1871 Frederiksberg C.

[47] Laboratory of Functional Ecology and Global Change (ECOFUN), Forest Science and Technology Centre of Catalonia (CTFC), Carretera de Sant Llorenç de Morunys, 25280 Solsona, Spain

[48] Group GAMES & Department of Horticulture, Botany and Landscaping, School of Agrifood and Forestry Science and Engineering, University of Lleida, Av. Rovira Roure 191, 25198 Lleida, Spain

[49] Norsk institutt for luftforskning, Postboks 100, 2027 Kjeller, Norway

[50] Earth and Life Institute (Environmental sciences), Université catholique de Louvain, Louvain-la-Neuve, Belgium.

[51] Department of Meteorology, Eötvös Loránd University, 1117 Budapest Pázmány Péter s. 1/A, Hungary

[52] Institute of Soil Research, Department of Forest and Soil Sciences, University of Natural Resources and Life Sciences Vienna, Peter Jordan Str. 82, A-1190 Vienna, Austria

*Correspondence to*: Chris R. Flechard (christophe.flechard@inra.fr)

**Abstract.** The impact of atmospheric reactive nitrogen ($N_r$) deposition on carbon (C) sequestration in soils and biomass of unfertilised, natural, semi-natural and forest ecosystems has been much debated. Many previous results of this dC/dN response were based on changes in carbon stocks from periodical soil and ecosystem inventories, associated with estimates of $N_r$ deposition obtained from large-scale chemical transport models. This study and a companion paper (Flechard et al., 2019) strive to reduce uncertainties of N effects on C sequestration by linking multi-annual gross and net ecosystem productivity estimates from 40 eddy covariance flux towers across Europe to local measurement-based estimates of dry and wet $N_r$ deposition from a dedicated collocated monitoring network. To identify possible ecological drivers and processes affecting the interplay between C and $N_r$ inputs and losses, these data were also combined with in situ flux measurements of NO, $N_2O$ and $CH_4$ fluxes, soil $NO_3^-$ leaching sampling, as well as results of soil incubation experiments for N and greenhouse gas (GHG) emissions, surveys of available data from online databases and from the literature, together with forest ecosystem (BASFOR) modelling.

Multi-year averages of net ecosystem productivity (NEP) in forests ranged from -70 to 826 g (C) $m^{-2}$ $yr^{-1}$ at total wet + dry inorganic $N_r$ deposition rates ($N_{dep}$) of 0.3 to 4.3 g (N) $m^{-2}$ $yr^{-1}$; and from -4 to 361 g (C) $m^{-2}$ $yr^{-1}$ at $N_{dep}$ rates of 0.1 to 3.1 g (N) $m^{-2}$ $yr^{-1}$ in short semi-natural vegetation (moorlands, wetlands and unfertilised extensively managed grasslands). The GHG budgets of the forests were strongly dominated by $CO_2$ exchange, while $CH_4$ and $N_2O$ exchange comprised a larger proportion of the GHG balance in short semi-natural vegetation. Nitrogen losses in the form of NO, $N_2O$ and especially $NO_3^-$ were of the order of 10-20% of $N_{dep}$ at sites with $N_{dep}$ < 1 g (N) $m^{-2}$ $yr^{-1}$, versus 50-80% for $N_{dep}$ > 3 g (N) $m^{-2}$ $yr^{-1}$, indicating





that perhaps one third of the sites were in a state of early to advanced N saturation. Net ecosystem productivity increased with $N_r$ deposition up to 2-2.5 g (N) m$^{-2}$ yr$^{-1}$, with large scatter associated with a wide range in carbon sequestration efficiency (CSE, defined as the NEP/GPP ratio). At elevated $N_{dep}$ levels (> 2.5 g (N) m$^{-2}$ yr$^{-1}$), where inorganic $N_r$ losses were also increasingly large, NEP levelled off and then decreased. The apparent increase in NEP at low to intermediate $N_{dep}$

levels was partly the result of geographical cross-correlations between $N_{dep}$ and climate, indicating that the actual mean dC/dN response at individual sites was significantly lower than would be suggested by a simple, straightforward regression of NEP vs. $N_{dep}$.

## 1 Introduction

The global terrestrial net sink for atmospheric carbon dioxide ($CO_2$), calculated at approximately 1.7 Pg (C) yr$^{-1}$ as the land-

based carbon (C) uptake of 3.2 ± 0.8 Pg (C) yr$^{-1}$ minus emissions from deforestation and other land-use changes of 1.5 ± 0.7 Pg (C) yr$^{-1}$, is roughly one fifth of global $CO_2$-C emissions by fossil fuel combustion and industry (9.4 ± 0.5 Pg (C) yr$^{-1}$). The ocean sink is of the same order (2.4 ± 0.5 Pg (C) yr$^{-1}$), while twice as much $CO_2$-C (4.7 ± 0.02 Pg (C) yr$^{-1}$) is added yearly to the atmosphere (Le Quéré et al., 2018). Data from atmospheric $CO_2$ inversion methods (e.g. Bousquet et al., 1999, Ciais et al., 2010), from national to global forest C inventory approaches (Goodale et al., 2002; Pan et al., 2011), and from eddy

covariance (EC) flux networks (Luyssaert et al., 2007), have suggested that a dominant part of this terrestrial $CO_2$ sink is currently occurring in forests, and especially in boreal and temperate forests of the Northern hemisphere (Ciais et al., 2010; Pan et al., 2011). Tropical forest areas are believed to be closer to carbon neutral (Pan et al., 2011), or even a net C source globally (Baccini et al., 2017), due to emissions from deforestation, forest degradation and land use change offsetting their sink potential, although others (Stephens et al., 2007) have argued that the tropical land $CO_2$ sink may be stronger – and the

Northern hemispheric land $CO_2$ sink weaker – than was generally believed. At the European scale, Schulze et al. (2010) calculated that the net biome productivity (NBP, the mean long-term carbon sink at a large spatial scale) of temperate and boreal forests was 81% of the total continental-scale land sink.

The large European and North American $CO_2$ sinks have been attributed to a combination of factors including afforestation of abandoned land and formerly cut forests, reduced forest harvest, $CO_2$ fertilisation, changes in management and age

structure legacy effects in Europe (Vilén et al., 2016), and atmospheric reactive nitrogen ($N_r$) deposition (Reay et al., 2008; Ciais et al., 2013, and references therein; De Vries et al., 2017). However, some studies (Nadelhoffer et al., 1999; Gundale et al., 2014; Fernández-Martínez et al., 2017) have questioned the widespread theory that elevated $N_r$ deposition boosts forest C sequestration, and the magnitude of the N «fertilisation» effect on forest C sequestration has been a matter of much debate (Magnani et al.,2007; Högberg, 2007; De Schrijver et al., 2008; de Vries et al., 2008; Magnani et al., 2008; Sutton et al.,

2008; Dezi et al., 2010; Binkley and Högberg, 2016). A better understanding of the impact of nitrogen deposition on natural and semi-natural ecosystems, in particular over forests, and the impact on the carbon and nitrogen cycles as an indirect effect resulting from anthropogenic activities (Canadell et al., 2007), remains key to improve the forecast of regional (de Vries et al., 2017) and global (Du and de Vries, 2018) models.

The relevance of $N_r$ deposition for the global C sequestration potential, or more explicitly the dC/dN response (change in C

storage with change in $N_r$ deposition), has been estimated typically through meta-analyses of $N_r$ addition experiments (e.g. Schulte-Uebbing and de Vries, 2018), or by combining forest growth inventories, together with estimates of Nr deposition obtained from large-scale forest monitoring plots (Solberg et al., 2009; Laubhann et al., 2009; De Vries et al., 2008). Both methods have many sources of uncertainty. One key difficulty in the latter approach lies in estimating total (wet+dry) $N_r$ deposition ($N_{dep}$), especially dry deposition, which is highly variable spatially, very challenging to measure, and

consequently hard to parameterize in regional-scale chemical transport models (CTM) (Flechard et al., 2011; Simpson et al., 2014; Schwede et al., 2018). The annual or long-term dry deposition component of $N_{dep}$ to forests, in all the diversity of N-





containing forms (gaseous vs. aerosol, reduced vs. oxidized, inorganic vs. organic, e.g. Zhang et al., 2009), has been actually measured (by micrometeorological methods) in very few forests worldwide (Neirynck et al., 2007; Erisman et al., 1996). Due to the large diversity of atmospheric compounds that contribute to total $N_r$ and the complexity of the measurement

techniques required for each compound (Flechard et al., 2011), it is even debatable that complete measurements of all $N_r$ deposition terms have ever been achieved anywhere. Thus virtually all studies of the forest dC/dN response so far have relied on modelled atmospheric $N_r$ deposition estimates, at least for the dry and occult deposition fractions, and further, that the $N_r$ deposition data being used were systematically provided by the outputs of large-scale regional (e.g. Sutton et al., 2008; Fernández-Martínez et al., 2017) or even global (Fleischer et al., 2013) models, with resolutions of typically 10 km x 10 km

or 1° x 1°, respectively. Grid averaging in such large-scale models introduces a large uncertainty in local (ecosystem-scale) $N_r$ dry deposition rates (Schwede et al., 2018), particularly when the forest sites are located near agricultural or industrial $N_r$ sources (Loubet et al., 2009; Fowler et al., 1998).

Additionally, nitrogen losses may significantly offset atmospheric $N_r$ inputs at eutrophicated and acidified sites, with the consequence that dC/dN may correlate better with net, rather than gross, atmospheric $N_r$ inputs. Depending especially on the

extent of ecosystem N saturation (De Schrijver et al., 2008), substantial N losses may occur in the form of nitrate ($NO_3^-$) leaching (Dise et al., 2009), nitric oxide (NO) and nitrous oxide ($N_2O$) emissions (Pilegaard et al., 2006), ammonia ($NH_3$) bi-directional exchange (Hansen et al., 2013), as well as emissions of di-nitrogen ($N_2$) from total denitrification (Butterbach-Bahl et al., 2002) (Fig. 1). The implication is that the carbon response to $N_{dep}$ would be non-linear, with larger dC/dN at low $N_{dep}$ rates, and a lowering of dC/dN as $N_{dep}$ increases, as suggested in the review by Butterbach-Bahl and Gundersen (2011)

and further elaborated in De Vries et al. (2014). The latter authors show in their review that above a certain N deposition level, the dC/dN response declines due to adverse effects of excess $N_r$ deposition and high soil ammonium ($NH_4^+$) concentration and nitrification (e.g. acidification, nutrient base cation losses, aluminium mobility), which are known to reduce soil fertility and affect ecosystem health and functioning (Aber, 1992).

Carbon losses through dissolved organic carbon (DOC) and biogenic dissolved inorganic carbon (DIC) leaching can also be

significant, especially for grassland and cropland ecosystems (Kindler et al., 2011; Gielen et al., 2011). This is relevant for the net ecosystem carbon balance (NECB) or the net biome productivity (NBP) estimates obtained on the basis of EC flux systems, and needs to be accounted for as a part of the net ecosystem productivity (NEP) that is not actually stored in the system (Chapin et al., 2006; Schulze et al., 2010) (Fig. 1). Dissolved and/or emitted methane ($CH_4$) may further represent a significant loss from organic soils (Hendriks et al., 2007), while $CH_4$ oxidation, which is often observed in well-aerated soils

and can be suppressed by $N_r$ addition, especially $NH_4^+$ (Steudler et al., 1989), may affect the net greenhouse gas (GHG) budget. Nitrogen deposition-induced $N_2O$ emissions from the forest floor (Pilegaard et al., 2006; Liu and Greaver, 2009), or from denitrification triggered by deposited $NO_3^-$ in peatland (Francez et al., 2011), can also offset the gain in the ecosystem GHG balance resulting from a hypothetical nitrogen fertilisation effect.

Nitrogen deposition or addition is known to affect soil microbial C cycling in many different ways, for example high level N

enrichment generally leading to reduced microbial biomass and suppressed soil $CO_2$ respiration (Treseder, 2008); a reduction of basal respiration without significant decline in total microbial biomass, following N addition to incubated peat cores (Francez et al., 2011); and added $NO_3^-$ altering directly the oxidative enzyme production by microbial communities and hence controlling extracellular enzyme activity (Waldrop and Zak, 2006). Nitrate addition can lead to a reduction in $CH_4$ emissions from wetlands and peatlands (Francez et al., 2011), since in anaerobic conditions and in the presence of $NO_3^-$ as

electron acceptor, denitrifying bacteria can oxidize organic C-substrates (*e.g.* acetate) and thus out-compete methanogenic communities (Boone, 1991). However, if chronic N enrichment of peatland ecosystems leads to floristic changes, especially an increase in vascular plants at the expense of bryophytes, the net effect may be an increase in $CH_4$ emissions (Nykänen et al., 2002), as the aerenchyma of tracheophytes provides a direct diffusion path to the atmosphere for soil-produced $CH_4$, bypassing oxidation in the peat by methanotrophs. Excess nitrogen-induced vegetation composition changes in Sphagnum





moss peatland are believed to reduce C sequestration potentials, and the effect is likely to be exacerbated by climate change (Limpens et al., 2011).

This complex web of interactions between the C and N cycles and losses shows the need for integrated approaches for studying the impacts of $N_r$ deposition on C sequestration and net GHG budgets. Ideally, all C and N gain and loss pathways (including infrequently or rarely measured fluxes such as $N_r$ dry deposition, organic C and N leaching fluxes, GHG fluxes,

etc; see Fig. 1) should be quantified at long-term experimental sites to improve and calibrate process-based models. Closing the C and N budgets experimentally at each site of large (e.g. FLUXNET) monitoring networks is unlikely to occur in the near future, but realistic and cost-effective measurement approaches can be used to progressively reduce the uncertainties for the large terms of the budgets. Such approaches were tested and implemented in this study, as part of a large-scale effort, within the NitroEurope Integrated Project (NEU, 2013; Sutton and Reis, 2011), to quantify $N_r$ deposition and N losses from

ecosystems, in parallel and coordinated with the CarboEurope Integrated Project (CEIP, 2011) to estimate the net C and GHG balance, for forest and semi-natural ecosystems in Europe.

The main aim of this paper is to build tentative C, N and GHG budgets for a wide range of European monitoring sites, and to critically examine uncertainties and knowledge gaps therein, prior to an assessment in the companion paper (Flechard et al., 2019) of the dC/dN response of C sequestration from the same datasets. To this end, we compiled the C, N and GHG flux

data from NEU, CEIP and other complementary datasets, using a combination of in situ measurements, empirical relationships, ecosystem modelling, literature and database surveys, at the scale of the CEIP and NEU flux monitoring networks. This study presents the methodologies and data, including atmospheric deposition from gas, aerosol and precipitation $N_r$ concentration monitoring, soil $NO_3^-$ leaching measurements and modelling, GHG and $N_r$ emission estimates from chamber measurements and laboratory-based soil bioassays, EC tower-based C budgets, as well as historical published

data. Forest ecosystem modelling (BASFOR) is used to simulate C, N and GHG fluxes, with the double objective to compare with actual measurements and to fill some gaps in the datasets. Wherever possible, alternative measurements, datasets or modelled data are shown alongside the primary data in order to provide an estimate of the uncertainty in the different terms.

*{Insert Figure 1 here}*

## 2 Materials and methods

**2.1 Monitoring sites**

The study comprised 40 terrestrial ecosystem-scale, carbon and nitrogen flux monitoring sites, including 31 forests (F) and 9 natural or semi-natural (SN) short vegetation ecosystems, primarily moorlands, wetlands and extensively managed, unfertilised grasslands (Table 1). The sites spanned a European geographical and climatic gradient from the Mediterranean to the Arctic and from the Atlantic to western Russia (Fig. S1), an elevation range of -2 m to 1765 m a.m.s.l., a mean annual

temperature (MAT) range of -1.0°C to 17.6°C, and a mean annual precipitation (MAP) range of 500 mm to 1365 mm. Selected references are provided for each site in Table S1. A list of the main acronyms and abbreviations used in the paper is provided in Table 2.

*{Insert Table 1 here}*

*{Insert Table 2 here}*

The forest sites of the study ranged from very young (< 10 years old) to mature (> 150 years old), and can be broadly classified into four plant functional types (PFT) or five dominant tree categories (Table 1): deciduous broadleaf (DB), evergreen needle-leaf (EN, comprising mostly spruce and pine species), mixed deciduous/coniferous (MF), and Mediterranean evergreen broadleaf (EB). Forest species composition, stand characteristics, C and N contents of different ecosystem compartments (leaves, wood, soil), soil physical properties and micro-climatological characteristics are described

in Tables S2, S3, S4 and S5. Semi-natural short vegetation ecosystems included unimproved (mountainous and semi-arid)





grasslands, wetlands and peatlands; they are included in the study as unfertilised, C-rich soil systems, providing a contrast with forests where storage also occurs above ground (thus with different C/N ratios). Among the 40 EC-CO$_2$ flux measurement stations, most sites (36) were part of the CEIP CO$_2$ flux network. A further three CO$_2$ flux sites were operated as part of the NEU network (EN2, EN16, and SN3), and one site (DB4) was included from the French F-ORE-T observation

network (F-ORE-T, 2012). Table S6 provides an overview of the available C, N and GHG flux measurements, detailed hereafter.

## 2.2 Nitrogen fluxes

Input and output fluxes of the ecosystem nitrogen and carbon budgets are represented schematically in Fig. 1. The following sections describe the methods used to quantify the different terms.

### 2.2.1 Atmospheric deposition

To obtain realistic estimates of total (dry + wet) N$_r$ deposition at the 40 sites of the network, it was necessary to measure ambient air concentrations of the main N-containing chemical species at each location, due to the large spatial heterogeneity in gas phase concentrations, especially for NH$_3$. The requirement for local measurements of wet deposition was relaxed because this is much less spatially variable. For both dry and wet components, measurements had to be complemented by

models, either to calculate fluxes based on local concentration data at each site, or to obtain local estimates from a large-scale CTM when data were missing.

Atmospheric inorganic N$_r$ concentrations, available from the NEU (2013) database, were measured monthly for 2-4 years in the gas phase (NH$_3$, HNO$_3$, HONO) and in the aerosol phase (NH$_4^+$, NO$_3^-$), using DEnuder for Long-Term Atmospheric sampling (DELTA) systems (Sutton et al., 2001; Tang et al., 2009). Concentrations of nitrogen dioxide (NO$_2$), not covered

by DELTA sampling, were measured by chemiluminescence at a few sites only, and were otherwise taken from gridded concentration outputs of the European-scale EMEP CTM (details given below). The N$_r$ data initially reported in Flechard et al. (2011) covered the first 2 years of the NEU project (2007-2008); here, the data from the entire 4-yr NEU monitoring period (2007-2010) were used and averaged to provide a more robust long-term 4-year estimate of N$_r$ dry deposition. The inferential modelling method was used to calculate dry deposition for N-containing gas and aerosol species, whereby

measured ambient N$_r$ concentrations were multiplied by a vegetation-, meteorology- and chemical species-dependent deposition velocity (V$_d$) (Flechard et al., 2011, 2013; Bertolini et al., 2016; Thimonier et al., 2018). In the case of NH$_3$, a canopy compensation point scheme was applied in some models, allowing bi-directional exchange between the surface and the atmosphere. Considering notoriously large uncertainties in deposition velocities and large discrepancies between the surface exchange schemes currently used in different CTMs, we tried here to minimise such uncertainties by using the

ensemble average dry deposition predicted by four different models, as in Flechard et al. (2011).

For wet deposition, several sources of data were used, and the final wet deposition estimate was derived from the arithmetic mean of the different sources, where available. First, within the NEU project, a survey was made of the available national and/or trans-national (e.g. EMEP, 2013; ICP Forests Level-II, ICP, 2019) wet deposition monitoring network concentration data for inorganic N (NH$_4^+$, NO$_3^-$) in the different European countries hosting one or several CEIP/NEU flux sites. These

data were checked for consistency and outliers, harmonized, and then spatially interpolated by kriging to provide measurement-based estimates of solute concentrations in rainfall for each of the 40 sites of this study. Wet deposition was then calculated as the product of interpolated concentration times measured precipitation at each site.

Next, thirteen sites (DB1, DB3, DB4, EN4, EN9, EN13, EN14, EB2, EB3, MF1, MF2, SN3, SN8) were identified as lacking local or nearby wet deposition measurements. These sites were equipped for three years (2008-2010) with bulk (open funnel)

precipitation samplers (Model B, Rotenkamp, Germany; Dämmgen, 2006), mounted above the canopy or inside a clearing for some of the forest sites, with monthly sample change and analysis. The precipitation samples were stabilized by addition





of thymol at the beginning of each exposure period, and were analyzed subsequently for inorganic $N_r$ ($NH_4^+$ and $NO_3^-$) as well as $SO_4^{2-}$, $Cl^-$, $PO_4^{3-}$, base cations ($Mg^{2+}$, $Ca^{2+}$, $K^+$, $Na^+$) and pH. A few other sites (EN2, EN8, EN10, EN16, DB2, SN9) were already equipped with wet-only or bulk precipitation collectors. No correction was applied to the bulk deposition

estimates to account for a possible contribution by dry deposition within the sampler glass funnel (e.g. Dämmgen et al., 2005), since there did not appear to be any systematic overestimation compared with wet deposition estimates from the monitoring networks or EMEP data (Fig. S2), even if a more significant bias may be expected in dry (Mediterranean) regions.

In addition to inorganic nitrogen, the wet deposition of water-soluble organic $N_r$ (WSON) compounds was also investigated

in precipitation samples at 16 sites (Cape et al., 2012). However, since WSON data were not available for all sites and the measurements were subject to considerable uncertainties (Cape et al., 2012), and also because the contribution of WSON to total $N_r$ deposition was on average less than 5%, WSON was not included in the final estimates of total $N_r$ deposition.

The last data source was the ca. 50 km x 50 km gridded modelled wet inorganic $N_r$ deposition (also $NO_2$ concentrations, discussed above), simulated by the European-scale EMEP CTM (Simpson et al., 2006a, 2006b, 2012, 2014) for the years

2007-2010, available from EMEP (2013). The data were downloaded in 2013, and it should be noted that in this data series different model versions were used for the different years. This leads to some uncertainty, especially in the dry deposition estimates, but it is hard to say which model version is the most realistic. Evaluation of the model against measurements over this period has shown quite consistent results for the wet-deposited components and $NO_2$ concentrations, but the dry deposition rates cannot be evaluated versus actual measurements at the European scale. We chose therefore to make use of

all versions and years, giving a small ensemble of simulations.

### 2.2.2 Soil gaseous and leaching losses

Nitrogen losses to the atmosphere (gaseous emissions) and to groundwater (N leaching), which are especially hard to quantify and thus typically cause large uncertainties in ecosystem N budgets, were estimated by direct flux measurements or by indirect empirical methods. Soil NO and $N_2O$ emissions were measured in the field using closed static and dynamic

chamber methods, as part of NEU (e.g. EN2, EN10, EN16, DB2, SN3, SN8, SN9) and/or collected from the literature (e.g. EN2, EN10, EN14, EN16, DB2, Pilegaard et al., 2006; long term data at EN2 in Luo et al., 2012). Such data were available for $N_2O$ at seven forest sites and four semi-natural sites, and at five forest sites for NO (Table S6). Manual static chamber $N_2O$ measurements were made manually at a typically bi-weekly (growing season) or monthly (winter half-year) frequency at many sites. Automatic chamber systems, allowing continuous $N_2O$ measurements at a frequency of four times per day,

were deployed at EN2, EN10, DB2 and SN3. Fluxes of NO were only measured by automatic dynamic (open) chambers. Measured fluxes were scaled up to yearly values by linear interpolation or using the arithmetic mean of all flux measurements.

To address the lack of direct in situ $N_r$ and non-$CO_2$ GHG gas flux measurements at many sites, soil $N_2O$, NO (and also $CH_4$) fluxes were also estimated, as part of NEU, from the temperature and moisture responses of soils. These responses were

established in a series of factorial soil incubation experiments in controlled conditions with four levels of temperature (5-20°C) and water-filled pore space (20-80 WFPS%), following the protocol described in Schaufler et al. (2010). Twenty-four undisturbed soil cores (top 5 cm of the mineral soil, Ah horizon) were taken from each of 27 forests and 8 semi-natural sites in spring after soils had warmed up above 8°C for one week in order to guarantee phenological comparability of the different climatic zones. Sampling was conducted in 2008, 2009 and 2010 and cores were sent to a common laboratory at the Federal

Research and Training Centre for Forests (BFW, Vienna, Austria) for the controlled environment bioassays, which were carried out straight away. The 5 cm top soil layer was selected as it represents the highest microbial activity and correspondingly high GHG production/consumption rates, although processes in deeper soil layers should not be neglected (Schaufler et al., 2010). Site-specific, empirical bi-variate (T, WFPS) relationships describing soil fluxes for $CO_2$, $N_2O$, NO





and $CH_4$ were derived from the incubation results and then applied to multi-annual time series of soil temperature and moisture measured at the sites, mimicking field conditions and providing scaled up estimates of potential annual trace gas emissions.

Leaching of dissolved inorganic nitrogen (DIN = $NH_4^+ + NO_3^-$) was measured using lysimeter setups, or estimated from a combination of suction cup measurements (typically ~1m soil depth) and a hydrological drainage model, at a few sites during the NEU monitoring period (EN2, EN10, EN16, DB2) and as part of parallel projects (EN4, EN8, EN15, DB1, DB4).

For the forest sites where no leaching measurements were available, the empirical algorithm by Dise et al. (2009) was applied to predict DIN leaching based on key variables (throughfall inorganic $N_r$ deposition $DIN_{TF}$, organic horizon C/N ratios, MAT). The algorithm, developed from the extensive Indicators of Forest Ecosystem Functioning (IFEF) database (>300 European forest sites), simulates the non-linearity of DIN leaching with respect to $DIN_{TF}$ and soil C/N ratio, with critical thresholds for the onset of leaching of $DIN_{TF} = 0.8$ g (N) $m^{-2}$ $yr^{-1}$ and C/N = 23, respectively. Since the algorithm

requires $DIN_{TF}$ as input, as opposed to total (above canopy) $N_{dep}$, in the present study we applied a reduction factor of 0.85 from $N_{dep}$ to $DIN_{TF}$ (i.e. a canopy retention of 15% of atmospheric N), which was calculated as the average of all available individual $DIN_{TF}/N_{dep}$ ratios in the IFEF database. A comparison with values of $DIN_{TF}/N_{dep}$ ratios actually measured at the EN2, EN8, EN10, EN16 and DB2 sites (0.71, 0.80, 0.29, 0.85, 1.11, respectively; mean ± st. dev. 0.75 ± 0.30) shows that the applied ratio of 0.85 is plausible but also that much variability in canopy retention/leaching may be expected between sites.

**2.3 Carbon fluxes**

**2.3.1 Ecosystem-atmosphere $CO_2$ exchange**

Half-hourly rates of net ecosystem-atmosphere $CO_2$ exchange (NEE) were measured over several years (on average 5 years; see Table S6) by the eddy covariance (EC) technique at all sites. The long term net ecosystem productivity (NEP) is defined following Chapin et al. (2006) as the difference between gross primary production (GPP) and ecosystem respiration ($R_{eco}$),

and thus calculated as the straightforward annual sum of NEE fluxes (with opposite sign). The net ecosystem carbon balance (NECB) may differ from the NEP if C fluxes other than assimilation and respiration, such as DIC/DOC leaching, $CH_4$ and other volatile organic compound (VOC) emissions, as well as lateral fluxes (harvest, thinning) and other disturbances (fire), are significant over the long term (Chapin et al., 2006). For convenience in this paper, we use the following sign convention for $CO_2$ fluxes: GPP and $R_{eco}$ are both positive, while NEP is positive for a net sink (a C gain from an ecosystem perspective)

and negative for a net source.

The EC technique is based on fast-response (sampling rates typically 10-20Hz) open-path or closed-path infra-red gas analyzer (IRGA) measurements of turbulent fluctuations in $CO_2$ concentration ($c$) in the surface layer above the ecosystem, coupled with ultra-sonic anemometer measurements of the three components of wind ($u$, $v$, $w$) and temperature. The NEE flux is calculated as the average product of $c$ and $w$ fluctuations, i.e. the covariance (Swinbank, 1951; Lee et al., 2004).

The EC-$CO_2$ flux measurements reported here followed the protocols established during the CEIP project, largely based on the EUROFLUX methodology (Aubinet et al., 2000). Briefly, post-processing of the raw high frequency EC data included typically: de-spiking to remove outliers; 2-D rotation of the coordinate system; time lag optimization by maximization of the covariance between $CO_2$ concentration and vertical component of wind speed (w); block-averaging over the flux averaging interval of 30 minutes. Corrections were applied for various methodological artefacts, including notably i) flux losses at the

different frequencies of flux-carrying eddies, caused e.g. by attenuation/damping in the inlet/tubing system (Ibrom et al. 2007; Fratini et al. 2012), path averaging, sensor separation, analyzer response time, high- and low-pass filtering; ii) effects of temperature fluctuations and dilution by water vapor on measured fluctuations in concentrations of $CO_2$ (Webb-Pearman-Leuning corrections; Webb et al., 1980); iii) $CO_2$ storage below sensor height. Quality assurance and quality control procedures were further developed and agreed upon within CEIP, including statistical tests, non-stationarity, integral

turbulence characteristics (Foken et al., 2004), and footprint evaluation (Göckede et al., 2008). Friction velocity ($u_*$)



threshold filtering was implemented using the moving point test according to Papale et al. (2006) and as described in REddyProc (2019), in order to discard flux data from periods of low turbulence.

Different EC post-processing softwares were used at the different sites within the project, such that the data were not evaluated in exactly the same way across the CEIP network, but a reasonably good overall agreement was found among the

different softwares, within 5-10% difference for 30-minute $CO_2$ flux values (Mauder et al., 2008; Mammarella et al., 2016). Similarly, for the gap-filling of the 30-minute flux time series, during periods of instrument malfunction or unsuitable measurement conditions (low turbulence, insufficient fetch, etc.), and for the partitioning of NEP into GPP and $R_{eco}$, a number of alternative algorithms have been developed in the past, based on different sets of principles (Falge et al., 2001; Barr et al., 2004; Reichstein et al., 2005; Lasslop et al., 2010). The gap-filling and partitioning algorithm used by default in

this study was the generic online REddyProc (2019) software, implemented also in the European Fluxes Database Cluster. REddyProc was based on i) Reichstein et al. (2005) for the filling of gaps in the NEE flux data on the basis of information from environmental conditions; ii) Reichstein et al. (2005) for the nighttime data based $R_{eco}$ parameterization (using an Arrhenius-type function of temperature); and iii) on Lasslop et al. (2010) for the daytime data based GPP evaluation (using a rectangular hyperbolic light–response curve for NEE and including a temperature sensitivity of respiration and limitation of

GPP by vapour pressure deficit).

In this study, for all CEIP flux sites, we have retrieved the fully analysed and validated half-hourly (level-3) and daily to annual (level-4) $CO_2$ flux (NEP, GPP, $R_{eco}$) data as available, initially from the CEIP database, later from the European Fluxes Database Cluster (2012) or from the GHG-Europe portal (GHG-Europe, 2012). For these data, although the evaluation methods were not necessarily harmonized between sites, we hold that the data available in the database were

obtained using the best possible, state-of-the-art evaluation methods at the time of retrieval. For the four non-CEIP flux sites, flux evaluation closely followed CEIP protocols; in the case of DB4 the EddyPro (v6.2) software was used, which was based on a synthesis of calculation and correction methods from CEIP and other FLUXNET flux networks around the globe.

The EC-$CO_2$ flux measurements used in this study mostly spanned the 5-year period of CEIP (2004-2008), except for a dozen sites where measurements continued until 2010, i.e. the end of NEU and of atmospheric $N_r$ sampling. Older EC data (since the mid-late 1990's) were also available at DB5, EN6 and EN13. Data collection started and ended later at DB4, at

which both EC-$CO_2$ flux and DELTA-$N_r$ measurements spanned the 7-year period 2009-2015. Data analyses presented in the paper, based on inter-annual mean $CO_2$ budgets and mean $N_r$ deposition, assume that five or more years of monitoring yield reasonably robust estimates of long-term fluxes for the different sites, and that the small time shift between CEIP and NEU project periods (2-3 year overlap) does not affect the results significantly. At some sites such as DB2, long-term NEE measurements showed multi-decadal variations (Pilegaard et al. 2011; Wu et al. 2013), thus it was essential to use the years

overlapping with NEU.

### 2.3.2 Soil $CO_2$ and $CH_4$ fluxes

In situ soil $CO_2$ efflux (SCE) measurements by opaque (static or dynamic) manual chambers were carried out at 24 of the forest sites, with typically weekly to monthly sampling frequency, with fluxes being measured continuously (hourly) by

automated chambers at a few sites (e.g. EN2). The SCE is usually considered a proxy for $CO_2$ production by soil respiration ($R_{soil}$), though the two may not be equal as part of the $CO_2$ production is dissolved into pore water and may reach the atmosphere only later, either on-site, or even off-site if dissolved $CO_2$ (DIC) leaches to groundwater. Annual $R_{soil}$ data, scaled-up from SCE measurements, are available for 18 forest sites and were collected from the CEIP or GHG Europe databases and/or from various peer-reviewed publications for the different sites (see Table S7). The ratio of heterotrophic

respiration ($R_{het}$) to $R_{soil}$ was determined on an annual scale at 15 sites by different techniques (root-exclusion meshes, trenching experiments, radiocarbon or stable isotope tracing, tree girdling; e.g. Subke et al., 2006) (Table S7).



Methane fluxes were measured by chamber methods or eddy covariance at six forest sites and five semi-natural (peatland, wetland) sites (Hendriks et al., 2007; Skiba et al., 2009; Drewer et al., 2010; Shvaleva et al., 2011; Luo et al., 2012; Kowalska et al., 2013; Juszczak and Augustin 2013) (Table S6). These data were complemented by bioassay measurements

of $CH_4$ emission or uptake (net oxidation) by the laboratory soil cores, as described previously for NO and $N_2O$ estimates (Schauffler et al., 2010).

### 2.3.3 Dissolved carbon losses

Dissolved inorganic (excluding $CO_2$ from weathering of carbonate rocks) and organic carbon (DIC/DOC) fluxes were measured at six forest sites (DB1, DB2, EN4, EN8, EN10, EN15), using suction cups for sampling soil water and combined

with soil drainage data, or by monitoring water runoff through weirs, as part of CEIP, NEU and other projects (Ilvesniemi et al., 2009; Kindler et al., 2011; Gielen et al., 2011; Verstraeten et al., 2014). Data were also available for peatland at SN7, with DIC, DOC and also dissolved $CH_4$ concentrations in pore water of the clayey peat, in groundwater from the sand aquifer and in ditch water, as described in Hendriks et al. (2007). For the peatland within SN9, Dinsmore et al. (2010) measured stream concentrations and export of DIC, DOC as well as particulate organic carbon (POC), and also estimated

stream evasion of $CO_2$, $CH_4$ and $N_2O$ in addition to the land-based flux (EC, chamber) measurements in the tower footprint.

### 2.4 Ecosystem greenhouse gas balance

Net GHG budgets were constructed from inter-annual mean EC-based NEP combined with measured and scaled up $N_2O$ and $CH_4$ fluxes wherever available (nine and six sites, respectively), or with bioassay-derived fluxes (most sites) or modelled data (BASFOR, forests/$N_2O$ only), using 100-yr global warming potentials (GWP) of 265 and 28 for $N_2O$ and $CH_4$,

respectively (Fifth Assessment Report, IPCC, 2013). The sign convention for non-$CO_2$ GHG fluxes and for the net ecosystem GHG balance in this paper adopts an atmospheric warming perspective, i.e. positive fluxes for emissions toward the atmosphere (warming), negative for uptake by the surface (cooling).

### 2.5 Ancillary soil, plant and ecosystem measurements

Ancillary data were collected mainly for the purpose of assembling input parameters and calibration datasets for forest ecosystem (BASFOR) modelling (see below). Texture (% clay, % sand, % silt), pH, soil organic carbon concentration (SOC)

and C/N ratios were measured in soils of 35 sites as part of the bioassay experiments described previously, but were otherwise also documented in the CEIP database and in papers previously published for the majority of sites. For the forest sites, ecosystem data for soil water content (SWC), porosity, saturation water content ($\Phi_{SAT}$), field capacity ($\Phi_{FC}$) and wilting point ($\Phi_{WP}$), and for canopy height (H), leaf area index (LAI), diameter at breast height (DBH), basal area (BA),

number of trees per unit area or stand density (SD) and thinning events, were obtained from CEIP and other project (e.g. FLUXNET) databases and complemented by various publications. Such was also the case for ecosystem carbon stocks in soil organic matter (CSOM) and in roots (CR), stems (CS), branches (CB), leaves (CL) and litter layers (CLITT), for which the global database assembled by Luyssaert et al. (2007) provided additional data. At sites for which published values of $\Phi_{FC}$ and $\Phi_{WP}$ were not available, default estimates were inferred from soil texture by means of van Genuchten (1980) pedo-

transfer functions, using tabulated values from the German soil description handbook (Eckelmann et al., 2005)

Foliar C and N contents (LeafC, LeafN) were measured as part of NEU for EN1, EN2, EN5, EN8, EN10, EN15, EN16, DB2 (Wang et al., 2013), DB4, SN3, SN4, SN8 and SN9, or were otherwise taken from CEIP, GHG Europe and FLUXNET databases as well as various publications; in total, leaf C/N measurements were available for 31 sites. By contrast, data were much rarer for C/N ratios for other compartments of the forest ecosystem, with data available at only 15 sites for litter, and

only five sites for roots, stems and branches.





### 2.6 BASFOR forest ecosystem model

### 2.6.1 General description

The BASic FORest model, BASFOR, is a process-based, deterministic forest ecosystem model, which simulates the growth and biogeochemistry (C, N and water cycles) of temperate deciduous and coniferous stands at a daily time step (van Oijen et al., 2005; Cameron et al., 2013, 2018). Model code and documentation are available on GitHub (BASFOR, 2016). Interactions with the atmospheric and soil environments are simulated in some detail, including the role of management (thinning or pruning). BASFOR is a one-dimensional model, i.e. no horizontal heterogeneity of the forest is captured, and BASFOR does not simulate some variables which are important in forest production, such as wood quality or pests and diseases.

Nine state variables for the trees describe i) C pools: leaves, branches, stems, roots, reserves (CL, CBS or collectively CLBS, CR, CRES; kg (C) m$^{-2}$); ii) N pool in leaves (NL; kg (N) m$^{-2}$); and iii) Stand density (SD, trees m$^{-2}$), tree phenology (only for deciduous trees): accumulated chill days (chillday; d) and accumulated thermal time (Tsum; °C d). Seven state variables for the soil can be divided into three categories, according to the three biogeochemical cycles being simulated: i) C pools in litter layers of the forest floor (CLITT), soil organic matter (SOM) with fast turn-over (CSOMF), SOM with slow turn-over (CSOMS) (kg (C) m$^{-2}$); ii) N pools as for C but also including mineral N (NLITT, NSOMF, NSOMS, NMIN; kg (N) m$^{-2}$); and iii) the water pool: amount of water to the depth of soil explored by the roots (WA; kg H$_2$O m$^{-2}$ = mm).

Carbon enters the system via photosynthesis, calculated as the product of photosynthetically active radiation (PAR) absorption by the plant canopy and light use efficiency (LUE). The leaf and branch pools are subject to senescence, causing carbon flows to litter. Roots are also subject to senescence, causing a flow to fast-decomposing soil organic matter. Litter carbon decomposes to fast-decomposing soil organic matter plus respiration. Fast-decomposing soil organic matter decomposes to slow-decomposing soil organic matter plus respiration. Finally, the slow organic carbon pool decomposes very slowly to CO$_2$. Nitrogen enters the system in mineral form through atmospheric deposition. Nitrogen leaves the system through leaching and through emission of N$_2$O and NO from the soil to the atmosphere. N$_2$ losses from denitrification and biological N$_2$ fixation are not simulated. Mineral nitrogen is taken up by the trees from the soil, and nitrogen returns to the soil with senescence of leaves, branches and roots, and also when trees are pruned or thinned. Part of the N from senescing leaves is re-used for growth. The availability of mineral nitrogen is a Michaelis-Menten function of the mineral nitrogen pool and is proportional to root biomass. Transformation between the four soil nitrogen pools are similar to those of the carbon pools, with mineral nitrogen as the loss term. Water is added to the soil by precipitation and lost through transpiration, evaporation, and drainage. Evaporation and transpiration are calculated using the Penman equation, as functions of the radiation intercepted by soil and vegetation layer, and atmospheric temperature, humidity and wind speed. Drainage of ground water results from water infiltration exceeding field capacity of the soil.

The major inputs to the model are daily time series of weather variables (global radiation, air temperature, precipitation, wind speed and relative humidity). The last two of these are used in the calculation of potential rates of evaporation and transpiration. Soil properties, such as parameters of water retention (field capacity, wilting point, soil depth) are provided as constants. Further, the model requires time series indicating at which days the stand was thinned or pruned. The model outputs include, amongst others, the state variable for trees and soil as well as evapotranspiration (ET), groundwater recharge, canopy height (H), leaf area index (LAI), diameter at breast height (DBH), GPP, R$_{eco}$ and R$_{soil}$, NEP, N mineralisation, N leaching, NO and N$_2$O emissions.

### 2.6.2 Model implementation and calibration

The primary purpose of BASFOR in this paper was to provide estimates of NO and N$_2$O emissions as alternatives to measured fluxes where chamber and bioassay data were missing for the calculation of the N balance (Fig. 3); the modelled



C fluxes (Fig. 6) were not used for GHG balance calculations since EC-$CO_2$ data were available throughout. BASFOR simulations of forest growth and C, N and $H_2O$ fluxes were made for all CEIP/NEU forest sites from planting (spanning the interval 1860-2002), until the end of the NEU project (2011). At a few sites, natural regeneration occurred, but for modelling

purposes a planting date was assigned based on the age of the trees. Meteorological data measured at each site over several years since the establishment of the flux towers (typically 5-10 yr) were replicated backwards in time in order to generate a time series of model inputs for the whole period since planting. Assumptions were made that inter-annual meteorological variability was sufficiently covered in the span of available measurements and that the impact of climate change since planting was small and could be neglected.

The atmospheric $CO_2$ mixing ratio was provided as an exponential function of calendar year, fitted to Mauna Loa data since the beginning of records in 1958 (NOAA, 2014) and extrapolated backwards to around 1860-1900 for the oldest forests included in this study. The global $CO_2$ mixing ratio driving the model thus increased from around 290 ppm in 1900, to 315 ppm in 1958, to 390 ppm in 2010. Similarly, atmospheric $N_r$ deposition was a key input to the model and was forced to vary over the lifetimes of the planted forests; $N_{dep}$ was assumed to rise from pan-European levels well below 0.5 g (N) m$^{-2}$ yr$^{-1}$ at

the turn of the 20$^{th}$ century, to increase sharply after World War II to reach an all-time peak around 1980, and to decrease subsequently from peak values by about one third until 2005-2010, at which point the NEU $N_{dep}$ estimates were obtained. We assumed that all sites of the European network followed the same relative time course of $N_{dep}$ over the course of the 20$^{th}$ century, taken from van Oijen et al. (2008), but scaled for each site using the NEU $N_{dep}$ estimates (Fig. S4).

Forest management was included as an input to the model in the form of a prescribed time course of stand density and

thinning from planting to the present date. Tree density was known at all sites around the time of the CEIP/NEU projects (Table S2), but information on thinning history since planting (dates and fractions removed) was much sparser. A record of the last thinning event was available at only one third of all sites, and a knowledge of the initial (planting) density and a reasonably complete record of all thinning events were available at only a few sites. For the purposes of BASFOR modelling, we attempted to recreate a plausible density and thinning history over the lifetime of the stands. The guiding

principle was that after the age of 20 years one could expect a decadal thinning of the order of 20%, following Cameron et al. (2013), while the initial reduction was 40% during the first 20 years. In the absence of an actual record of planting density (observed range: 1400-15000 trees ha$^{-1}$), a default initial value of 4500 trees ha$^{-1}$ was assumed (for around two thirds of the sites). The general principles of this default scheme were then applied to fit the available density and thinning data for each site, preserving all actual data in the time series while filling in the gaps by plausible interpolation. The density time courses

thus obtained, underlying all subsequent model runs, are shown in Fig. S5.

BASFOR was calibrated for three groups of site: DBF, ENF-spruce (EN1-7) and ENF-pine (EN8-18), by means of a multiple site Bayesian calibration (BC) procedure, described in detail in Cameron et al. (2018), using as reference data the measured ecosystem C/N/$H_2O$ fluxes and pools from the CEIP/NEU network.

## 3 Results

### 3.1 Nitrogen inputs and outputs

#### 3.1.1 Nitrogen deposition

Total inorganic $N_r$ deposition ranged from 0.1 to 4.3 g (N) m$^{-2}$ yr$^{-1}$ across the CEIP/NEU networks (Table 1), with the largest values observed in The Netherlands, northern Belgium and southern Germany, and the lowest levels observed at latitudes > 60°N (Fennoscandia). Nitrogen deposition was dominated by the dry fraction in forests (Fig. 2), with an average contribution

to total deposition of 63% versus 39% for short semi-natural vegetation. This contribution was even larger (> 2/3) for high deposition sites ($N_{dep}$ > 2 g (N) m$^{-2}$ yr$^{-1}$). Total $N_{dep}$ was more strongly correlated to dry deposition across all sites (R$^2$ = 0.94) than to wet deposition (R$^2$ = 0.56). Important differences in the ratio of dry to wet deposition are evident across climatic





regions, with the share of dry deposition being especially large at Mediterranean sites (e.g. Sanz et al., 2002), where annual rainfall is smaller. However, the share of dry deposition was also larger for sites that are located near (large) anthropogenic

(industrial, vehicular, agricultural) $N_r$ emission sources. Total $N_r$ deposition was around 25% smaller on average at short semi-natural vegetation sites compared with forests (Fig. S2), even though the mean total atmospheric $N_r$ concentrations (reduced and oxidized, N-containing gas and aerosol compounds) were quite similar between the two data sets (Flechard et al., 2011), the difference being driven by higher dry deposition rates over forests due to higher aerodynamic roughness and deposition velocities (Fig. S3; see also Schwede et al., 2018). Reduced $N_r$ ($NH_3$ gas and $NH_4^+$ in aerosol and rain,

collectively $NH_x$) contributed on average 56% of total deposition; oxidised $N_r$ ($HNO_3 + NO_2$ gas and $NO_3^-$ in aerosol and rain, collectively $NO_y$) was dominant at only six forest sites of the network (EN7, EN10, EN18, EB2, SN3, SN5; Fig. 2).

For comparison, dry deposition, calculated here as the ensemble average of four inferential model estimates based on in situ $N_r$ concentration measurements, was on average more than a factor of two larger than the ca. 50 km x 50 km grid square-averaged EMEP model estimate (taken from EMEP, 2013) (see Fig. S2). However, since each EMEP grid square contains

variable proportions of different land uses with different deposition velocities, it is more meaningful to compare DELTA-based inferential estimates for each study site with ecosystem-specific EMEP dry deposition rates in the relevant grid squares. In this case, the EMEP dry deposition rates are 32% smaller than the inferential estimates. This difference reflects discrepancies and uncertainties in the four dry deposition schemes used (Flechard et al., 2011); the mean coefficient of variation (CV = $\sigma/\mu$) between the four inferential model estimates was 36%, i.e. larger than the difference between

ecosystem-specific EMEP values and the mean inferential estimates. Other sources of discrepancy between the two methods include the use of measured *vs.* modelled meteorology to drive the deposition models, and site-specific *vs.* generic values of canopy height and leaf area index, as discussed in Flechard et al. (2011).

By contrast, wet deposition was generally reasonably consistent between the different data sources for inorganic $N_r$ (in situ bulk or wet-only measurement, kriging of monitoring network data, EMEP model output). For the 18 sites where all three

sources of data were available, the mean CV of the three estimates was 21% (range 2%-56%, with 15 CV values out of 18 below 30%), and the mean (± 95% conf. int.) wet deposition estimates across the 18 sites were 0.63 ±0.14, 0.64 ±0.15 and 0.68 ±0.16 g (N) m$^{-2}$ yr$^{-1}$ for the three methods, respectively (Fig. S2), showing no systematic bias between mehtods. Wet deposition of organic nitrogen (WSON), measured at 16 sites, represented on average 11% (range 2-36%) of total inorganic + organic wet deposition (Fig. S2), but only 4% (range 1-30%) of total dry + wet $N_r$ deposition, since total $N_{dep}$ was

dominated by dry deposition at most forest sites.

*{Insert Figure 2 here}*

### 3.1.2 Nitrogen losses

Total ecosystem losses of inorganic $N_r$ were computed for the forest sites as the sum of DIN leaching and NO and $N_2O$ emissions (Fig. 3 A-D). We assumed that $NH_3$ emissions by soil and vegetation were negligible due to generally acidic forest

soils, as well as low values of stomatal compensation point (the leaf $NH_3$ emission potential), respectively (Flechard et al., 2013). Inorganic $N_r$ losses (Fig. 3D) increased sharply with $N_r$ deposition and were largely dominated by DIN leaching at $N_{dep}$ levels above 2 g (N) m$^{-2}$ yr$^{-1}$ (Fig. 3C). For these large $N_{dep}$ levels, the fraction of deposited $N_r$ lost as DIN, NO or $N_2O$ was generally larger than 50% (Fig. 3F). The inorganic $N_r$ balance ($N_r$ deposition minus NO, $N_2O$ and DIN losses) was probably still positive for most sites (Fig. 3E), although the confidence intervals of the budget term (accounting for uncertainties in all terms including deposition) were very large for the elevated $N_r$ deposition sites. Note that the DIN

leaching estimate by BASFOR, shown for comparison on Fig. 3C, was not used in the calculation of total inorganic N losses in Fig. 3D; this is because BASFOR does not simulate $N_2$ loss by denitrification, and thus part of the soil N surplus that would in reality denitrify is assumed to drain, resulting in an over-estimation of the leaching term, though not necessarily of the total N losses.





Emissions of NO estimated from bioassay measurements (Schaufler et al., 2010) and by BASFOR modelling were generally
of the same order in forests (average values across all forest sites of 0.22 and 0.21 g (N) m$^{-2}$ yr$^{-1}$, respectively), but validation
by in situ chamber flux data was difficult owing to the limited number of available measurements (only five forest sites,
mean value 0.27 g (N) m$^{-2}$ yr$^{-1}$). Nonetheless, the largest NO emissions by the three methods were all found at N$_{dep}$ levels
above 2 g (N) m$^{-2}$ yr$^{-1}$. By contrast, N$_2$O emissions did not show any marked dependence on N$_{dep}$ and were on average

smaller than NO emissions by a factor of two to five, with mean values across all sites of 0.12, 0.08 and 0.04 g (N) m$^{-2}$ yr$^{-1}$
for bioassay, BASFOR and chamber fluxes, respectively. The mean N$_2$O fluxes (averaged over the different methods) were
larger than mean NO fluxes at only one third of the forests sites; by contrast, at SN sites N$_2$O emissions were larger than NO
emissions at all but one location. The dominance of NO over N$_2$O in forests could in principle reflect the generally well
aerated conditions of (especially coniferous) forest litter layers on well-drained top soils, more conducive to NO formation

by nitrification than N$_2$O by denitrification (Davidson et al., 2000; Pilegaard et al., 2006). This would be perhaps especially
true for the four highest (>3 g (N) m$^{-2}$ yr$^{-1}$) N$_r$ deposition sites (EN2, EN8, EN15, EN16, all coniferous forests) with the
highest NO emissions (Fig. 3), which all had sand-dominated (64-96%) soil textures (Table S4). On the other hand, given the
acidity of many forest top soils (Table S4), nitrification could be inhibited, but chemodenitrification could produce
significant amounts of NO (Pilegaard, 2013).

For a complete ecosystem net N budget, additional measurements of dissolved organic nitrogen (DON) leaching, as well as
dinitrogen (N$_2$) fluxes (biological fixation and total denitrification) would be required (Fig. 1), but they were not quantified
in most cases. A tentative ballpark estimate of the potential magnitude of denitrification N$_2$ emissions for the DB2 forest site
may be calculated by considering the mean N$_2$/N$_2$O ratio of 74 (± 0.85 st. err.), which was measured in He-O$_2$ mixture soil
incubation experiments performed on DB2 soil cores (unpublished data). This mean ratio, multiplied by the mean field

measured N$_2$O emission flux of 0.074 g (N$_2$O-N) m$^{-2}$ yr$^{-1}$ (Pilegaard et al., 2006), yields an estimate of the order of 5.5 g (N$_2$-
N) m$^{-2}$ yr$^{-1}$, or 55 kg (N$_2$-N) ha$^{-1}$ yr$^{-1}$. There is considerable uncertainty in this number, since the mean N$_2$/N$_2$O ratio was
calculated from short-term investigations in the laboratory, which may or may not be representative of the prevailing soil and
weather conditions in the field. This uncertainty is reinforced by the low sensitivity of the N$_2$ detector, which was a factor of
20-80 lower than that of the N$_2$O detector used in the experiment (Buchen et al., 2019). Another estimate of forest soil

denitrification loss obtained through a soil core incubation method was given by Butterbach-Bahl et al. (2002) for the EN2
spruce site, with an annual N$_2$ emission flux of 0.72 g (N$_2$-N) m$^{-2}$ yr$^{-1}$ and a mean N$_2$/N$_2$O ratio of 7. The N$_2$ emissions thus
estimated suggest that total denitrification may be a very significant term in the total N budget of forests, possibly of the
same order as atmospheric N$_r$ deposition.

        Measurements of DON leaching were available at very few sites, but proved to be significant. At the pine forest site of EN8,

DON leaching was of the order of 0.3 g (N) m$^{-2}$ yr$^{-1}$, i.e. a factor of three lower than DIN losses (Verstraeten et al., 2014). At
the beech forest site of DB2, DIN and DON leaching were of the same order (0.07-0.08 g (N) m$^{-2}$ yr$^{-1}$), but both very small
in comparison to N$_{dep}$ (2.15 g (N) m$^{-2}$ yr$^{-1}$); while at the pine forest site of EN10 the leaching/runoff N$_r$ loss was actually
dominated by DON (0.012 g (N) m$^{-2}$ yr$^{-1}$), which was around an order of magnitude larger than DIN leaching (Korhonen et
al., 2013) and a factor of four smaller than N$_{dep}$.

*{Insert Figure 3 here}*

**3.2 Net carbon and greenhouse gas balance**

**3.2.1 Spatial variability of the carbon sink in relation to climate and nitrogen deposition**

        The ultimate objective of the project was to quantify the response of C sequestration to atmospheric N$_r$ deposition
(addressed in Flechard et al., 2019), but this is not straightforward. We follow first in this paper a descriptive approach, in

similar fashion to previous studies (e.g. Magnani et al., 2007), whereby variations of C fluxes and other productivity
indicators (*e.g.* leaf area index and N content) are examined graphically as a function of N$_{dep}$ (Fig. 4). However, this is done





with the strong reservation that a simple empirical relationship does not necessarily prove causality, as other confounding and co-varying factors, *e.g.*, climate, soil, age, etc, may exist. Figures 4-5 show for example that the large inter-site differences in MAT and MAP at the European scale also need to be considered, beside the variability in $N_{dep}$. Note that in

assessing the variability of ecosystem carbon sink strength within the network, we use EC-derived NEP (the long term NEE sum) as a proxy for the net ecosystem carbon balance (NECB), because estimates of DIC/DOC leaching, $CH_4$ emissions and other C loss processes were not systematically measured at all sites.

Inter-annual mean NEP ranged from a small net source of -70 g (C) m$^{-2}$ yr$^{-1}$ (EN6, a waterlogged peat-based spruce stand in the southern Russian taiga) to a large net sink of +826 g (C) m$^{-2}$ yr$^{-1}$ (EN5, upland spruce forest in N. Italy) (Table 1, Fig.

4C); GPP ranged from 377 g (C) m$^{-2}$ yr$^{-1}$ (SN3, a boreal peatland site with the lowest MAT = -0.6°C) to 2256 g (C) m$^{-2}$ yr$^{-1}$ (EN14, a pine stand in Italy, one of the warmest sites with MAT of 14.9 °C and non-limiting rainfall with MAP = 920 mm) (Fig. 4A). Ecosystem respiration peaked at 1767 g (C) m$^{-2}$ yr$^{-1}$ at EN4 (upland spruce forest in E. Germany) and was lowest at 345 g (C) m$^{-2}$ yr$^{-1}$ at SN3 (boreal peatland), the coldest site (Fig. 4B); $R_{eco}$ was strongly and positively related to GPP (Fig. 4F) ($R^2 = 0.62$, slope = 0.64).

The data show a positive correlation between GPP and $N_{dep}$ in the range 0-2.5 g (N) m$^{-2}$ yr$^{-1}$ ($R^2 = 0.55$, p < 0.01). By contrast the five sites with $N_{dep}$ > 2.5 g (N) m$^{-2}$ yr$^{-1}$ tend to show visually an inverse relationship (Fig. 4A), despite the fact that they lie in comparatively favourable climates. Similar patterns are observed for $R_{eco}$ and NEP (Fig. 4B-C), but with much larger scatter and lower $R^2$ (0.24, p < 0.01, and 0.30, p < 0.01, respectively, for the $N_{dep}$ range 0-2.5 g (N) m$^{-2}$ yr$^{-1}$), with the same apparent decline for higher deposition sites. However, a closer inspection of Fig. 4A-C reveals a potential

cross-correlation with climate: i) the lower end of the $N_{dep}$ range, coinciding with the lowest GPP, $R_{eco}$ and NEP, also coincides with the lowest MAT and MAP (e.g. Finnish sites); and ii) the sites in the intermediate $N_{dep}$ range (1.5-2.5 g (N) m$^{-2}$ yr$^{-1}$), coinciding mostly with the largest observed GPP values (>1500 g (C) m$^{-2}$ yr$^{-1}$), were on average 1.8°C warmer (10.2 vs. 8.4 °C) and 89 mm yr$^{-1}$ wetter (887 vs. 798 mm) than the sites in the lower $N_{dep}$ range (0-1.5 g (N) m$^{-2}$ yr$^{-1}$).

Other proxies of the ecosystem C and N cycles and productivity, such as the LAI (defined as 1-sided for broad-leaf, or half

of total for needle-leaf; Table 1 and Fig. 4D) and the foliar N content (LeafN, Fig. 4E), also showed positive relationships to $N_{dep}$ (see below for differences between vegetation types). The inter-annual peak in leaf area index ($LAI_{max}$) increased from around 1 to 7 m$^2$ m$^{-2}$ for $N_{dep}$ increasing from 0.1 to 4.5 g (N) m$^{-2}$ yr$^{-1}$, with the lower half of the $LAI_{max}$ distribution (< 4.5 m$^2$ m$^{-2}$) mostly occurring at boreal, Mediterranean and upland sites and thus under temperature and/or water limitations.

*{Insert Figure 4 here}*

Clearly, therefore, the continental-scale variability in ecosystem/atmosphere $CO_2$ fluxes was to a large extent controlled by climate, namely by limitations in temperature and water availability. Gross ecosystem productivity was limited, as expected, by low temperatures at high latitudes (or high elevations) and by low rainfall and/or high evaporative demand at Mediterranean, boreal and continental sites. The distribution of the forest monitoring sites in the European climate space, with MAP and MAT on the x and y axes, respectively (Fig. 5A, 5B), shows that for sites with MAT > 5 °C there was a broad

negative correlation between MAT and MAP, i.e. the warmest sites in southern Europe tend to be the driest and therefore potentially water-limited. Maximum GPP (and also $R_{eco}$, not shown) occurred in the mid-climate range, around 9-15 °C MAT and around 700-1000 mm MAP. Similarly, the larger $N_{dep}$ values (> 2 g (N) m$^{-2}$ yr$^{-1}$) occurred almost exclusively at sites with MAT in the narrow range of 6-11°C, and although these large $N_{dep}$ values were found in a broad MAP range (550-1200 mm), they peaked sharply around 800-900 mm MAP (Fig. 5A). Modelled $N_{dep}$ values from the EMEP CTM (Fig. 5C,

5D) show that this is a generic pattern at the European scale.

*{Insert Figure 5 here}*

Ecosystem DIC + DOC losses estimated by Kindler et al. (2011) for 4 forest sites of this study (DB1, DB2, EN4, EN15) were on average 13 ± 7 g (C) m$^{-2}$ yr$^{-1}$ (range 3-35 g (C) m$^{-2}$ yr$^{-1}$), with contributions by DIC to total (DIC + DOC) losses varying between 18% and 83%. By contrast, Gielen et al. (2011) estimated DOC leaching losses of 10 ± 2 g (C) m$^{-2}$ yr$^{-1}$ for



the EN8 pine stand on an acidic sandy soil, in which DIC concentrations in soil water were negligibly small. Ilvesniemi et al. (2009) found DOC losses in runoff at EN10 of 0.8 g (C) m$^{-2}$ yr$^{-1}$, which was negligible compared with NEP. These leaching or runoff losses of DOC and DIC were on average over all forest sites equivalent to a very small mean fraction of 0.6% of GPP (range 0.1-1.9%), but a more significant fraction of NEP (mean 6%, range 0.3-13%). At the SN7 peatland site, fluxes of total dissolved carbon (including CH$_4$) through seepage, infiltration and drainage were relatively small by comparison to

NEP and to other peat bogs (17 g (C) m$^{-2}$ yr$^{-1}$, only 5% of NEP) (Hendriks et al., 2007); by contrast, at the SN9 peatland site, net stream C export (including DIC, DOC and POC) was on average 29.1 g (C) m$^{-2}$ yr$^{-1}$ (81% of which being DOC), equivalent to a mean leached fraction of 37% of NEP (Dinsmore et al., 2010).

### 3.2.2 Differences between plant functional types

Forests (F) and short semi-natural (SN) vegetation showed similar relationships with GPP as a function of N$_r$ deposition,

increasing with a broadly similar slope at low N$_{dep}$ values, then levelling off beyond 2g (N) m$^{-2}$ yr$^{-1}$, except for the fact that GPP was lower by typically 200-500 g (C) m$^{-2}$ yr$^{-1}$ in SN compared with F sites, for a given N$_{dep}$ level (Fig. 4). The behaviour was different for NEP, where the slope against N$_{dep}$ in the range 0-2 g (N) m$^{-2}$ yr$^{-1}$ was much steeper for F than for SN, which occurred because R$_{eco}$ values are of the same order for F and SN at a given N$_{dep}$ level. No systematic difference was observed between the forest PFT, based on the available data, in the apparent relationships of the C fluxes *vs*. N$_{dep}$.

However, this may be a result of the small number - and large diversity - of deciduous broadleaf (DB) and evergreen broadleaf (EB) forest sites in the dataset, compared with evergreen needleleaf (EN) sites (Table 1).

The relationship of LeafN to N$_{dep}$ (Fig. 4E) showed three distinct groups, with the smallest values (0.8-1.8 % N in dry weight, DW) for evergreen needleleaf and broadleaf (EN, EB) forests being positively correlated to N$_{dep}$ in the range 0.5-4.3 g (N) m$^{-2}$ yr$^{-1}$ ($R^2 = 0.71$, $p < 0.01$). Values for short semi-natural (SN) vegetation were found in an intermediate range (1-2.7

% N DW), with a steep and significant relationship to N$_{dep}$ ($R^2 = 0.51$, $p < 0.05$). The largest values occurred for deciduous broadleaf (DB) forests (mostly >2 % N DW), but with little relationship to N$_{dep}$ ($R^2 = 0.18$, not significant). Seasonal variations in forest LeafN could reach a factor of 2, as did differences between tree species within the same forest, which may account for some of the scatter observed in Fig. 4E.

### 3.2.3 Carbon fluxes and pools derived from forest ecosystem modelling

In the BASFOR base run (Fig. 6), reasonable overall model performance was achieved for GPP, ecosystem C pools, H, DBH, LAI and LeafN, while more scatter was present for R$_{eco}$, NEP and ET. In particular, in apparent contrast to GPP, R$_{eco}$ stands out as a more challenging variable to model. Predictably, because BASFOR was calibrated using a subset of 22 sites from this dataset (Cameron et al., 2018), the range and mean values of modelled R$_{eco}$ were close to mean observations by EC across the study sites, but differences between sites were poorly reproduced with much scatter around the 1/1 line and a low

$R^2$. One possible reason was that BASFOR assumed that autotrophic respiration (R$_{aut}$) is a constant fraction of GPP, which may be an over-simplification (Collalti and Prentice, 2019). Also, heterotrophic respiration (R$_{het}$) appeared to be a much more variable fraction of R$_{eco}$ in reality (Table S7) than was predicted by the model, leading to sizeable divergence in the overall modelled R$_{eco}$. As the direct measurement, NEP was the least uncertain term in EC-derived data, compared with GPP and daytime R$_{eco}$, which were inferred from measured (half-hourly) EC-NEE by empirical partitioning models. By contrast,

in BASFOR, NEP was calculated as the residual between two large numbers (GPP and R$_{eco}$) and thus compounds the uncertainties of both component terms. The modelled result for NEP appeared to be an over-estimation of net C uptake at low productivity sites and an under-estimation at high productivity ones (slope <1). A broadly similar pattern emerged for ET.

*{Insert Figure 6 here}*



### 3.2.4 Net ecosystem greenhouse gas budgets

Carbon dioxide largely dominated the net GHG budget at all forest sites, with only three sites where either $N_2O$ or $CH_4$ GWP-equivalent fluxes were larger than 10% of NEP in absolute terms (Fig. 7). Most of the forest soils (22 out of 27 sites) investigated in the bioassay experiment behaved as small net sinks for $CH_4$, with a mean (± st. err.) net oxidation flux of -0.14 ± 0.03 g (C) m$^{-2}$ yr$^{-1}$ (range -0.61 to +0.16 g (C) m$^{-2}$ yr$^{-1}$). The mean $CH_4$ flux measured by soil chambers at the 6 forest sites where such measurements were available (EN2, EN6, EN10, EN16, DB2, EB5) was also a net oxidation flux of -0.32 ± 0.15 g (C) m$^{-2}$ yr$^{-1}$ (range -1.0 to -0.0 g (C) m$^{-2}$ yr$^{-1}$). For these 6 sites, there was a significant correlation ($R^2 = 0.74$, $p < 0.05$) between annual soil $CH_4$ flux estimates derived from the bioassay experiment and from in situ flux measurements (Figure S6 in Supplement), with the largest net annual soil $CH_4$ uptake flux being observed by both methods at the EN10 pine forest site (Skiba et al., 2009). By contrast, at the elevated $N_{dep}$ sites EN2 and EN16, the net soil $CH_4$ flux was close to zero, consistent with previous research (e.g. Steudler et al., 1989; Smith et al., 2000) showing that the $CH_4$ oxidation capacity of forest soils in negatively affected by $N_r$ addition or deposition. In terms of C uptake, soil $CH_4$ oxidation was negligible compared to $CO_2$ fluxes, representing on average only 0.1% of NEP (range 0.0-0.4%). In terms of GWP the $CH_4$ flux was larger, being equivalent to 0.8% of NEP (range 0-4.5%), but on average still a factor of three smaller than the warming by $N_2O$ emissions equal to 3.9% of NEP (range 0-18.5%).

By contrast to forests, at semi-natural, short vegetation sites $N_2O$ or $CH_4$ emissions had a larger impact on the net GHG balance, where most (seven out of nine) sites showed non-$CO_2$ GHG contributions larger than 10% of NEP. Three of these seven sites were unfertilised, extensively grazed upland (SN2, SN5, SN6) grasslands (small $N_2O$ sources), while three sites (SN3, SN7, SN8) were $CH_4$-emitting peatlands or wetlands (EC-$CH_4$ and chamber flux data from Drewer et al., 2010; Hendriks et al., 2007; Juszczak and Augustin, 2013 and Kowalska et al., 2013). At SN3 and SN8, the small to moderate NEP sinks were turned by large $CH_4$ emissions into net GHG sources (net warming budgets of +127 and +242 g $CO_2$-C Eq m$^{-2}$ yr$^{-1}$, respectively), though not into actual net C sources (Fig. 7). At SN8, $CH_4$ emissions generally ranged from 25-45 g $CH_4$-C m$^{-2}$ yr$^{-1}$ but reached 86 g $CH_4$-C m$^{-2}$ yr$^{-1}$ during a particularly wet year, when the whole area was flooded. At the SN9 peatland site, Dinsmore et al. (2010) calculated that stream GHG evasion – at the scale of the 335-ha peatbog encompassing the flux tower footprint – together with downstream export represented 50-60 g $CO_2$-Eq m$^{-2}$ yr$^{-1}$ (13-16 g $CO_2$-C Eq m$^{-2}$ yr$^{-1}$), 96% of which being de-gassed $CO_2$, i.e. in the range 11-23% of the GHG budget from the tower footprint.

*{Insert Figure 7 here}*

## 4 Discussion

Previous observations of simple empirical relationships found between N deposition and forest productivity have been criticized for, amongst other things, their low number of replications, unreasonably high sensitivities of productivity to N additions, and limitations of the data and simplistic univariate statistical approaches used (Magnani et al., 2007; Högberg, 2007; de Vries et al., 2008; Sutton et al., 2008). A special feature of this study is that it aims to assemble N deposition rates and budgets together with variables of the carbon cycling for a large number of sites across the European continent in more depth and completeness than hitherto attempted, in order to seek more robust empirical evidence for the response of the terrestrial carbon cycle to different regimes of atmospheric N inputs. The quality of the individual data sets is, however, not uniformly high. Some of the data were measured in situ with known uncertainty, while others were simulated, derived from laboratory experiments and adapted to the field situation using measured time series of soil T and soil moisture, or taken from existing databases and literature. Also, data may not be fully comparable between sites (different methods used), nor even fully representative of each site (spatial heterogeneity). In the following sections, we discuss limitations of the measured, empirical and simulated data, both in terms of the component C and N fluxes, their budgets and interactions, as





well as the challenges faced when attempting to establish empirical/statistical evidence for possible N effects on carbon
sequestration in natural and semi-natural terrestrial ecosystems in Europe.

**4.1 Constraining the ecosystem nitrogen balance through combined measurements and modelling**

The compilation of $N_r$ flux data (Fig. 3), based on several independent sources for each component term, provides a realistic
picture of inorganic $N_r$ inputs and losses; their balance suggests that for forests subjected to large deposition loads (> 2 g (N)

$m^{-2}$ $yr^{-1}$), typically more than half of the incoming $N_r$ is lost to neighbouring environmental compartments such as
groundwater and the atmosphere, and thus not available to promote C storage in the forest ecosystem. Since N losses
increase - and N retention decreases - exponentially when $N_{dep}$ exceeds a critical load of approximately 2-2.5 g (N) $m^{-2}$ $yr^{-1}$,
it seems unlikely that the C sink strength of semi-natural ecosystems, including forests, increases linearly with $N_r$ deposition,
especially not with wet N deposition only. The linear relationship between C sequestration and wet $N_r$ deposition as

proposed e.g., by Magnani et al. (2007) is also challenged by the large contribution of dry $N_r$ deposition and therefore by the
poor correlation between total $N_{dep}$ and wet deposition. We argue that our multiple-constraint approach for the nitrogen
balance (measurement-model combination, model ensemble averaging, alternative data sources) provides overall a more
robust basis for studying the impact of $N_{dep}$ on the C cycle, even though uncertainties in individual terms remain significant.

**4.1.1 Reducing uncertainty in nitrogen deposition**

The uncertainty in dry deposition based on measured $N_r$ concentrations and inferential modelling is likely not smaller than
30%, due to limitations in process understanding and differences between models (Fig. 2). The uncertainty in total $N_r$
deposition is probably of the same order since even wet deposition can be deceptively difficult to measure (Dämmgen et al.,
2005), and organic N, especially wet soluble organic N (WSON), may be significant but challenging to quantify (Cape et al.,
2012) and generally ignored in the literature. WSON appears to be a generally small fraction of total (wet + dry) $N_{dep}$ at most

sites except at remote locations in Fennoscandia (EN10, SN3), where WSON deposition could represent up to 20-30% of
total $N_{dep}$. Also, potential double-counting due to dry deposition to the bulk deposition collectors (e.g. Thimonier et al.,
2018) was not considered in this study, although on the basis of the comparison to other data sources (Fig. S2), bulk
samplers did not appear to significantly over-estimate wet deposition.

Despite these uncertainties, measuring gas-phase and aerosol $N_r$ concentrations locally did provide a better estimate of total

ecosystem $N_r$ inputs, but also of the partitioning of wet *vs* dry deposition, reduced *vs* oxidized N, and canopy absorption *vs*
soil deposition, all of which are useful in interpreting ecosystem N cycling processes. In particular, for ammonia, with its
high spatial variability on a local scale, the inferential modelling approach based on local measurements is likely to provide
more realistic deposition estimates than a coarse-resolution chemical transport model (Flechard et al., 2013; Thimonier et al.,
2018). In addition to low-cost methods for $N_r$ concentrations, more actual micrometeorological $N_r$ flux measurements are

needed to further process understanding and better constrain surface exchange models over many ecosystems (Fowler et al.,
2009). For example, ammonia flux measurements at DB2 have revealed unexpected features such as net $NH_3$ emissions from
the forest in summer and autumn, in particular in response to leaf fall (Hansen et al., 2013, 2017). DB2 is likely not a net
$NH_3$ source at the annual scale, but short-term emission pulses, which are not represented in most dry deposition models
(Flechard et al., 2011), could significantly offset total $N_r$ deposition.

An improved knowledge of $N_r$ exchange patterns over $CO_2$ flux monitoring sites, either through inferential modelling or
direct flux measurements, is also essential to quantify the fraction of deposited $N_r$ that is absorbed by the canopy, reaching
more or less directly the seat of photosynthesis in leaves, thus favouring a higher nitrogen use efficiency (NUE) (Nair et al.,
2016; Wortman et al., 2012; Gaige et al., 2007). Canopy nitrogen retention occurs via several processes, including gaseous
uptake by stomatal diffusion, a well-documented process (Monteith and Unsworth, 1990), but also through cuticular

diffusion and stomatal penetration by aqueous solutions, with surface-deposited and dissolved gases and particles acting as



direct leaf nutrients (Burkhardt, 2010; Burkhardt et al., 2012). By contrast, the $N_r$ fraction initially deposited to soil (as simulated by the majority of fertilisation tracer experiments, e.g. Nadelhofer et al., 1999) is subject to various losses via nitrification, denitrification and microbial uptake, before being eventually taken up by roots and moving upwards in xylem flow. The more advanced, emerging multi-layer canopy exchange models for atmospheric pollutants ($N_r$ species, but also $O_3$,

$SO_2$, etc.) can now partition dry deposition into stomatal, non-stomatal and soil pathways with increasing detail (Zhou et al., 2017; Simpson and Tuovinen, 2014; Flechard et al., 2013), thanks to improved understanding and parameterizations of surface and air column interactions and of photosynthesis-driven stomatal conductance (Büker et al., 2007; Grote et al., 2014). However, particular attention must be paid to measurement quality for an improved deposition accuracy, because such models are still very much dependent on local atmospheric concentration data for all main $N_r$ forms (gas and aerosol,

reduced and oxidized, mineral and organic).

### 4.1.2 Uncertainty in ecosystem nitrogen losses and net balance

The comparison of DIN leaching values by different methods shows that the Dise et al. (2009) algorithm performs reasonably well for low to moderate $N_r$ deposition levels, but underestimates DIN losses for some of the highest (>4 g (N) m$^{-2}$ yr$^{-1}$) deposition sites. This observation was also made by Dise et al. (2009) themselves, who argued that their simple

relationships involving external forcings ($N_{dep}$) and internal factors (soil N status) are adequate "for early to intermediate stages of nitrogen saturation", but may fail at sites where historical, chronically enhanced $N_r$ deposition has so strongly impacted forest ecosystems that N leaching has become dependent also on stochastic factors such as e.g. insect defoliation or a drought period followed by re-wetting of the soil. As was the case for field measured NO emissions (Fig. 3A), the four highest DIN leaching fluxes (0.9-3.2 g (N) m$^{-2}$ yr$^{-1}$) occurred in the four highest $N_{dep}$ forests growing on well-drained acidic

sandy soils. In addition, it is noteworthy that the two sites with the largest $N_{dep}$ and DIN leaching rates (EN15, EN16) were dominated by pine or Douglas fir (Table S1), which have been shown in a common garden experiment (Legout et al., 2016) to cause larger nitrification, $NO_3^-$ leaching and acidification rates (as well as larger losses of calcium, magnesium and aluminium), compared with other tree species such as beech or oak. This is consistent with deciduous trees being known to take up and store more nitrogen per unit biomass in stems and branches than coniferous trees (Jacobsen et al., 2003). Typical

stem N content values, proposed for N uptake calculations in the Convention on Long-range Transboundary Air Pollution (CLRTAP) manual for critical loads mapping, are 1 and 1.5 g N kg$^{-1}$ dry matter for conifers and deciduous trees, respectively, for steady state conditions (CLRTAP, 2017). Tree species traits may therefore, in our study, have exacerbated an existing DIN leaching predisposition resulting from edaphic factors and pollution climate. At the lower end of the $N_{dep}$ range, the dataset is consistent with previous studies, which have shown that DIN leaching is unlikely to occur in forests

where $N_{dep}$ < 1 g (N) m$^{-2}$ yr$^{-1}$ (de Vries et al., 2009), although under these conditions there may still be significant N losses as NO and $N_2O$ (Fig. 3).

The best empirical fit for the relationship of the sum DIN + NO + $N_2O$ to $N_{dep}$ was slightly non-linear (Fig. 3D) and may indicate that at the upper end of the $N_{dep}$ range, above 4 g (N) m$^{-2}$ yr$^{-1}$, the sum of inorganic $N_r$ losses might approach or even exceed the estimated atmospheric deposition, which corresponds to one of the several existing definitions of ecosystem N

saturation (see below). Whether these ecosystems turn into net N sources depends on the relative magnitudes of the missing terms: $N_2$ fixation (likely small in temperate compared with tropical forests; Vitousek et al., 2002), $N_2$ losses from denitrification (possibly the largest of the unknown terms at forest sites that are frequently waterlogged), $N_2O$ losses from the litter layers of the forest floor, DON leaching; and also incoming organic nitrogen in precipitation (WSON) as well as dry deposition of organic $N_r$ species, not quantified here (Fig. 1). The presumably small, and unaccounted for, N inputs via

$N_2$ fixation and organic $N_r$ deposition are at least partly compensated by denitrification $N_2$ losses and DON leaching losses. Moreover DON leaching typically responds much less strongly than DIN leaching to N inputs (Siemens and Kaupenjohann, 2002). Under these assumptions, the inorganic $N_r$ budget calculated from Fig. 3 may provide a reasonable proxy for the



overall ecosystem N balance. In this case, N outputs by gaseous and dissolved losses represent on average across all forest sites 43% of N inputs. More important than the average N loss for judging $N_r$ deposition effects on C sequestration, is the

large range of losses from 6% to 85%, with typically 10-20% loss for $N_{dep} < 1$ g (N) m$^{-2}$ yr$^{-1}$, 30-50% loss for intermediate $N_{dep}$ levels, and 50-80% loss for $N_{dep} > 3$ g (N) m$^{-2}$ yr$^{-1}$. However, if the very few available data or estimates for DON leaching and especially denitrification $N_2$ fluxes are correct and may be extrapolated to other sites, they may often outweigh the inputs through organic $N_r$ deposition and biological $N_2$ fixation, and thus the inorganic $N_r$ budget (Fig. 3) may under-estimate the overall N losses.

**4.2 Drivers and uncertainties of the carbon and GHG balance**

**4.2.1 Carbon sequestration efficiency**

The fraction of accumulated carbon in the ecosystem relative to the gross $CO_2$ uptake by photosynthesis is a useful metric to compare carbon cycling in different terrestrial ecosystems and it is directly related to climate effects and other drivers such as site fertility (Vicca et al., 2012) and management (Campioli et al., 2015). By analogy with the carbon use efficiency

(CUE, commonly defined from a plant's perspective as the NPP/GPP ratio), of which the biomass production efficiency (BPE = BP/GPP; Vicca et al., 2012) is a proxy, we thus define here an ecosystem-scale, medium-term indicator of carbon sequestration efficiency (CSE) as the NEP/GPP ratio, calculated from measurable fluxes over the CEIP/NEU project observation periods. Quantifying the accumulated carbon in terrestrial ecosystems requires, however, much longer observations (one or several decades), to ensure statistical significance of a small change over a large C stock, particularly

when soils are considered. This is often impractical, but also of limited use, because N deposition rates are unlikely to be constant over such long periods.

Observed CSE values (CSE$_{obs}$) varied widely among observation sites, ranging from -9 to 61% (Fig. 8), much more so than the values derived from BASFOR model simulations for a contemporaneous 5-yr period (CSE$_{mod}$, 17-31%) (comparison made for the 22 sites that were included in the model calibration by Cameron et al., 2018). Negative CSE values (EN6,

EN11) imply a net carbon source and may be explained by a number of factors, including soil carbon loss, lateral DOC/DIC water flow from adjacent ecosystems, tree mortality, low fertility, poor ecosystem health, a recently planted forest or other disturbances with long-lasting consequences on the C budget. For EN6 the main reasons are a large SOC concentration, leading to large $R_{eco}$ values, and a relatively old age of the forest, responsible for a small GPP. The large discrepancy between observation-based and modelled CSE estimates may not be entirely caused by the model's inability to reproduce all

fine patterns of GPP and especially $R_{eco}$ across all ecosystems (Fig. 6). Some of the largest CSE$_{obs}$ values may be less ecologically plausible and might result from methodological biases and/or incorrect interpretation of the EC measurements, in terms of their representativeness for the ecosystem considered.

Multi-annual values of GPP and $R_{eco}$ derived from EC flux data are not measurements *sensu stricto*; they compound problems in EC measurements, post-processing of high frequency data, partitioning and gap-filling. Some partitioning

algorithms (Barr et al., 2004; Reichstein et al., 2005) evaluate GPP as the difference between measured daytime NEE and an estimate of daytime $R_{eco}$ that is based on an empirical model of night-time $R_{eco}$ measurements. In this case, any problem with nighttime and thus with estimated daytime $R_{eco}$ would directly impact GPP in the same way (Vickers et al., 2009): GPP and $R_{eco}$ would both be under-estimated, or both over-estimated, in absolute terms and by the same absolute magnitude, thereby impacting the annual or long term NEP/GPP (CSE$_{obs}$) ratio.

In this study, however, the use of the daytime data based partitioning method by Lasslop et al. (2010), within the REddyProc algorithm embedded in the European Fluxes Database Cluster, was intended to ensure the independence of GPP and $R_{eco}$ estimates, since $R_{eco}$ was estimated from the intercept of the Michaelis-Menten light–response curve fitted to daytime measured NEE. This partitioning procedure should avoid the propagation into the GPP estimate of potential errors in nighttime $R_{eco}$ data, although it still assumes similar dependencies of day- and nighttime respiration to environmental factors,



which is debatable from a biological standpoint (e.g., Kok, 1949; Wehr et al., 2016; Wohlfahrt and Galvagno, 2017). From a micrometeorological perspective, the nighttime flux can be underestimated due to low turbulence conditions and the transport of $CO_2$ by horizontal and/or vertical advection, and the decoupling of soil-level and understorey fluxes from the turbulent fluxes measured above the canopy (Feigenwinter et al., 2008; Etzold et al., 2010; Montagnani et al., 2010; Paul-Limoges et al., 2017). Further, in principle, the $u_*$ threshold filtering (Gu et al., 2005; Papale et al., 2006), carried out to

discard low turbulence flux data at the start of the gap-filling and partitioning algorithm (REddyProc, 2019), should alleviate the issue of nighttime $R_{eco}$ underestimation, which affects annual $R_{eco}$ and $CSE_{obs}$ even if the error does not propagate into GPP in the Lasslop et al. (2010) method. However, the choice of the value for the $u_*$ threshold can be critical if advection-affected flux values are to be discarded, especially for sites and data sets where the independence of the gap-filled annual NEP value from the $u_*$ threshold value cannot be demonstrated. Advective flux contributions remain a largely unresolved

issue, as Aubinet et al. (2010) conclude that «direct advection measurements do not help to solve the night-time $CO_2$ closure problem». Others (e.g. Kutsch and Kolari, 2015) have commented on the need to assign appropriate uncertainties when dealing with CSE and C balances derived from EC flux towers, which only measure turbulent fluxes and $CO_2$ storage change in the air column underneath the sensor but not the other terms of the conservation equation of a scalar in the atmospheric boundary layer (see Eq. (1) in Aubinet et al., 2000).

Despite all these precautions, at sloping or complex terrain sites where advection can be important, it cannot be excluded that the Lasslop et al. (2010) daytime data based approach may still underestimate $R_{eco}$ (and overestimate $CSE_{obs}$) if advection is not accounted for explicitly. This is because the $R_{eco}$ estimate based on the the intercept of the light response curve for the measured NEE (at PAR = 0) is strongly influenced by measurements made around sunrise and sunset, when a clear impact of advection on the light response curve ordinate has been observed, as shown at the EN5 subalpine site by Montagnani et al.

(2009) (see their Fig. 13).

It is important to note that advection may also be a problem at flat lowland sites if there is strong spatial land surface heterogeneity, e.g. differences in albedo or in Bowen ratio, a gradient in tree species, a nearby lake, a gradient in water availability. Conversely, there may also be sites where EC underestimates $CSE_{obs}$ for similar reasons, albeit in the opposite direction, for example additional $CO_2$ being advected into the ecosystem, then released by turbulent diffusion to the

atmosphere within the tower footprint. Another possibility is that basal $R_{eco}$, measured at dawn or dusk over a different (larger) footprint, is lower than during the day. Flux partitioning may again in this case underestimate $R_{eco}$ during the warmer daytime hours, and therefore also underestimate GPP, resulting in overestimated NEP/GPP ($CSE_{obs}$) ratios.

Given this uncertainty, the fact that most of the forest stands with $CSE_{obs}$ values larger than 40% (EN1, EN5, DB6, MF2) were located at elevations above 700 m a.m.s.l. (Table 1 and Fig. 8A), i.e. in hilly or mountainous areas with topographically

more complex terrain than typically encountered at lowland sites, may be coincidental, or partly a consequence of advection or decoupling issues (Paul-Limoges et al., 2017). In such conditions, consistency crosschecks involving additional flux, advection, soil and biometric measurements, even ecosystem modelling, provide useful reference points to assess the plausibility of EC-derived C budgets and to better constrain the problem. At the EN5 site, the annual total tree biomass C increment based on biometric measurements was on average 218 g (C) m$^{-2}$ yr$^{-1}$ over the period 2010-2017 (L. Montagnani,

unpublished data), i.e. 26% of the reported mean EC-derived NEP value of 826 g (C) m$^{-2}$ yr$^{-1}$ for the CEIP-NEU period, and it seems unlikely that the increase in soil carbon and fine roots stocks could account for the large difference. By contrast, the DB6 site was a fertile and managed beech forest, with a significantly higher efficiency conversion of photosynthates into biomass compared to less fertile and unmanaged sites (Vicca et al., 2012; Campioli et al., 2015). The long term annual total NPP at the site was 780 g (C) m$^{-2}$ yr$^{-1}$ over the period 1992-2007, with a significant part allocated belowground (Alberti et

al., 2015), while heterotrophic respiration estimated at the site using either bomb-carbon (Harrison et al., 2000) or mineralization rates (Persson et al., 2000) was around 200 g (C) m$^{-2}$ yr$^{-1}$, resulting in similar NEP estimates by EC flux measurements versus biometric data combined with process studies.



At the MF2 site, Etzold et al. (2011) calculated inter-annual mean EC-derived NEP, GPP and $R_{eco}$ values (for the same 2005-2009 period used in this study) of 415, 1830 and 1383 g (C) m$^{-2}$ yr$^{-1}$, respectively, using customized gap-filling and partitioning algorithms and thus providing alternative estimates to those from the REddyProc algorithm within the European Fluxes Database Cluster (Table 1). Values of $R_{eco}$ and NEP were 82% larger and 40% lower, respectively, in Etzold et al. (2011) compared with the default database values that do not explicitly correct for advection. However, the Etzold et al. (2011) mean EC-derived NEP was much closer to NEP values calculated from the net annual increment in the woody and non-woody biomass and soil C storage using four different biometric and modelling methods (range 307-514 g (C) m$^{-2}$ yr$^{-1}$, mean 421 g (C) m$^{-2}$ yr$^{-1}$). The $CSE_{obs}$ value derived from Etzold et al. (2011) was 23%, and comparable to the value of 25% that can be calculated from the decoupling-corrected EC budget computed by Paul-Limoges et al. (2017) for the same site for the years 2014-2015, in which the decoupling correction to account for undetected below-canopy fluxes doubled $R_{eco}$ and reduced NEP from 758 to 327 g (C) m$^{-2}$ yr$^{-1}$. These alternative $CSE_{obs}$ estimates were thus much lower than the default $CSE_{obs}$ value of 48% (Fig. 8) but fully consistent with model predictions (Fig. 8A).

The four upland sites EN1, EN5, DB6, MF2, were also among the wettest, with MAP > 1000 mm (Fig. 8B) in principle promoting larger leaching and runoff. The overall distribution of $CSE_{obs}$ as a function of MAP (Fig. 8B) shows an apparent increase of $CSE_{obs}$ with precipitation, though with large scatter, which would be consistent with a reduction in EC tower-based $R_{eco}$ through an increase in the dissolved leached fraction. At sites where significant leaching occurs, $R_{eco}$ determined from the atmospheric flux is no longer a reliable indicator of total C losses by respiration since the dissolved, then leached fraction of $R_{soil}$ is not captured by the flux tower (Gielen et al., 2011), which implies that $CSE_{obs}$ is over-estimated. As observed in the case of GPP, such apparent correlations of $CSE_{obs}$ to single factors like elevation or MAP may not be (entirely) causal, potentially concealing underlying cross-correlations (such as large but unmeasured advection components occurring at the same sites where MAP is largest). The data by Kindler et al. (2011) and Gielen et al (2011) do suggest that the overestimation of C sequestration (as estimated by EC-derived NEP), caused by not accounting for dissolved C leaching, was likely smaller than 10% for forests (7% of NEP on average), but all five sites they investigated had MAP < 1000 mm and only one (EN4) was an upland site (785 m).

To summarize a set of unresolved issues, the largest $CSE_{obs}$ values (> 45%) are likely to result from a combination of ecological factors and methodological biases, but they occurred at sites in mid range for $N_{dep}$ (1.2–2.2 g (N) m$^{-2}$ yr$^{-1}$) and thus did not introduce confounding trends in the overall C/N relationships we seek to establish across the whole $N_{dep}$ spectrum in this study.

*{Insert Figure 8 here}*

### 4.2.2 Forest net greenhouse gas balance dominated by carbon

Based on the available data, the net GHG balance of the 31 forests investigated was generally not significantly affected by $N_2O$ or $CH_4$ (Fig. 7), with the caveat that these fluxes were not actually measured in situ everywhere, nor with the same intensity and duration as $CO_2$. Thus, the uncertainty in non-$CO_2$ GHG fluxes is much larger (possibly > 100%) than for multi-annual EC-based $CO_2$ datasets, where a typical uncertainty is of the order of 10-30% (Loescher et al., 2006). Nonetheless, the $N_2O$ and $CH_4$ emissions observed by different methods in forest soils were typically two orders of magnitude smaller than the $CO_2$ sink (in GWP equivalents), which means that the quality of $CO_2$ estimates dominates the overall uncertainty in our forest GHG budgets. Note that such results cannot be extended to waterlogged, organic soils of temperate and boreal zones, where $CH_4$ emissions can be large (Morison et al., 2012), nor to the tropics especially in degraded forests (Pearson et al., 2017). Also, $N_2O$ fluxes can be highly episodic, with emission events linked to, e.g., freeze-thaw cycles (Risk et al., 2013; Medinets et al., 2017) and such episodes would have been missed by the bioassay approach.

By contrast, for the short semi-natural vegetation sites of our study, NEP was on average a factor of 2.7 smaller than in forests, but only a factor of 1.5 smaller for GPP, which implies that total C losses were much larger in proportion to gross





assimilation, especially non-respiratory, non-$CO_2$ losses (i.e., a much lower CSE). Large wetland $CH_4$ emissions and dissolved DIC/DOC fluxes were much more likely to offset or even determine the C and GHG balance (Fig. 7; Kindler et al., 2011). In these systems, studying the impact of $N_r$ deposition on C sequestration requires much more robust estimates of the gaseous and dissolved budgets for all components and over the long term, since the estimation of NECB requires in addition to EC-$CO_2$ the knowledge of non-atmospheric, non-$CO_2$ fluxes (Fig. 1). Technological developments in the field of (routine)

EC measurements for $N_2O$ and $CH_4$ (e.g. Nemitz et al., 2018) are likely to reduce uncertainties in net GHG budgets in the foreseeable future, but DIC/DOC losses in wetlands probably represent a bigger challenge.

     It should however be remembered that such short-term GHG budgets, based on a few years flux data and GWP multipliers for a 100-yr time horizon, do not actually reflect the long term climate impact of northern mires, which may be thousands of years old, and despite their $CH_4$ emissions, typically have an overall climate-cooling effect. As shown by Frolking et al.

(2006), pristine mires typically start cooling the climate some hundreds of years after their formation, the exact timing of course depending on the magnitude of the $CH_4$ and $CO_2$ fluxes; thus the history of the site should be accounted for when dealing with ecosystem radiative forcing assessments. For the SN3 site, Drewer et al. (2010) actually used a 500-yr time horizon GWP (instead of the usual 100-yr) for $CH_4$, reducing the GHG source strength of the site by a factor of 4 to 10, depending on the year considered.

## 4.3 Challenges in understanding the coupling of carbon and nitrogen budgets

### 4.3.1 Tangled effects of nitrogen deposition and climate on ecosystem productivity

     The analysis of $N_{dep}$ variability and spatial patterns at the scale of the monitoring network, as well as the European scale (Fig. 5), showed that the impact of $N_r$ deposition on ecosystem C sequestration cannot be considered independently of climate in the regional context of this study. Through the continent-wide geographical distribution of population, human,

industrial and agricultural activities, and of precursor emissions, combined with mesoscale patterns of meteorology-driven atmospheric circulation and chemistry, the elevated $N_{dep}$ levels in this study happened to co-occur geographically with temperate climatic zones of Central-Western Europe (Fig. 5 C-D) that are the most conducive to vegetation growth at the continental scale. This means adequate water supply as precipitation, reasonably low summertime evaporative demand, mild winters and temperate summers, long growing seasons. In other words, there are many gaps in the multi-dimensional

variable space, which is incompletely explored by the available dataset. Thus, any regression analysis that would correlate NEP and other C fluxes with $N_{dep}$, without simultaneously accounting for climate, would be flawed, as Sutton et al. (2008) concluded from their re-analysis of the data used by Magnani et al. (2007). A dC/dN slope calculated directly from a (linear or non-linear) mono-factorial regression analysis of GPP or NEP vs. $N_{dep}$ would misleadingly attribute the whole C flux variability to $N_{dep}$ while ignoring climate effects (Fleischer et al., 2013). In addition, a range of other potential explanatory

variables such as soil type, especially the water holding capacity ($\Phi_{FC}$ - $\Phi_{WP}$), soil fertility (Vicca et al., 2012; Legout et al., 2014), tree species, stand age (Besnard et al., 2018), are potentially needed to explain the observed variability (Flechard et al., 2019). Our attempts with more advanced forms of regression analyses (e.g. multiple, stepwise, mixed non-linear models, residual) did not prove successful at untangling the multiple inter-relationships, even when normalising GPP to climate proxies such as the length of the growing season, growing degree days, etc, due to the limited size and very large diversity of

the dataset. This shows that a simple pattern to explain the coupling of carbon and nitrogen budgets with the available data and knowledge is unlikely.

### 4.3.2 Evidence of nitrogen saturation from various indicators

     Various definitions of nitrogen saturation have been proposed (Aber, 1992; De Schrijver et al., 2008; Binkley and Högberg, 2016), including i) the absence of a growth response in the case of further N addition (dC/dN = 0); ii) the onset of $NO_3^-$

leaching and/or gaseous emissions; and iii) the equivalence of N inputs and N losses. The underlying concept of a dC/dN





response is that the C and N cycles are closely coupled through stoichiometric ratios in the different parts of the ecosystem, with very different C/N ratios in soil organic matter, roots, leaves, tree branches and stems (de Vries et al., 2009; Zechmeister-Boltenstern et al., 2015). A difference in dC/dN response could, for example, be expected between forests, where carbon is stored in both woody and root biomass (C/N ratio 300-500) and below ground in SOM (C/N ratio 30-40),

versus short semi-natural vegetation, where most of the stock is in SOM, and thus with a much lower overall ecosystem C/N ratio. This would be consistent with the observations in Fig. 4, where the apparent increase of NEP with increasing $N_{dep}$ is smaller in short semi-natural vegetation than in forests. But the theoretical stoichiometric approach becomes more uncertain in the event of N saturation, as the C and N cycles have become much less tightly coupled than in pristine, N-limited environments, and thus defining a dose-response relationship requires a precise quantification of all C and N inputs and

losses, not just productivity and $N_r$ deposition.

Another possible indicator of N saturation in the present dataset may be provided by the comparison of the relationships of C/N ratios of foliage and top soil (5 cm) to atmospheric $N_r$ deposition (Fig. 9A-B). Since leaf N content was not only dependent on $N_r$ deposition but also on the ecosystem type (Fig. 4E), C/N ratios are shown separately for the different vegetation classes in Fig. 9. There was a clear negative correlation of leaf C/N ratio to $N_{dep}$ for coniferous forests (ENF,

spruce and pine pooled: exponential fit $R^2 = 0.86$, $p < 0.01$) and a similar but not significant trend for SN (linear $R^2 = 0.29$) (Fig. 9A); for the other ecosystems (DBF, MF, EBF) there were not enough data to derive trends. In top soils (Fig. 9B), there was also a broad downward trend of C/N ratios with increasing $N_{dep}$ within the ENF and SN classes, but only for $N_{dep}$ up to 2.5 g (N) m$^{-2}$ yr$^{-1}$. Again, as for GPP and NEP, the relationship is highly non-linear as the four ENF sites above this $N_{dep}$ threshold break the trend observed in the lower $N_{dep}$ sites, and the overall best fit is quadratic ($R^2 = 0.49$, $p < 0.01$) with an

inflexion point around this threshold. While the relationship of foliar C/N ratio to $N_{dep}$ was almost linear for ENF (a consequence of the linear trend in ENF leaf N content, Fig. 4E), the non-linear behaviour of the topsoil C/N ratio and its stabilization or increase for $N_{dep} > 2.5$ g (N) m$^{-2}$ yr$^{-1}$ indicate a possible threshold for saturation. Atmospheric nitrogen was therefore apparently efficiently taken up by vegetation when reaching the leaves; but after leaf fall, and following litter decomposition and incorporation into the topsoil, there appeared to be a limit to the amount of nitrogen that can be stabilized

into soil organic matter of the ENF sites. However, forest soil organic N stocks are very large (in the range 200-700 g (N) m$^{-2}$ at the sites we investigated), and therefore changes in C/N ratios in response to atmospheric $N_r$ deposition must be very slow. The soil C/N ratio at a given time reflects centuries of land use as well as a more recent history of multi-decadal changes in $N_r$ deposition (Fig. S4). This complicates the interpretation of the downward trends observed from instantaneous snapshots of soil and foliar C/N ratios versus $N_{dep}$ since the ecosystems cannot be considered to be in steady state, neither for

$N_{dep}$ nor for growth or productivity. There was a positive correlation across all vegetation types between topsoil and foliar C/N ratios (Fig. 9C; $R^2 = 0.19$, $p < 0.05$), but this was mostly driven by differences between plant functional types (no significant correlation within each PFT).

*{Insert Figure 9 here}*

Following definition ii) of N saturation given above, the sum of inorganic $N_r$ losses, heavily dominated by DIN leaching at

the upper end of the $N_{dep}$ range in our datasets (Fig. 3), may indicate various stages of N saturation in all forests with $N_{dep} > 1$-1.5 g (N) m$^{-2}$ yr$^{-1}$. However, given the large uncertainties in the N budgets, a more confident threshold for an advanced saturation stage could be placed at 2-2.5 g (N) m$^{-2}$ yr$^{-1}$. Such numbers are entirely consistent with the leaching risk classification of European forests proposed by Dise and Wright (1995), with low leaching risk at $N_{dep} < 1$ g (N) m$^{-2}$ yr$^{-1}$, intermediate risk at $N_{dep}$ in the range 1-2.5 g (N) m$^{-2}$ yr$^{-1}$, and high risk at $N_{dep} > 2.5$ g (N) m$^{-2}$ yr$^{-1}$. The results are also in line

with the review by De Vries et al. (2014); based on literature results of dC/dN responses derived from stoichiometric scaling, meta-analysis of N addition experiments and field observations of both growth changes and $N_r$ deposition, accounting for other drivers, the data showed beneficial $N_r$ deposition effects up to 2-3 g (N) m$^{-2}$ yr$^{-1}$ and adverse effects at higher levels. A lower $N_{dep}$ threshold of 1 g (N) m$^{-2}$ yr$^{-1}$ had also been suggested by de Vries et al. (2007), but this was using throughfall



deposition, which generally under-estimates total deposition through canopy retention processes (Thimonier et al., 2018). It
must be stressed, however, that the definition of an all-purpose, generic $N_{dep}$ threshold for N saturation may be misleading,
or at least qualified with an uncertainty, since some tree species (Douglas fir, pine, spruce), grown on the same soil and
under the same climate and $N_{dep}$ regime, may result in significantly higher $NO_3^-$ leaching rates than others (Legout et al.,
2016). This also means that the $NO_3^-$ leaching flux is not necessarily a good proxy of the severity of N saturation, though this
depends on which of the several definitions of N saturation is considered.

The upper threshold of 2-2.5 g (N) $m^{-2}$ $yr^{-1}$ happens to coincide with the levelling off of GPP, $R_{eco}$ and NEP, and the further
reduction in C fluxes at higher $N_{dep}$ levels (Fig. 4A-C). Whether this should be interpreted as a negative impact of advanced
N saturation on soil processes and plant functioning and, hence, C sequestration potential, is not straightforward (Binkley
and Högberg, 2016). If the parallel effects of climate, soil fertility, other nutrient limitations, tree species traits, age and
planting density are overlooked in a simplistic, first-order interpretation, the dataset hints at an "optimum" $N_{dep}$ level around
2 g (N) $m^{-2}$ $yr^{-1}$, beyond which no further benefits (in carbon terms) could be gained from further atmospheric $N_r$ additions,
which would be consistent with the 2-2.5 g (N) $m^{-2}$ $yr^{-1}$ $N_{dep}$ threshold derived by Etzold et al. (2014) for Swiss forests. The
high soil $N_r$ losses observed in these ecosystems growing under relatively favourable climates would then suggest that
whatever fertilisation effect $N_r$ deposition may have at low to moderate deposition rates (<2 g (N) $m^{-2}$ $yr^{-1}$) is unlikely to be
sustained at high deposition levels, especially on acidic sandy soils. However, the very limited number of affected sites with
$N_{dep}$ > 3 g (N) $m^{-2}$ $yr^{-1}$ leaves too few degrees of freedom to make the argument statistically compelling. More importantly, a
knowledge of all other limitations to growth (climate, soil, fertility, nutrients, age structure) would be required to confirm the
hypothesis. Additional measurement- and model-based investigations to untangle the $N_{dep}$ effect on C sequestration (the
dC/dN term) are presented in Flechard et al. (2019), drawing from the results, fluxes and budgets presented here.

**5 Conclusion**

We provided estimates of carbon, nitrogen and greenhouse gas budgets for 40 flux tower sites over European forests and
semi-natural vegetation, compiled from a large variability of state of the art methods that can be applied in such a network
approach. The $CO_2$ budgets from well-established EC methods were the least uncertain, followed by GHG budgets of
forests, then the $CH_4$ and DIC/DOC fluxes of wetlands; uncertainty levels were likely highest in the net N budgets,
especially at the elevated $N_r$ deposition sites where $NO_3^-$ leaching was almost of the same order as $N_{dep}$. The uncertainty was
still compounded by the lack of some data on biological $N_2$ fixation, $N_2$ loss by denitrification, and organic $N_r$ in rainwater,
in dry deposition and in soil leaching, but some of these unknown terms would compensate mutually to some extent.
Nevertheless, the low-cost network to monitor atmospheric gas-phase and aerosol $N_r$ contributed to substantially reducing
the large uncertainty in total $N_{dep}$ rates at individual sites (compared with gridded outputs of a regional chemical transport
model), because dry deposition almost systematically heavily dominates over wet deposition in forests, except at very remote
sites (away from sources of atmospheric pollution), and the uncertainty in dry deposition and its modelling is much larger.

The greenhouse gas balances of the 31 forest sites included in this study were almost entirely determined by the $CO_2$
budgets, with small to negligible contributions by $N_2O$ and $CH_4$. The GHG balance of nine extensively managed and upland
grasslands, moorlands and wetlands was much more dependent on $CH_4$ and $N_2O$ fluxes. Ecosystem productivity (GPP, NEP)
data across Europe showed an apparent increase with atmospheric $N_{dep}$, though only up to 2.5 g (N) $m^{-2}$ $yr^{-1}$, while the larger
$N_{dep}$ rates also happen to coincide geographically with regions of Europe where climate is optimal for tree growth (neither
too cold nor too dry). The data thus underpinned a strong covariation of $N_r$ deposition with variables like elevation and
climate, and indicated that the ecosystem response of carbon sequestration to nitrogen deposition cannot be calculated
simply and directly from the observed apparent dNEP/d$N_{dep}$ using bivariate statistics. Other co-varying influences such as



climate, soil, fertility, nutrient availability, forest age, ecophysiological processes, etc., should be analyzed alongside, so the nitrogen deposition effect can be isolated.

The site-specific analysis of C and N fluxes and budgets across a large geographical and climatic gradient supports the concept of a non-linear response of C sequestration to N deposition. Large nitrogen losses (especially nitrate) from forests suggest that up to one third of the sites investigated can be classified as in early to advanced stages of N saturation. At the sites with the largest $N_r$ deposition rates (> 2.5 g (N) m$^{-2}$ yr$^{-1}$), a stagnation or reduction in forest productivity, compared to

mid-range deposition sites, was observed. Beyond the conclusion that the apparent C response to increased $N_r$ deposition was non-linear, we do not have enough data to test the hypothesis that the reduction in productivity and C sequestration is linked to N saturation-induced ecological impacts on soil and ecosystem functioning, rather than just the confounding effects of variability in meteorological and other drivers. Further efforts are required to disentangle $N_{dep}$ effects and climatic as well as pedological effects on C sequestration at the continental scale.

**Code and data availability**


The data used in this study are publicly available from online databases and from the literature as described in the Materials and Methods section.

The codes of models and other software used in this study are publicly available online as described in the Materials and Methods section.

**Author Contributions**


CRF, MAS, AI, WdV, MvO, UMS conceived the paper; MAS, EN, UMS, KBB, WdV conceived or designed the NEU study; CRF performed the data analyses, ran model simulations and wrote the text; MvO, DRC wrote and provided the BASFOR model code and performed Bayesian calibration; YST, NvD, HU, UD, SV, VD, MMit, FS, YF performed DELTA and bulk deposition chemical analyses; AF collected wet deposition databases; DS provided modelled EMEP $N_r$ deposition

data; BKit, SZB conceived and performed the soil bioassay experiment; JKS provided foliar nitrogen analyses; AI, UMS, JFJK, ALeg, MJS, MAub, DL, LM, JN, IAJ, MP, RK, JA, AV, JO, RJ, MAur, BHC, JD, WE, AF, AG, PG, YH, CH, AH, LH, BKru, WLK, RLdV, ALoh, BL, MVM, GM, VM, JMO, KP, GP, MTS, MU, TV, CV, TW provided eddy covariance and/or other field data, or contributed to data collection from external databases and literature; AI, WdV, MAS, UMS, MvO, EN, KBB, SZB, DRC, NBD, JFJK, NB, ALeg, DS, MJS, MAub, DL, LM, JN, IAJ, MP, RK, JS, AJF, JA, AV, JO, RJ,

MAur, WE, BKit, BKru, RLdV, ALoh, GM, AN, MU contributed substantially to discussions and revisions.

**Competing Interests**

The authors declare that they have no conflict of interest

**Acknowledgements**

The authors gratefully acknowledge financial support by the European Commission through the two FP-6 integrated projects
CarboEurope-IP (Project No. GOCE-CT-2003-505572) and NitroEurope-IP (Project No. 017841), the FP-7 ECLAIRE project (Grant Agreement No. 282910), and the ABBA COST Action ES0804. We are also thankful for funding from the French GIP-ECOFOR consortium under the F-ORE-T forest observation and experimentation network, and from the MDM-2017-0714 Spanish grant. We are grateful to Christian Bernhofer, Robert Clement, Han Dolman, Axel Don, Eric Dufrêne, Damiano Gianelle, Ruediger Grote, Anders Lindroth, John Moncrieff, Dario Papale, Corinna Rebmann, Alex Vermeulen, for



the data they provided, and to Klaudia Ziemblińska for her comments on the manuscript. Computer time for EMEP model runs was supported by the Research Council of Norway through the NOTUR project EMEP (NN2890K). Finalisation of the manuscript was supported by the UK Natural Environment Research Council award number NE/R016429/1 as part of the UK-SCAPE programme delivering National Capability.

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





**Figures and tables**

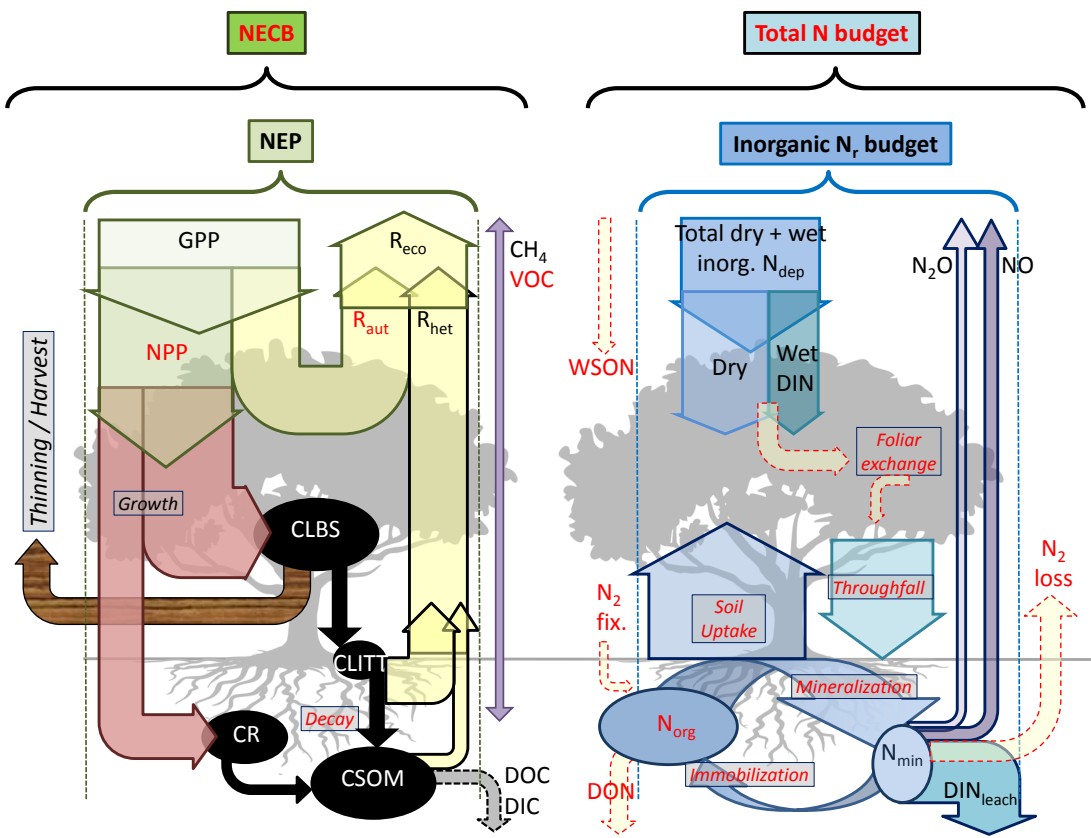


**Figure 1. Flux terms and boundaries of the carbon (left) and nitrogen (right) budgets discussed in this paper. Net ecosystem productivity NEP = GPP − $R_{eco}$ (≈ NPP − $R_{het}$) based on multi-annual eddy covariance $CO_2$ flux data. The net ecosystem carbon balance (NECB) includes in addition other C loss fluxes such as DIC/DOC, $CH_4$ and VOC, as well as harvest, thinning or other disturbances (e.g. fire). Inorganic reactive nitrogen ($N_r$) budget = $N_{dep}$ − $DIN_{leach}$ − NO − $N_2O$. The total N budget includes in addition organic nitrogen deposition (WSON) and leaching (DON), as well as $N_2$ inputs and losses from biological fixation and denitrification, respectively. CLBS, CSOM, CR, CLITT: carbon stocks in leaves, branches and stems, in soil organic matter, in roots, and in litter layers, respectively. Terms highlighted in red indicate that direct or measurement-based estimates were not available for some or all sites in our datasets (see also Table 2 for a list of acronyms; see Table S6 for data availability).**




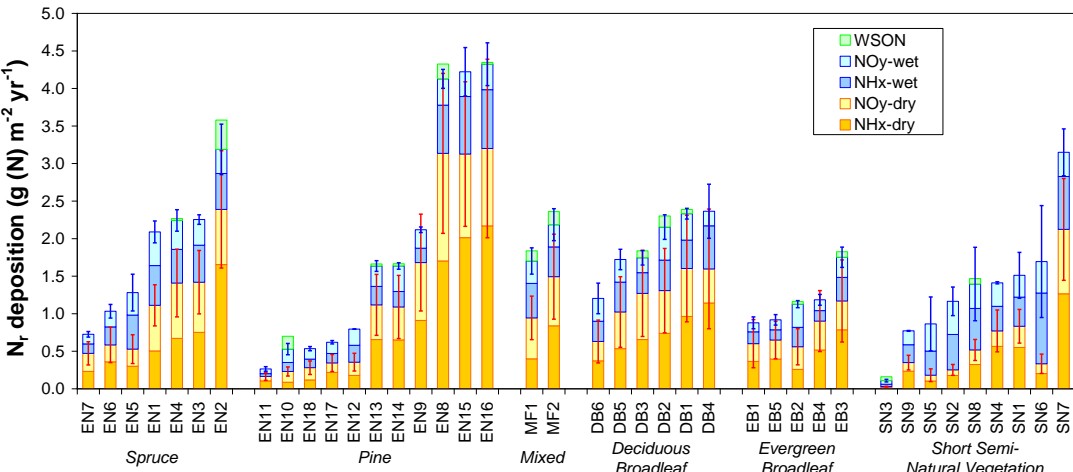


**Figure 2. Total reactive nitrogen deposition (N$_{dep}$) and breakdown into inorganic wet and dry, oxidized (NO$_y$) and reduced (NH$_x$) deposition estimates at the 31 forest sites (evergreen needleleaf EN1-7 (spruce), EN8-18 (pine), mixed MF, deciduous broadleaf DB, evergreen broadleaf EB), and at 9 short semi-natural (SN) vegetation sites of the NitroEurope monitoring network. Data are arithmetic means over the years 2007-2010 of i) inferential dry deposition estimates by four different models based on in situ atmospheric N$_r$ measurements, and ii) of different wet deposition estimates from precipitation monitoring datasets and from European-scale atmospheric chemistry and transport modelling (EMEP). Error bars indicate standard deviations of the four dry deposition models (red bars) and standard deviations of the different data sources for inorganic N$_r$ wet deposition (blue bars). Wet deposition of water-soluble organic nitrogen (WSON) was measured at a few selected sites and is shown here for comparison with total inorganic N$_r$ deposition.**







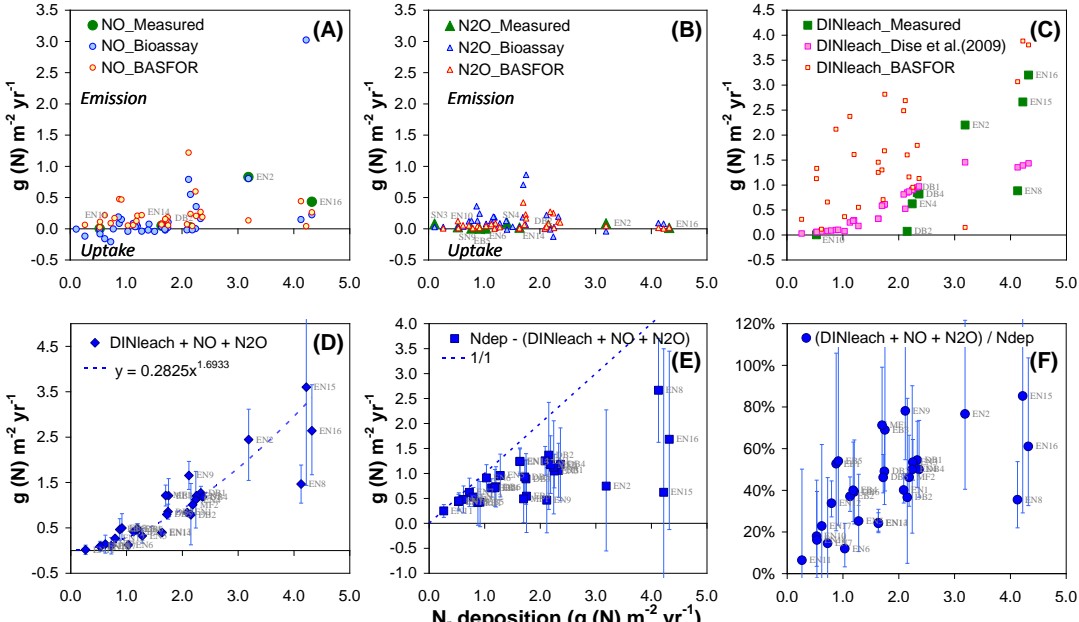

**Figure 3. Comparison of measured and estimated ecosystem inorganic N$_r$ losses and their relationships to total atmospheric N$_r$ deposition (x-axis) at the forest sites. NO fluxes (A) and N$_2$O fluxes (B) were either i) measured in situ using static or dynamic flux chambers, ii) scaled up from laboratory bioassay-derived T/WFPS relationships, or iii) simulated using the BASFOR ecosystem model (see text for details). DIN leaching (C) was either measured (lysimeter or suction cups), or predicted from the Dise et al. (2009) empirical algorithm. The sum of inorganic N$_r$ losses (DINleach + NO+ N$_2$O) was computed as the mean of measured values and modelled estimates. In panels A-C, site names are indicated for sites where in situ measurements were available.**








**Figure 4. Overview of inter-annual mean EC-derived C flux estimates ( GPP, R$_{eco}$ and NEP), ecosystem LAI and leaf N content, in relation to total (dry + wet) atmospheric N$_r$ deposition (A-E), and relationship of R$_{eco}$ to GPP (F), for forests (filled circles, black labels) and short semi-natural vegetation (filled stars, magenta labels). In all plots, the colour scale indicates mean annual temperature (MAT), while the symbol size is proportional to mean annual precipitation (MAP, scale provided in panel A).**



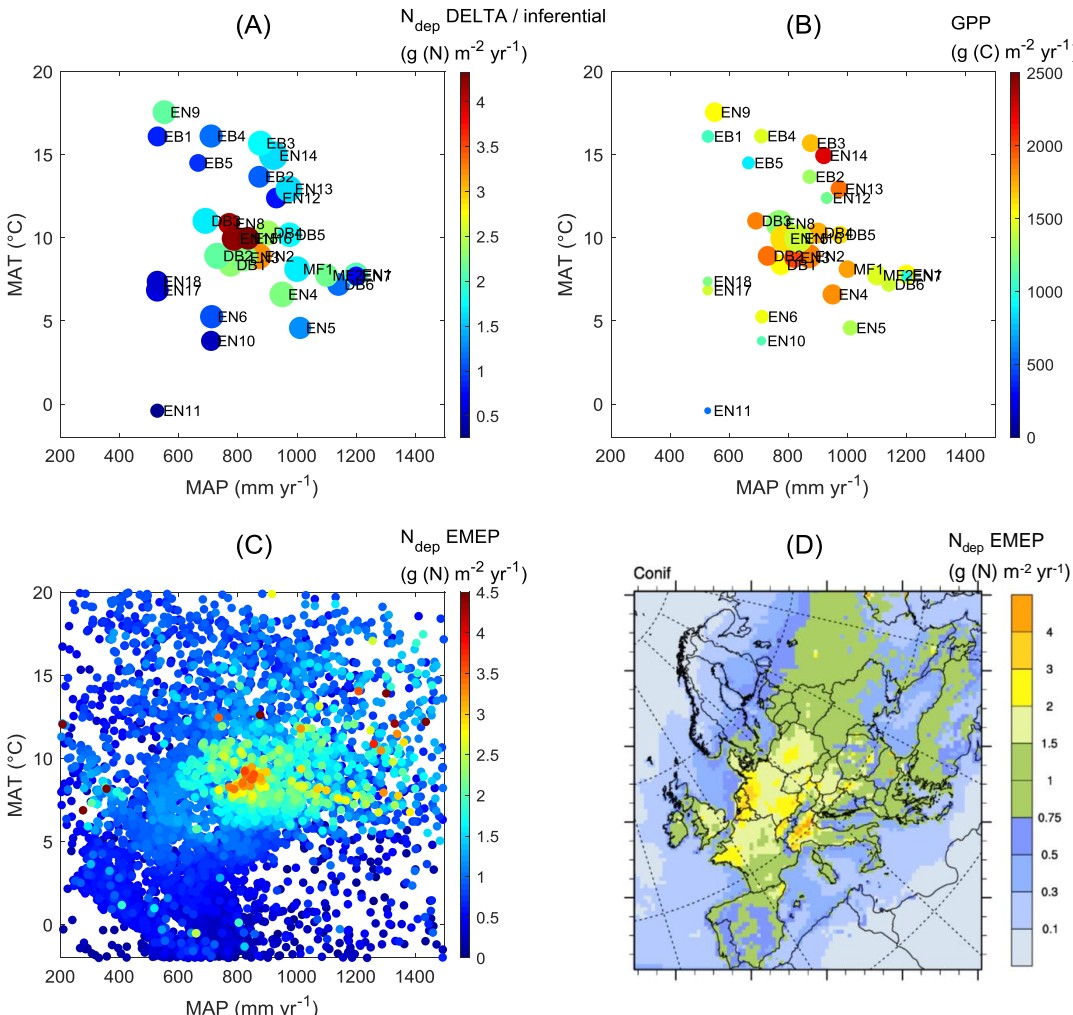

**Figure 5. Distribution of observation-based nitrogen deposition (N$_{dep}$) (A) and gross primary productivity (GPP) (B) for the forest sites of this study, within the European climate space represented by mean annual temperature (MAT) and precipitation (MAP). In plot A the symbol color indicates N$_{dep}$ while the symbol size is proportional to GPP; in plot B the symbol color indicates GPP, while the symbol size is proportional to N$_{dep}$. Plots C shows modelled N$_{dep}$ from the EMEP model over coniferous forests (year 2010), represented in climate space (1 data point for each grid square of the EMEP domain containing coniferous forests), also shown as a map (D). The MAT axis can be seen as a proxy for latitude and/or elevation, while the MAP axis expresses to some extent longitude (distance to the ocean) and/or orographic precipitation enhancement.**





**Figure 6.** BASFOR baseline simulations for all forest sites; model outputs and observation-based values were averaged over the years between the first and last available observations. Note that model simulations include MF and EBF sites, for which the model was not calibrated in Cameron et al. (2018); the two MF runs were made using the parameter table for DBF, while the five EBF runs were made using the parameter table for ENF to allow continued growth throughout the year. H: mean tree height; DBH: mean diameter at breast height; CLBS, CSOM, CR, CLITT: carbon stocks in leaves, branches and stems, in soil organic matter, in roots, and in litter layers, respectively; MAE: mean absolute error; NRMSE: root mean square error normalised to the mean.






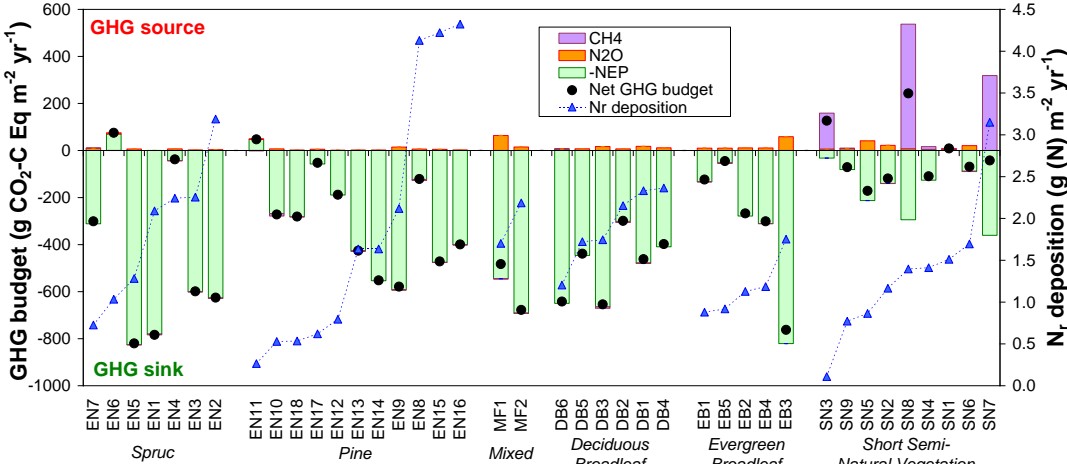


**Figure 7.** Net greenhouse gas (GHG) budgets calculated from a combination of inter-annual mean (around 2005-2010) net ecosystem productivity (NEP) from eddy covariance, and $N_2O$ and $CH_4$ flux data measured in situ or estimated by extrapolated bioassay data and forest ecosystem BASFOR modelling. Global warming potential values (100-yr time horizon) of 265 and 28 were used for $N_2O$ and $CH_4$, respectively; the sign convention is with respect to the atmosphere, negative for a sink, positive for a source. The data were grouped by ecosystem type (evergreen needleleaf EN-spruce and EN-pine, MF-mixed forests, DB-deciduous broadleaf, EB-evergreen broadleaf, SN-short semi-natural vegetation); within each group the data were sorted by increasing $N_r$ deposition.






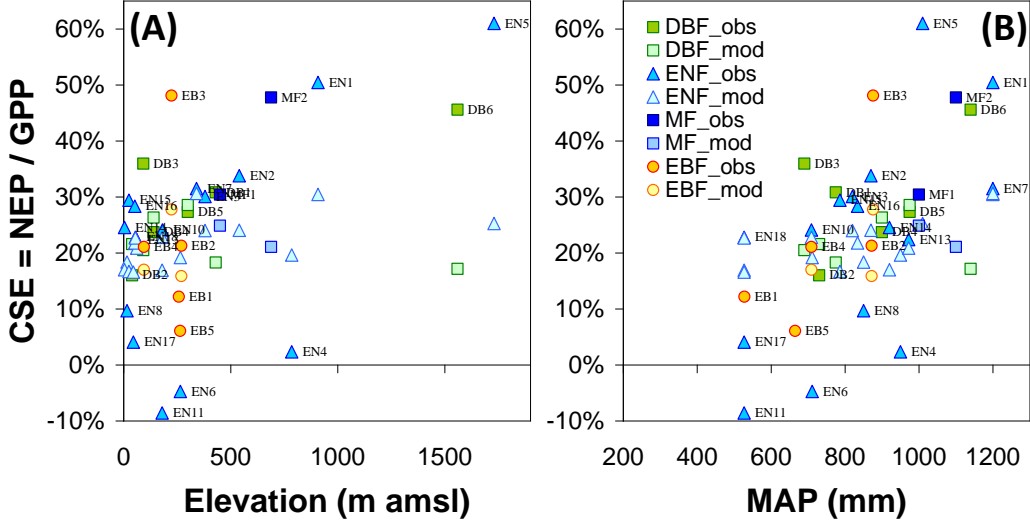

**Figure 8. Variability of observation-based and modelled carbon sequestration efficiency (CSE, defined as the NEP/GPP ratio), as a function of (A) site elevation above mean sea level (m), and (B) MAP: mean annual precipitation (mm).**





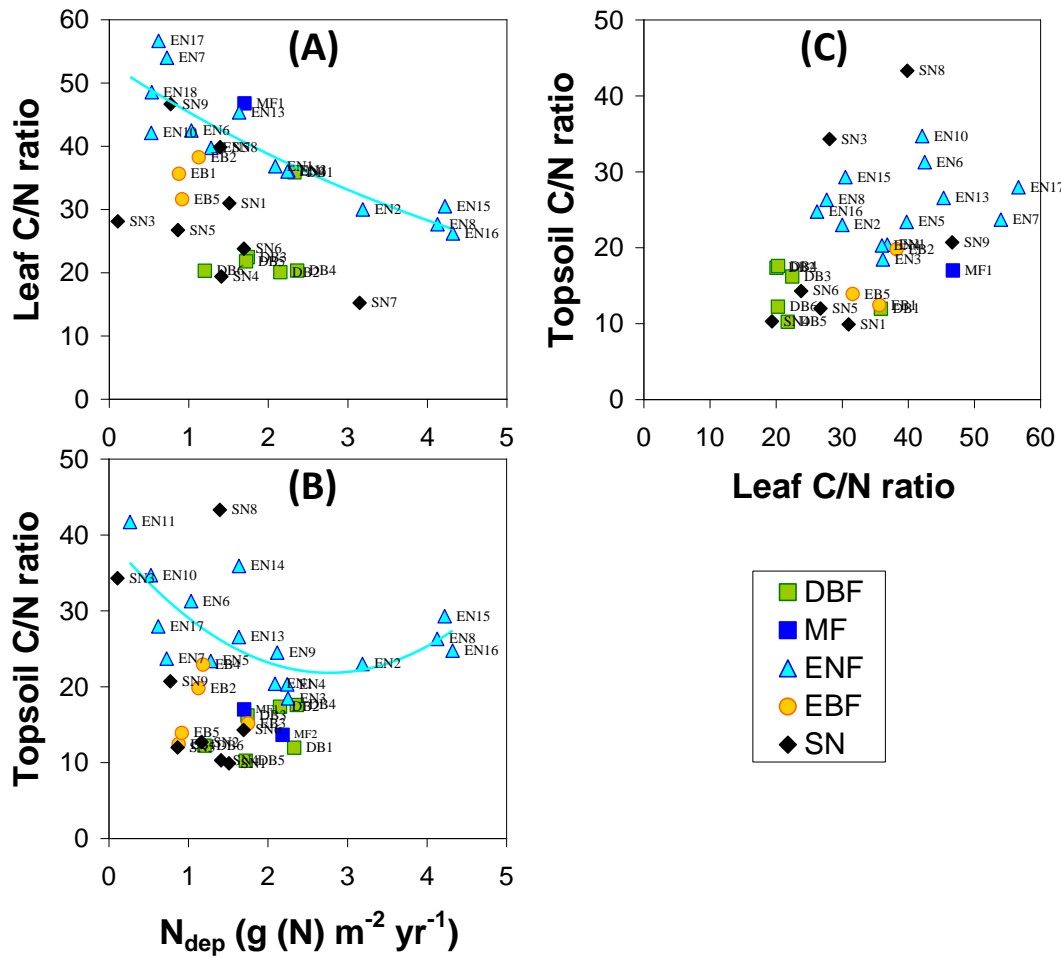

**Figure 9.** Relationships of leaf (A) and top soil (B) C/N ratios with atmospheric nitrogen deposition ($N_{dep}$), and to each toher (C), in different ecosystem types (DBF: deciduous broadleaf forests, MF: mixed forests, ENF: evergreen needleleaf forests, EBF: evergreen broadleaf forests, SN: short semi-natural vegetation).






**Table 1. Overview of ecosystem and climatic characteristics and inter-annual mean ecosystem/atmosphere exchange fluxes for forest and semi-natural short vegetation sites.**

| Site Name | Location, Country | PFT[1] Short name | Dominant vegetation | Forest age (2010) | Hmax[2] m | LAImax[3] m² m⁻² | Lat. °N | Long. °E | Elevation[4] m amsl | MAT[5] °C | MAP[6] mm | Ndep[7] g N m⁻² yr⁻¹ | GPP[8] g C m⁻² yr⁻¹ | Reco[9] g C m⁻² yr⁻¹ | NEP[10] g C m⁻² yr⁻¹ |
|---|---|---|---|---|---|---|---|---|---|---|---|---|---|---|---|
| DE-Hai | Hainich, Germany | DB1 | *Fagus sylvatica* | 142 | 23 | 4.0 | 51.079 | 10.452 | 430 | 8.4 | 775 | 2.3 | 1553 | 1074 | 479 |
| DK-Sor | Soro, Denmark | DB2 | *Fagus sylvatica* | 91 | 31 | 4.6 | 55.487 | 11.646 | 40 | 8.9 | 730 | 2.2 | 1883 | 1581 | 301 |
| FR-Fon | Fontainebleau, France | DB3 | *Quercus robur, Quercus petraea* | 111 | 28 | 5.1 | 48.476 | 2.780 | 92 | 11.0 | 690 | 1.7 | 1850 | 1185 | 665 |
| FR-Fgs | Fougères, France | DB4 | *Fagus sylvatica* | 41 | 20 | 6.0 | 48.383 | -1.185 | 140 | 10.3 | 900 | 2.4 | 1725 | 1316 | 409 |
| FR-Hes | Hesse, France | DB5 | *Fagus sylvatica* | 45 | 16 | 6.7 | 48.674 | 7.066 | 300 | 10.2 | 975 | 1.7 | 1634 | 1187 | 446 |
| IT-Col | Collelongo, Italy | DB6 | *Fagus sylvatica* | 120 | 22 | 5.7 | 41.849 | 13.588 | 1560 | 7.2 | 1140 | 1.2 | 1425 | 776 | 650 |
| CZ-BK1 | Bily Kriz, Czech Rep. | EN1 | *Picea abies* | 33 | 13 | 9.8 | 49.503 | 18.538 | 908 | 7.8 | 1200 | 2.1 | 1548 | 767 | 781 |
| DE-Hoe | Höglwald, Germany | EN2 | *Picea abies* | 104 | 35 | 6.3 | 48.300 | 11.100 | 540 | 8.9 | 870 | 3.2 | 1856 | 1229 | 627 |
| DE-Tha | Tharandt, Germany | EN3 | *Picea abies* | 120 | 27 | 6.7 | 50.964 | 13.567 | 380 | 8.8 | 820 | 2.3 | 1997 | 1396 | 601 |
| DE-Wet | Wetzstein, Germany | EN4 | *Picea abies* | 56 | 22 | 7.1 | 50.453 | 11.458 | 785 | 6.6 | 950 | 2.2 | 1809 | 1767 | 43 |
| IT-Ren | Renon, Italy | EN5 | *Picea abies* | 111 | 29 | 5.1 | 46.588 | 11.435 | 1730 | 4.6 | 1010 | 1.3 | 1353 | 528 | 826 |
| RU-Fyo | Fyodorovskoye, Russia | EN6 | *Picea abies* | 190 | 21 | 2.8 | 56.462 | 32.922 | 265 | 5.3 | 711 | 1.0 | 1488 | 1559 | -70 |
| UK-Gri | Griffin, UK | EN7 | *Picea sitchensis* | 29 | 12 | 6.5 | 56.617 | -3.800 | 340 | 7.7 | 1200 | 0.7 | 989 | 677 | 311 |
| BE-Bra | Brasschaat, Belgium | EN8 | *Pinus sylvestris* | 82 | 21 | 1.9 | 51.309 | 4.521 | 16 | 10.8 | 850 | 4.1 | 1272 | 1149 | 123 |
| ES-ES1 | El Saler, Spain | EN9 | *Pinus halepensis* | 111 | 10 | 2.6 | 39.346 | -0.319 | 5 | 17.6 | 551 | 2.1 | 1552 | 960 | 593 |
| FI-Hyy | Hyytiälä, Finland | EN10 | *Pinus sylvestris* | 48 | 18 | 3.4 | 61.848 | 24.295 | 181 | 3.8 | 709 | 0.5 | 1114 | 845 | 268 |
| FI-Sod | Sodankylä, Finland | EN11 | *Pinus sylvestris* | 100 | 13 | 1.2 | 67.362 | 26.638 | 180 | -0.4 | 527 | 0.3 | 551 | 598 | -47 |
| FR-Bil | Bilos, France | EN12 | *Pinus pinaster* | 9 | 4 | 0.5 | 44.522 | -0.896 | 50 | 12.4 | 930 | 0.8 | 1178 | 989 | 189 |
| FR-LBr | Le Bray, France | EN13 | *Pinus pinaster* | 41 | 22 | 1.9 | 44.717 | -0.769 | 61 | 12.9 | 972 | 1.6 | 1906 | 1479 | 427 |
| IT-SRo | San Rossore, Italy | EN14 | *Pinus pinaster* | 61 | 18 | 4.0 | 43.728 | 10.284 | 4 | 14.9 | 920 | 1.6 | 2256 | 1702 | 554 |
| NL-Loo | Loobos, Netherlands | EN15 | *Pinus sylvestris* | 101 | 18 | 1.5 | 52.168 | 5.744 | 25 | 10.0 | 786 | 4.2 | 1617 | 1141 | 476 |
| NL-Spe | Speulderbos, Netherlands | EN16 | *Pseudotsuga menziesii* | 51 | 32 | 7.5 | 52.252 | 5.691 | 52 | 10.0 | 834 | 4.3 | 1416 | 1015 | 401 |
| SE-Nor | Norunda, Sweden | EN17 | *Pinus sylvestris* | 112 | 28 | 4.6 | 60.083 | 17.467 | 45 | 6.8 | 527 | 0.6 | 1414 | 1356 | 58 |
| SE-Sk2 | Skytorp, Sweden | EN18 | *Pinus sylvestris* | 39 | 16 | 3.2 | 60.129 | 17.840 | 55 | 7.4 | 527 | 0.5 | 1235 | 953 | 282 |
| ES-LMa | Las Majadas, Spain | EB1 | *Quercus ilex* | 111 | 8 | 0.6 | 39.941 | -5.773 | 258 | 16.1 | 528 | 0.9 | 1091 | 958 | 133 |
| FR-Pue | Puechabon, France | EB2 | *Quercus ilex* | 69 | 6 | 2.9 | 43.741 | 3.596 | 270 | 13.7 | 872 | 1.1 | 1309 | 1030 | 279 |
| IT-Ro2 | Roccarespampani, Italy | EB3 | *Quercus cerris* | 21 | 16 | 3.8 | 42.390 | 11.921 | 224 | 15.7 | 876 | 1.8 | 1707 | 886 | 821 |
| PT-Esp | Espirra, Portugal | EB4 | *Eucalyptus globulus* | 25 | 20 | 2.7 | 38.639 | -8.602 | 95 | 16.1 | 709 | 1.2 | 1473 | 1163 | 311 |
| PT-Mi1 | Mitra, Portugal | EB5 | *Quercus ilex, Quercus suber* | 91 | 8 | 3.4 | 38.541 | -8.000 | 264 | 14.5 | 665 | 0.9 | 870 | 817 | 53 |
| BE-Vie | Vielsalm, Belgium | MF1 | *Fagus sylvatica, Pseudotsuga menziesii* | 86 | 30 | 5.1 | 50.305 | 5.997 | 450 | 8.1 | 1000 | 1.7 | 1792 | 1247 | 545 |
| CH-Lae | Lägeren, Switzerland | MF2 | *Fagus sylvatica, Picea abies* | 111 | 30 | 3.6 | 47.478 | 8.365 | 689 | 7.7 | 1100 | 2.2 | 1448 | 757 | 692 |
| DE-Meh | Mehrstedt, Germany | SN1 | Afforestated grassland | n.a. | 0.5 | 2.9 | 51.276 | 10.657 | 293 | 9.1 | 547 | 1.5 | 1171 | 1175 | -4 |
| ES-VDA | Vall d'Alinya, Spain | SN2 | Upland grassland | n.a. | 0.1 | 1.4 | 42.152 | 1.448 | 1765 | 6.4 | 1064 | 1.2 | 669 | 528 | 140 |
| FI-Lom | Lompolojänkkä, Finland | SN3 | Peatland | n.a. | 0.4 | 1.0 | 67.998 | 24.209 | 269 | -1.0 | 521 | 0.1 | 377 | 345 | 32 |
| HU-Bug | Bugac, Hungary | SN4 | Semi-arid grassland | n.a. | 0.5 | 4.7 | 46.692 | 19.602 | 111 | 10.7 | 500 | 1.4 | 1044 | 918 | 126 |
| IT-Amp | Amplero, Italy | SN5 | Upland grassland | n.a. | 0.4 | 2.5 | 41.904 | 13.605 | 884 | 9.8 | 1365 | 0.9 | 1241 | 1028 | 213 |
| IT-MBo | Monte Bondone, Italy | SN6 | Upland grassland | n.a. | 0.3 | 2.5 | 46.029 | 11.083 | 1550 | 5.1 | 1189 | 1.7 | 1435 | 1347 | 89 |
| NL-Hor | Horstermeer, Netherlands | SN7 | Peatland | n.a. | 2.5 | 6.9 | 52.029 | 5.068 | -2 | 10.8 | 800 | 3.1 | 1584 | 1224 | 361 |
| PL-wet | POLWET/Rzecin, Poland | SN8 | Wetland (reeds, sedges, mosses) | n.a. | 2.1 | 4.9 | 52.762 | 16.309 | 54 | 8.5 | 550 | 1.4 | 937 | 642 | 295 |
| UK-AMo | Auchencorth Moss, UK | SN9 | Peatland | n.a. | 0.6 | 2.1 | 55.792 | -3.239 | 270 | 7.6 | 1165 | 0.8 | 786 | 705 | 81 |

(1) PFT (plant functional types): DB: deciduous broadleaf forest; EN: evergreen needleleaf coniferous forest; EB: evergreen broadleaf Mediterranean forest; MF: mixed deciduous/coniferous forest; SN: short semi-natural, including moorland, peatland, shrubland and unimproved/upland grassland; (2) maximum canopy height; (3) maximum leaf area index, defined as 1-sided or half of total; (4) above mean sea level; (5) mean annual temperature; (6) mean annual precipitation; (7) nitrogen deposition; (8) gross primary productivity; (9) ecosystem respiration; (10) net ecosystem productivity; n.a.: not available/ not applicable.



**Table 2. Main acronyms and abbreviations used in the study**

| | |
|---|---|
| **Carbon fluxes and stocks** | |
| NEE | Net ecosystem exchange |
| GPP | Gross primary productivity |
| NPP | Net primary productivity |
| NEP | Net ecosystem productivity |
| NECB | Net ecosystem carbon balance |
| NBP | Net biome productivity |
| $R_{eco}$ | Ecosystem respiration |
| $R_{aut}$ | Autotrophic respiration |
| $R_{het}$ | Heterotrophic respiration |
| $R_{soil}$ | Soil (heterotrophic and rhizospheric) respiration |
| SCE | Soil $CO_2$ efflux measured by chamber methods |
| $CSE_{obs}$, $CSE_{mod}$ | Carbon sequestration efficiency, calculated from EC observations or by modelling |
| SOM | Soil organic matter |
| CSOM | Carbon stock in soil organic matter |
| CR | Carbon stock in roots |
| CLITT | Carbon stock in litter layers of the forest floor |
| CLBS | Carbon stock in leaves, branches and stems |
| LeafC | Leaf carbon content |
| DIC, DOC | Dissolved inorganic or organic carbon |
| dC/dN, $dNEP/dN_{dep}$ | Response (slope) of ecosystem C productivity versus atmospheric $N_r$ deposition |
| **Nitrogen fluxes and stocks** | |
| $N_{dep}$ | Total (wet+dry) atmospheric reactive nitrogen deposition |
| $N_r$ | Reactive nitrogen |
| $N_{min}$, $N_{org}$ | Mineral or organic reactive nitrogen forms |
| LeafN | Leaf nitrogen content |
| DIN, DON | Dissolved inorganic or organic nitrogen |
| WSON | Wet deposition of water-soluble organic nitrogen |
| **Water budget terms** | |
| SWC | Soil water content |
| WFPS | Water-filled pore space |
| ET | Evapotranspiration |
| **Ecosystem characteristics** | |
| PFT | Plant functional type |
| ENF | Evergreen needleleaf forest |
| DBF | Deciduous broadleaf forest |
| MF | Mixed (needleleaf/broadleaf) forest |
| EBF | Evergreen broadleaf forest |
| SN | Short semi-natural vegetation |
| H | Canopy height |
| DBH | Tree diameter at breast height (forests) |
| LAI | Leaf area index |
| SD | Stand density (forests): number of trees per unit area |
| MAT | Mean annual temperature |
| MAP | Mean annual precipitation |
| **Methods and general terminology** | |
| EC | Eddy covariance |
| DELTA | DEnuder for Long-Term Atmospheric sampling |
| BASFOR | BASic FORest ecosystem model |
| CTM | Chemical transport model |
| EMEP | European Monitoring and Evaluation Programme (www.emep.int) |
| GHG | Greenhouse gas |
| GWP | Global warming potential |
| CEIP | CarboEurope Integrated Project |
| NEU | NitroEurope Integrated Project |
| FLUXNET | Worldwide carbon flux monitoring network |