# Peer review of "Carbon/nitrogen interactions in European forests and semi-natural vegetation. Part I: Fluxes and budgets of carbon, nitrogen and greenhouse gases from ecosystem monitoring and modelling"

_Biogeosciences, 2019_

## Referee Comment (RC1) · Anonymous Referee #1 · 4 Oct 2019

Review for the paper submitted by Flechard et al. MS No.: bg-2019-333

General Comment This is a very well written paper in the suite of other papers on N deposition at the European scale lead by the principal author. In fact, this paper acts as a "prequel" of the companion paper, the Part II dedicated to untangling climatic, edaphic, management and N deposition effects on C sequestration. A previous paper to Part II is important since uncertainties and gaps of knowledge associated to the different com-

ponents of the N and C cycles in terrestrial ecosystems need to be examined previously to the attempt of disentangling. Therefore, in this paper, to evaluate the uncertainties and gaps in the estimates in N and C budgets, the authors have made a remarkable effort of gathering N and C data from 31 forests and 9 seminatural ecosystems extended over Europe and covering a wide span of climates, from Mediterranean to boreal. To constrain the N budget they have taken advantage of local measurements of dry and wet deposition at specific sites from the NEU (NitroEurope) database, complemented with the use of deposition models (EMEP model) in some cases. Loss of N by nitrate leaching and by gaseous emissions have been estimated by measurements in some sites and modelling when no measurements were available. For the C budget, data were mostly obtained from eddy covariance sites within the CarboEurope Integrated Project (CEIP) combined with laboratory bioassays and literature mining. The results of this big effort of compilation constitute an important contribution to the evaluation of N deposition on C sequestration at a European scale, by critically evaluating the uncertainties in the quantification of some of the drivers. Also, it calls for attention to neglected fluxes that might have a considerable role in the budgets, e.g. N2 emissions by denitrification. The paper is well written, well documented, scientifically sound, and it fits the scope of Biogeosciences Discussions, so I recommend it for publication with only very minor changes.

Specific Comments Abstract The abstract summarizes the main findings, so it is very important to give accurate figures. In this sense, I suggest to review the sentence in line 110, since from Fig. 3F one can see a different range of values of % N losses to total N dep than those reported in the text (10-35% at Ndep below 1 gN m-1 yr-1 and 35-80% at Ndep above 3 gN m-1 yr-1 in the Figure). This sentence is followed with a consideration that 1/3 of the sites might be in a state of early – advanced N saturation. But, from Fig 3F, I deduce that one third of the sites result from considering a threshold value of 2 gN m-1 yr-1. I suggest rewriting this paragraph, also including a suggested of Ndep that might indicate early N saturation. Introduction I like it very much. In line 175, wetlands might be also included for DOC leaching. Methods For

dry deposition, the inferential method is based in the same 4 models as in Flechard et al (2011). However, it is not clear to me whether the retained value is the averaged estimate from the 4 models, can you clarify? Later on (line 533) it is mentioned that DD is calculated as ensemble average of 4 inferential models, but this should be stated and explained in Methods. For wet deposition, an estimated NO3 and NH4 deposition was attributed to every one of 40 sites through kriging interpolation of EMEP and ICP-Forest data. Furthermore, 13 sites were provided with BD samplers for 3 years so that BD Ndep was actually measured in these sites, and six more sites already were equipped with BD or WD collectors. Can you comment here on how well did compare the kriging estimates to the actual measures? When calculating losses by leaching, it is mentioned that lysimeters were used to obtain soil N (or DOC) concentrations that were combined with a hydrological drainage model to derive the export fluxes. Can you explain better this hydrological model? Was it possible, in any one of the sites of the survey, compare results from N and C exports calculated using the hydrological drainage model and from actual water runoff at gauged sites? I understand that this is not the main focus of the paper; however, just to know how the two approaches estimate losses can be of interest. Results In line 548, when considering organic N deposition, it is seen that WSON is a small fraction of Inorganic and organic N deposition. But, can you comment on the possible role of dry WSON (e.g. urea is important in some cases) deposition? Should this also need to be considered in the N budgets? And then, if the dry organic N flux is considered to be relevant, should it be included in Figure 1? When commenting on N losses, in line 556 reference is made to Fig 3D to indicate greater losses with Ndep above 2 gN m-1 yr-1. It should be specified that this statement is based on measured leaching and Dise's leaching model, but not on BASFOR estimates. Discussion The N balance is presented in Fig 3D (N losses compared to N inputs) and it is shown that a non-linear fit best describes this relationship. Then the authors argue that above a Ndep of 4 gN m-2 yr-1, N losses might "even exceed " the estimated N deposition, but this should occur when extrapolating the line into a region devoided of data. On the other hand, sites of lower N dep (e.g. EN9) have a leaching loss as close to the N

dep value than other sites with higher deposition. In my opinion, the pattern towards N saturation is best shown in Fig. 3F, when plotting the % of N losses to Ndep. In this plot, to me it is clear that, at Ndep above 2 gN m-2 yr-1, all site mean leaching values are above 35%. The classification in three ranges of depositions (low, intermediate and high) is OK, but as commented in the Abstract, the % ranges of N losses to Ndep should be revised (e.g. for Ndep above 3 gN m-2 yr-1, mean loss% ranges from 35 to 80%.

Minor corrections Line 524: large Line 569: include here Fig. 3A after Ndep 2 g m-2 yr-1 Line 577: Fig 3A Line 586: why include here the units in kg ha-1 yr-1, besides g m-2 y-1? Line 631: This inter-annual peak in LAI, is it the average of various years? What does it mean "peak"? Line 574: better than? Maybe use: provide a good estimate... Fig 6: for CSOM, r2 = 0.00, but this seems too low given the distribution of points...can you revise it? Fig. 9: include regression, r2 and p in plot A and B. Figures: Identification of sites are generally difficult to read, especially in Figs. 3 and 6

---

## Referee Comment (RC2) · Anonymous Referee #2 · 5 Oct 2019

General comments: This article presents estimates of nitrogen, carbon and greenhouse fluxes and budgets at 40 European flux towers compiled from different sources (observations, models, literature). Overall, the article contains an integrated approach to estimate the total nitrogen flux at these sites as complete as possible, accounting for different N pathways. It includes a lot of useful measurements, such as local wet and dry Nr deposition from a collocated measurement network, as well as other in-situ observations (NO, N2O, soil samples etc.). These nitrogen flux estimates are comple-

mented with carbon and greenhouse fluxes, as well as auxiliary information (such as climate variables, forest characteristics, etc.), and as such provide a useful database as a basis for assessment of carbon/nitrogen interactions in European ecosystems, which is also in part included in this article. Even though the content of this article is useful and interesting, it is very lengthy and it can benefit largely from shortening and restructuring. It contains plenty of information which could be transferred as Supplementary information. The materials and method section is for instance very long (almost 8 pages) and puts a lot of emphasis on some of the different measurements and data sources. I would highly recommend adding a schematic overview of the different data sources at the beginning of this section to make clear what is used and from what source. Also, a couple of subsections may be added to the 'nitrogen fluxes' section to subdivide the text into different components. The section about the BASFOR forest ecosystem model is also unnecessarily lengthy, while the output of this model is only a small part of the discussed results. To make the article more focused, I recommend to greatly shorten this section or even better refer to existing publications. The results section seems to focus on two major parts: one being the discussion of the resulting nitrogen, carbon and greenhouse fluxes and budgets themselves (and their inter-comparison and validation) and the other being the link between these N and C fluxes and the interpretation. I would recommend separating these two parts as much as possible. Moreover, the "results" and "discussion" sections seem to be partially intertwined. It would be better to exclusively include discussion points in the latter. Some sections, for instance, the one about "carbon sequestration efficiency", introduce new concepts and results and seem to fit better in the results section (see specific comments below). As for the "discussion" section, it would be more structured if there was a clear separation between the methodological uncertainties used in the estimation of the N, C and greenhouse gas fluxes and the uncertainties related to the interplay of N and C. Some of the results need to be updated, e.g. in section 2.2.1 it is written that EMEP data were downloaded in 2013 and different model versions were used? In a lot of statements there is an unevidenced qualification, such as: 'there did not appear to

be any systematic overestimation ..', '0,85 is plausible but also much variability ..', We assumed that all sites of the European network followed the same relative time course ..', 'reveals a potential cross correlation ..', 'reasonable overall model performance ..', etc. It would help if such statements are underpinned/quantified. Finally, there seems to be some parts of the paper that are outdated, such as the start of the discussion about the Magnani paper. These can be omitted. It raises the question to me about what the purpose of the paper is and how it relates to the second part. If it is a description of the database, including reference to the data sources, models and including an uncertainty estimate it can be a much shorter well focused paper with detailed information as supplementary information. The second part can then focus on the use and interpretation of the data. It would be my recommendation to split it in such a way. I therefore recommend major revision of the paper. Specific comments: Line 127-130 – The information in the introduction about tropical forest could be shortened, as the paper focusses on temperate/boreal forest. Line 159 – 1 by 1 degree does not correspond to 10 by 10 km, it covers a larger area. Line 195 – How about vegetation changes at all sited, e.g. due to growth and/or composition change? Line 220 - Not all FLUXNET sites seem to be included in this analysis, what are the criteria used to in- or exclude individual sites? Line 262 – In this section it is sometimes unclear to what "models" are referred. Specify what type of models, e.g. dry deposition models. What are the key differences between these deposition models? It would also be good to specify the EMEP model domain and resolution in this section. Line 284 – leaving out the organic N data leads to a systematic bias. Line 290 – There are probably new data available? Line 308 – Could you elaborate on why you decided to scale up the N2O and NO fluxes using linear interpolation? Line 326 – DINTF is not included in the abbreviations table. Line 334 – there is an enormous range, why not used in some way? Line 336 – This section, about the EC processing, is lengthy. Consider shortening it. Line 389 – Missing comma between "some sites" and "such as" Line 427 – Add a reference to the table in the supplement that includes the major publications per site. Line 480 – It would make more sense to move the primary purpose of the BASFOR

model to the beginning of the section. Line 538 – 550 – The discussion on the model uncertainties would fit better in the discussion section. Line 580 – 594 – This section discusses the computation of the denitrification losses and the uncertainty associated with it. Consider moving a part of this section to the methods/discussion. Line 603 – 612 – It seems a bit sudden to address the ultimate objective of the whole project here. It would be better to discuss this earlier on in the paper. Line 640 – You state that there is a "broad negative correlation between MAP and MAT", could you support this with a R2 value? Moreover, the MAP alone does not seem appropriate to address site-specific water-availability, without considering other parameters such as soil water holding capacity etc. Line 647 – Missing "were" Line 659 – 674 – In this section you are discussing results related to Figure 4 again. Consider moving this section up. Line 737 – Add a reference for this number. Line 754 – how can you demonstrate that it 'provides a better estimate' and that it is more realistic than an transport model? Line 804 – It would be better to either add the definitions for N saturation here or refer to this later. Line 930 – Why is the uncertainty in the non-CO2 fluxes possibly >100% larger? Line 972–975 – You state here that your attempts with more advanced forms of regression analysis were not successful. This is a bit on the vague side. You should either elaborate on the attempts that were made and/or for instance add some references, or exclude this statement from this section. Line 1016 – It is unclear why considering the uncertainties would amount to a confident threshold of 2-2.5 g N, rather than just widening the range of the given estimate, could you elaborate on this? Line 1052 – 1055 – This sentence is a bit confusing. It refers to the large uncertainty in dry deposition modelling, but both deposition estimates from CTM and denuders make use of dry deposition models in the end. So I would say that it is not so much the deposition models that improved substantially, but more the estimates of the concentrations of these compounds. Figure 4 – What type of LAI data is used to determine LAImax? Figure 8 – Some of the labels seem to be missing. Figure 9 – "to each tohet" – to each other Consider splitting up / reformulating these sentences for clarity/readability: Line 119-121 – "The global terrestrial . . . Pg (C ) yr-1" Line 127-130 – "Tropical forest areas

. . . generally believed." Line 297-299 – "Nitrogen losses to . . . empirical methods." Line 308 -309 – "To address . . . responses of soils." Line 525-528 – "Total Nr deposition . . . Schwede et al., 2018)." Line 548-550 – "Wet deposition . . . most forest sites." Line 754-756 – "Despite these . . . cycling processes." Line 790-793 – "In addition . . . as beech or oak." Line 824-828 – "By analogy . . . observation periods." Line 959-963 – "Through the continent-wide. . . continental scale." Line 972-975 – "Our attempts . . . the dataset." Line 1052-1055 – "Nevertheless, . . . is much larger."
* * *

---

## Author Response (AR1)

**"Carbon / nitrogen interactions in European forests and semi-natural vegetation. Part I: Fluxes and budgets of carbon, nitrogen and greenhouse gases from ecosystem monitoring and modelling" by Chris R. Flechard et al.**

**Point by point reply to Referees' comments**

We are thankful to both referees for their interest in our study and for their constructive comments, which have helped improve the manuscript. For clarity's sake, we have provided our point-by-point **responses to each comment in blue**, and provided **changes to the manuscript text in green**.
* * *
**Anonymous Referee #1**

**General Comment** This is a very well written paper in the suite of other papers on N deposition at the European scale lead by the principal author. In fact, this paper acts as a "prequel" of the companion paper, the Part II dedicated to untangling climatic, edaphic, management and N deposition effects on C sequestration. A previous paper to Part II is important since uncertainties and gaps of knowledge associated to the different components of the N and C cycles in terrestrial ecosystems need to be examined previously to the attempt of disentangling. Therefore, in this paper, to evaluate the uncertainties and gaps in the estimates in N and C budgets, the authors have made a remarkable effort of gathering N and C data from 31 forests and 9 seminatural ecosystems extended over Europe and covering a wide span of climates, from Mediterranean to boreal. To constrain the N budget they have taken advantage of local measurements of dry and wet deposition at specific sites from the NEU (NitroEurope) database, complemented with the use of deposition models (EMEP model) in some cases. Loss of N by nitrate leaching and by gaseous emissions have been estimated by measurements in some sites and modelling when no measurements were available. For the C budget, data were mostly obtained from eddy covariance sites within the CarboEurope Integrated Project (CEIP) combined with laboratory bioassays and literature mining. The results of this big effort of compilation constitute an important contribution to the evaluation of N deposition on C sequestration at a European scale, by critically evaluating the uncertainties in the quantification of some of the drivers. Also, it calls for attention to neglected fluxes that might have a considerable role in the budgets, e.g. N2 emissions by denitrification. The paper is well written, well documented, scientifically sound, and it fits the scope of Biogeosciences Discussions, so I recommend it for publication with only very minor changes.

We are grateful to Referee #1 for this positive assessment of our paper.

**Specific Comments**

Abstract The abstract summarizes the main findings, so it is very important to give accurate figures. In this sense, I suggest to review the sentence in line 110, since from Fig. 3F one can see a different range of values of % N losses to total N dep than those reported in the text (10-35% at Ndep below 1 gN m-1 yr-1 and 35-80% at Ndep above 3 gN m-1 yr-1 in the Figure). This sentence is followed with a consideration that 1/3 of the sites might be in a state of early – advanced N saturation. But, from Fig 3F, I deduce that one third of the sites result from considering a threshold value of 2 gN m-1 yr-1. I suggest rewriting this paragraph, also including a suggested of Ndep that might indicate early N saturation.

We have provided the mean value, as well as the range, for inorganic N losses at $N_{dep}$ below 1 g N m$^{-2}$ yr$^{-1}$ and $N_{dep}$ above 3 g N m$^{-2}$ yr$^{-1}$ (see below). However, as discussed in Section 4.3.2, there are several definitions of N saturation, which potentially lead to different thresholds, while the presence of different tree species, of different age classes and growing on very different soils, makes it difficult to pinpoint exactly at what $N_{dep}$ level N saturation begins, and what level corresponds to advanced N saturation. This is why we were cautious and admittedly a little vague ('perhaps one third') in the second part of the sentence. We stand by the decision to remain cautious about $N_{dep}$ thresholds for saturation in the abstract (while leaving the debate open in the discussion). The paragraph was rephrased thus:

'…Nitrogen losses in the form of NO, $N_2O$ and especially $NO_3^-$ were on average 27% (range 6-54%) of $N_{dep}$ at sites with $N_{dep}$ < 1 g (N) $m^{-2}$ $yr^{-1}$, versus 65% (range 35-85%) for $N_{dep}$ > 3 g (N) $m^{-2}$ $yr^{-1}$. Such large levels of $N_r$ loss likely indicate that different stages of N saturation occurred at a number of sites. The joint analysis of the C and N budgets provided further hints that N saturation could be detected in altered patterns of forest growth. Net ecosystem productivity increased with $N_r$ deposition up to 2-2.5 g (N) $m^{-2}$ $yr^{-1}$, …'

Introduction I like it very much. In line 175, wetlands might be also included for DOC leaching.

Yes, wetlands should in fact be mentioned first, before grasslands and croplands. A mention and reference was added: '…leaching can also be significant, especially for **wetlands (Dinsmore et al., 2010) and also** grassland and cropland ecosystems (Kindler et al., 2011; Gielen et al., 2011). This is relevant for…'

Methods For dry deposition, the inferential method is based in the same 4 models as in Flechard et al (2011). However, it is not clear to me whether the retained value is the averaged estimate from the 4 models, can you clarify?

Later on (line 533) it is mentioned that DD is calculated as ensemble average of 4 inferential models, but this should be stated and explained in Methods.

Yes, the ensemble average of the same 4 models was used for dry deposition. This is already stated in section 2.2.1, lines 264-265: '…*we tried here to minimise such uncertainties by using the ensemble average dry deposition predicted by four different models, as in Flechard et al. (2011)*.'

For wet deposition, an estimated NO3 and NH4 deposition was attributed to every one of 40 sites through kriging interpolation of EMEP and ICP-Forest data. Furthermore, 13 sites were provided with BD samplers for 3 years so that BD Ndep was actually measured in these sites, and six more sites already were equipped with BD or WD collectors. Can you comment here on how well did compare the kriging estimates to the actual measures?

This issue is explored in Results section 3.1.1, lines 543-547. We have written that '… *By contrast, wet deposition was generally reasonably consistent between the different data sources for inorganic Nr (in situ bulk or wet-only measurement, kriging of monitoring network data, EMEP model output). For the 18 sites where all three sources of data were available, the mean CV of the three estimates was 21% (range 2%-56%, with 15 CV values out of 18 below 30%), and the mean (± 95% conf. int.) wet deposition estimates across the 18 sites were 0.63 ±0.14, 0.64 ±0.15 and 0.68 ±0.16 g (N) m-2 yr-1 for the three methods, respectively (Fig. S2), showing no systematic bias between methods…*' Figure S2 in the supplement shows the comparison between precipitation samplers, EMEP model, and spatial interpolation of network data.

When calculating losses by leaching, it is mentioned that lysimeters were used to obtain soil N (or DOC) concentrations that were combined with a hydrological drainage model to derive the export fluxes. Can you explain better this hydrological model? Was it possible, in any one of the sites of the survey, compare results from N and C exports calculated using the hydrological drainage model and from actual water runoff at gauged sites? I understand that this is not the main focus of the paper; however, just to know how the two approaches estimate losses can be of interest.

Several approaches were used to quantify dissolved C and N losses, and the sampling methods and hydrological models were site- or paper-specific. Unfortunately, there was no comparison anywhere of C/N exports estimated by actual water runoff and by lysimeter/suction cup setups. It would be difficult to describe each hydrological model in any detail; however, the text (lines 410-411) does provide references to the papers on DIC/DOC, in which the hydrological approaches are described: Kindler et al., 2011; Gielen et al., 2011; Verstraeten et al., 2014. We have added the following sentence and references to section 2.2.2 (ca line 325), where the method is first mentioned for the case of DIN leaching: "…One-dimensional (1-D) drainage models were based on the soil water balance equation using evapotranspiration, observed precipitation and changes in soil water content (Kindler et al., 2011; Gielen et al., 2011).

Results

In line 548, when considering organic N deposition, it is seen that WSON is a small fraction of Inorganic and organic N deposition. But, can you comment on the possible role of dry WSON (e.g. urea is important in some cases) deposition? Should this also need to be considered in the N budgets? And then, if the dry organic N flux is considered to be relevant, should it be included in Figure 1?

We had indeed omitted to show a conceptual arrow for $N_{org}$ dry deposition in Fig.1; this was included in the revised version. We agree that the dry deposition of organic $N_r$ (ON) is in principle a component of the total $N_{dep}$ budget, but we believe this is a minor fraction of total $N_{dep}$. Gas-phase organic $N_r$ (e.g. PAN and other organic nitrates, amines) have relatively small ambient concentrations and deposition velocities, as discussed in Flechard et al. (2011), and as shown by EMEP model calculations involving PAN and organic nitrates from isoprene chemistry. However, Kanakidou et al. (2016) suggest that particulate ON largely dominates the atmospheric ON load, and for particles the main atmospheric removal mechanism is through precipitation.

The original article version did mention that dry deposition of organic $N_r$ is not quantified, p19, lines 808-809: '…*incoming organic nitrogen in precipitation (WSON) as well as dry deposition of organic Nr species, not quantified here (Fig. 1)…*' But we have added the following sentence to Section 2.2.1, line 265, to explain why we think this can be considered minor:

'…The dry deposition of atmospheric organic $N_r$ (ON) species not accounted for by the EMEP model (e.g. amines, urea), and not included in DELTA measurements, can contribute a fraction of total $N_r$ deposition. However, Kanakidou et al. (2016) suggest that particulate ON largely dominates the atmospheric ON load, and for particles the main atmospheric removal mechanism is through precipitation. Thus, dry deposition of ON is expected to be much smaller than wet deposition of water soluble organic compounds (see below).'

Additional reference:

Kanakidou, M., Myriokefalitakis, S., Daskalakis, N., Fanourgakis, G.S., Nenes, A., Baker, A.R.,Tsigaridis, K., Mihalopoulos, N.: Past, present, and future atmospheric nitrogen deposition, J. Atmos. Sci., 73, 2039–2047, https://doi.org/10.1175/JAS-D-15-0278.1, 2016.

When commenting on N losses, in line 556 reference is made to Fig 3D to indicate greater losses with $N_{dep}$ above 2 gN m-1 yr-1. It should be specified that this statement is based on measured leaching and Dise's leaching model, but not on BASFOR estimates.

The manuscript does mention, a few lines further down (560-563), that '… *the DIN leaching estimate by BASFOR, shown for comparison on Fig. 3C, was not used in the calculation of total inorganic N losses in Fig. 3D; this is because BASFOR does not simulate N2 loss by denitrification, and thus part of the soil N surplus that would in reality denitrify is assumed to drain, resulting in an over-estimation of the leaching term…*', so we don't believe it is necessary to further stress the point.

Discussion

The N balance is presented in Fig 3D (N losses compared to N inputs) and it is shown that a non-linear fit best describes this relationship. Then the authors argue that above a Ndep of 4 gN m-2 yr-1, N losses might "even exceed " the estimated N deposition, but this should occur when extrapolating the line into a region devoided of data. On the other hand, sites of lower N dep (e.g. EN9) have a leaching loss as close to the Ndep value than other sites with higher deposition. In my opinion, the pattern towards N saturation is best shown in Fig. 3F, when plotting the % of N losses to Ndep. In this plot, to me it is clear that, at Ndep above 2 gN m-2 yr-1, all site mean leaching values are above 35%. The classification in three ranges of depositions (low, intermediate and high) is OK, but as commented in the Abstract, the % ranges of N losses to $N_{dep}$ should be revised (e.g. for Ndep above 3 gN m-2 yr-1, mean loss% ranges from 35 to 80%.

What was meant by '…*N losses might even exceed the estimated N deposition*…' was actually not that N losses would exceed $N_{dep}$ if one extrapolates $N_{dep}$ to 5 or 6 g N m$^{-2}$ yr$^{-1}$ (a region in which we have no data indeed); what we meant was that the error bars on the total inorganic loss term are large when $N_{dep}$ > 4 g N m$^{-2}$ yr$^{-1}$ and the confidence interval overlaps a range where $N_{loss}$ can be larger than $N_{dep}$.

We have revised the sentence on the ranges of % Nlosses (lines 814-816) using mean values (and the range) for the three classes of $N_{dep}$, as suggested by the referee:

'…the large range of losses from 6% to 85%, with on average 27% loss (range 6-54%) for $N_{dep}$ < 1 g (N) m$^{-2}$ yr$^{-1}$, 45% loss (12-78%) for intermediate $N_{dep}$ levels, and 65% loss (35-85%) for $N_{dep}$ > 3 g (N) m$^{-2}$ yr$^{-1}$…'

**Minor corrections**

Line 524: large Done

Line 569: include here Fig. 3A after Ndep 2 g m-2 yr-1  Done

Line 577: Fig 3A Done

Line 586: why include here the units in kg ha-1 yr-1, besides g m-2 y-1?

We have deleted the value in kg ha-1 yr-1, we agree this is not needed.

Line 631: This inter-annual peak in LAI, is it the average of various years? What does it mean "peak"?

This is the inter-annual mean value of the annual maximum leaf area index. We have rephrased thus: '…The inter-annual mean value of the annual maximum leaf area index (LAI$_{max}$) increased from…'

Line 574: better than? Maybe use: provide a good estimate. . .

We are not sure what the referee means here, we do not find any instance of '*better than*' in line 574.

If the Referee means **line 754** instead of 574 (?), we agree we should specify that we compare local depositon estimates to the outputs of a large-scale chemical transport model:

'…*Despite these uncertainties, measuring gas-phase and aerosol Nr concentrations locally* should provide a better estimate of total ecosystem $N_r$ inputs than the outputs of a large-scale chemical transport model…'

Fig 6: for CSOM, r2 = 0.00, but this seems too low given the distribution of points. . .can you revise it?

CSOM is predictably a difficult variable to get right in forest ecosystem modelling, since this depends much less on carbon accumulated during the lifetime of the forest, than on the initial CSOM value at the start of the simulation, about which little is known in most cases. There is however stronger confidence in the change in CSOM over time than in the actual absolute CSOM value at any given time.

To answer the Referee's question, we are not sure how this can be revised. The R$^2$ is what it is. It is always possible to select data points to improve the correlation, but this is not necessarily helpful in this case.

Fig. 9: include regression, r2 and p in plot A and B.  Done

Figures: Identification of sites are generally difficult to read, especially in Figs. 3 and 6

We have increased font size for site labels in these 2 figures, and also moved labels to avoid overlapping. We also now use black instead of grey for site labels in Fig.3, which improves readability.

| Anonymous Referee #2 |
|---|

**General comments**: This article presents estimates of nitrogen, carbon and greenhouse fluxes and budgets at 40 European flux towers compiled from different sources (observations, models, literature). Overall, the article contains an integrated approach to estimate the total nitrogen flux at these sites as complete as possible, accounting for different N pathways. It includes a lot of useful measurements, such as local wet and dry Nr deposition from a collocated measurement network, as well as other in-situ observations (NO, N2O, soil samples etc.). These nitrogen flux estimates are complemented with carbon and greenhouse fluxes, as well as auxiliary information (such as climate variables, forest characteristics, etc.), and as such provide a useful database as a basis for assessment of carbon/nitrogen interactions in European ecosystems, which is also in part included in this article.

Even though the content of this article is useful and interesting, it is very lengthy and it can benefit largely from shortening and restructuring. It contains plenty of information which could be transferred as Supplementary information. The materials and method section is for instance very long (almost 8 pages) and puts a lot of emphasis on some of the different measurements and data sources. I would highly recommend adding a schematic overview of the different data sources at the beginning of this section to make clear what is used and from what source. Also, a couple of subsections may be added to the 'nitrogen fluxes' section to subdivide the text into different components. The section about the BASFOR forest ecosystem model is also unnecessarily lengthy, while the output of this model is only a small part of the discussed results. To make the article more focused, I recommend to greatly shorten this section or even better refer to existing publications.

We acknowledge that this is a long paper. This is because we have endeavoured, here (Part I) and also in Part II of this study (BG_2019-335), to bring together many of the results of two large European-scale projects (CarboEurope and NitroEurope), to investigate the linkages between the carbon cycle, the nitrogen cycle and greenhouse gas budgets, but also the interactions with the water cycle, climate and soil.

In this paper, we wish to demonstrate that significant patterns of interactions can be identified from the joint analysis of the various datasets involved. But, importantly, we also need to make it clear that, while significant advances have been made in measurements and observation networks, in our understanding of processes, and in biogeochemical and ecosystem models, there remain very significant uncertainties in measurements and models. Closing the C and N cycles of ecosystems is an ambitious objective that requires observations of many variables, but also many assumptions and inevitably a dose of modelling, and therefore, a critical assessment of methods and uncertainties is needed, as underlined by the other Referee (#1).

Therefore, this paper (part I of the study) was built on two axes: i) a clear methodological focus on the ways to assemble the main components – and evaluate the uncertainties – of the C and N budgets on the basis of observations, as far as possible; and ii) a scientific focus on the patterns of biogeochemical interactions across the monitoring networks, which need to be identified prior to the assessment of the quantitative links between C sequestration and N deposition (in Part II of the study). The second objective is tightly connected to the first: our understanding of the interactions between the C and N cycles is only as good as our measurements and models are at estimating the C and N cycle components.

We believe that the methodological part of the paper and the discussion of uncertainties in C and N budgets require a detailed description and assessment of the different methods used in compiling the budgets. Moving a large part of Materials and Methods to the supplement (as suggested by Referee #2) would weaken this methodological focus, since uncertainties are directly related – and must be discussed in relation to – the methods that are used. A good example of this is the description of methods for the interpretation of eddy covariance data (section 2.3.1), which Referee #2 suggests shortening. The discussion of uncertainties in the $CO_2$ budgets and carbon sequestration efficiency (4.2.1) relies on methods being described in some detail, because the assumptions made in EC data post-processing, gap-filling and partitioning are critical for the C budgets derived from the raw data.

Nevertheless, we do agree with Referee #2 that forest ecosystem (BASFOR) modelling plays a secondary role in this paper (Part I of the study). Its main quantitative contribution to the elemental budgets is the simulation of soil NO and $N_2O$ fluxes, and indeed the size of the model description in 2.6 is not in proportion to the actual importance of BASFOR results in the overall paper. Meanwhile, in the companion paper (Part II, BG_2019-335), Referee #1 writes that it is not clear and transparent '...*how the model is constructed and how it handles the critical assumptions involved. A reader will also need to read the companion paper. Most readers will still be left with many queries. This is not uncommon in the case of modelling papers. Vital assumptions are deeply embedded and not clearly visible although the outcome is constrained by the assumptions...*'.

We have therefore moved the detailed BASFOR model description from Part I to Part II. We now describe the BASFOR model in a few sentences in Part I, and we point to Part II for details of the modelling. This satisfies three objectives: i) the Part I paper is shortened by approximately 1000 words; ii) a more thorough description of BASFOR assumptions, workings and implementation will be found in the Part II paper, where it is more needed; iii) we have redressed the balance in size and contents of the two parts of the study.

In addition, we also agree with Referee #2 that a table summarizing the methods used in the compilation of fluxes would add clarity to the paper and guide the reader through the following sections. This new table (see 'Table 3' on the following page) is now placed in the manuscript at the start of section 2.2 and lists methods and references for each component flux of the C, N and GHG budgets. For each item a small bar diagram shows the percentage of forest and semi-natural sites where measured and modelled data were available.

The results section seems to focus on two major parts: one being the discussion of the resulting nitrogen, carbon and greenhouse fluxes and budgets themselves (and their inter-comparison and validation) and the other being the link between these N and C fluxes and the interpretation. I would recommend separating these two parts as much as possible.

For most of the flux tower sites, the inter-annual carbon fluxes have been presented and published in several previous papers (with key references provided in Table S1), sometimes on a site-by-site basis, sometimes as part of various meta-analyses. We felt it would be superfluous to dwell again on the C budgets by themselves, and that the added value of this paper was to describe the geographical variations in C fluxes in relation to a number of factors including N deposition (hence section 3.2.1). As for N deposition, N losses, and the GHG budgets, which have not all been published previously, they are described in detail by themselves in dedicated sections.

Moreover, the "results" and "discussion" sections seem to be partially intertwined. It would be better to exclusively include discussion points in the latter. Some sections, for instance, the one about "carbon sequestration efficiency", introduce new concepts and results and seem to fit better in the results section (see specific comments below). As for the "discussion" section, it would be more structured if there was a clear separation between the methodological uncertainties used in the estimation of the N, C and greenhouse gas fluxes and the uncertainties related to the interplay of N and C.

We agree that the carbon sequestration efficiency metric should be introduced in Methods (2.3.1), and that the description of CSE results should be moved to results sections 3.2.1 and 3.2.3. We have done this in the revised version.

The uncertainties related to the interplay of N and C are discussed in Section 4.3, while methodological uncertainties in N deposition, N losses, C balance and GHG budgets are discussed in Sections 4.1.1, 4.1.2, 4.2.1 and 4.2.2, respectively, so there is a clear separation of these two types of uncertainties in the discussion.

Table 3. Summary of the main methods used to quantify carbon, nitrogen and greenhouse gas fluxes and budgets for the 31 forests and 9 short semi-natural vegetation sites included in this study. Horizontal bars (green: forests; blue: short semi-natural vegetation) indicate the percentages of study sites with available data (filled bars), or without available data (open bars). See also Supplement Tables S6-S7 for details at individual sites.

| Fluxes and budgets | Components | Experimental data (this study) Methods (selected references) | Literature and other data mining Methods (selected references) | Modelling (this study) Models (selected references) |
|---|---|---|---|---|
| **Carbon** | Net ecosystem exchange (NEE) | Eddy covariance (1) | | |
| | Net ecosystem productivity (NEP) | Gap-filled from NEE (14) | | BASFOR (18) |
| | Gross primary productivity (GPP) | Inferred from NEE (14) | | BASFOR (18) |
| | Ecosystem respiration ($R_{eco}$) | Inferred from NEE (14) | | BASFOR (18) |
| | Soil respiration ($R_{soil}$) | Static/dynamic chambers (12) | Static/dynamic chambers (19) | |
| | Heterotrophic respiration ($R_{het}$) Ratio $R_{het}$ / $R_{soil}$ | | Root exclusion, trenching, girdling, isotopic methods (19) | BASFOR (18) |
| | Dissolved organic/inorganic carbon (DIC / DOC) losses | Suction cups (9); peatbog stream sampling (3) | Lysimeter / suction cups (6); weir (8) ; ground- and ditch-water sampling (7) | |
| | Soil-atmosphere $CH_4$ fluxes | Static chambers (12) / Laboratory soil bioassay (15) | Eddy covariance (10) ; static chambers (7) | |
| **Nitrogen** | Atmospheric $N_r$ concentrations | DELTA (17) | | EMEP (16) |
| | Atmospheric dry deposition | Inferential method (5) | | EMEP (16) |
| | Atmospheric wet deposition (Inorganic $N_r$) | Bulk samplers (2) / Wet-only samplers | Regional networks / kriging | EMEP (16) |
| | Atmospheric wet deposition (wet-soluble organic $N_r$, WSON) | Bulk samplers (Dämmgen, 2006) / Wet-only samplers | | |
| | Throughfall $N_r$ deposition | Throughfall precipitation collectors | | |
| | Dissolved inorganic nitrogen (DIN) losses | Suction cups (9) | Lysimeter / suction cups (11) | IFEF model (4) |
| | Dissolved organic nitrogen (DON) losses | Suction cups (9) | Lysimeter / suction cups (11) | |
| | Soil-atmosphere NO fluxes | Dynamic open chambers (12) / Laboratory soil bioassay (15) | Dynamic open chambers (13) | BASFOR (18) |
| | Soil-atmosphere $N_2O$ fluxes | Static chambers (Luo et al., 2012) / Laboratory soil bioassay (15) | Static chambers (13) | BASFOR (18) |

1 Aubinet et al. (2000) ; 2 Dämmgen (2006) ; 3 Dinsmore et al. (2010) ; 4 Dise et al. (2009) ; 5 Flechard et al. (2011) ; 6 Gielen et al. (2011) ; 7 Hendriks et al. (2007) ; 8 Ilvesniemi et al. (2009) ; 9 Kindler et al. (2011) ; 10 Kowalska et al. (2013) ; 11 Legout et al. (2016) ; 12 Luo et al. (2012) ; 13 Pilegaard et al. (2006) ; 14 REddyProc (2019) ; 15 Schaufler et al. (2010) ; 16 Simpson et al. (2012) ; 17 Tang et al. (2009) ; 18 van Oijen et al. (2005) ; 19 See references in Table S7.

Some of the results need to be updated, e.g. in section 2.2.1 it is written that EMEP data were downloaded in 2013 and different model versions were used?

Some EMEP model results were indeed used for the calculation of total N deposition in this paper, but only for i) wet deposition of inorganic N and ii) the NO2 concentration used in dry deposition. The comment about different versions of the model leading to different results is really only relevant for dry deposition of inorganic N, but this output of the model was not used in our total N deposition estimate; it is only used as a comparison in Fig. S2 of the supplement. Regarding wet deposition and NO2 concentrations from EMEP, we have stated at the end of section 2.2.1 that '…*Evaluation of the model against measurements over this period has shown quite consistent results for the wet-deposited components and NO2 concentrations…*', and therefore the data obtained in 2013 are not significantly different from the most up-to-date version of the model.

In a lot of statements there is an unevidenced qualification, such as: 'there did not appear to be any systematic overestimation ..', '0,85 is plausible but also much variability ..', We assumed that all sites of the European network followed the same relative time course ..', 'reveals a potential cross correlation ..', 'reasonable overall model performance ..', etc. It would help if such statements are underpinned/quantified.

We address each of these statements below:

'…*there did not appear to be any systematic overestimation compared with wet deposition estimates from the monitoring networks or EMEP data*…' This statement appears in methods, and it would not be appropriate to provide numbers, i.e. results, here. The data substantiating this statement are provided on lines 543-547. We have added a pointer to Results in the statement. ('see Results')

'…*the applied ratio of 0.85 is plausible but also that much variability*…' We presume Referee means that the variability is not shown, but the whole sentence (lines 332-334) actually reads '…*A comparison with values of DINTF / Ndep ratios actually measured at the EN2, EN8, EN10, EN16 and DB2 sites (0.71, 0.80, 0.29, 0.85, 1.11, respectively; mean ± st. dev. 0.75 ± 0.30) shows that the applied ratio of 0.85 is plausible but also that much variability in canopy retention/leaching may be expected between sites*…*"*, so we do in fact provide the numbers and the actual range, mean and standard deviation of observed data in the same sentence.

'…*We assumed that all sites of the European network followed the same relative time course*…' This sentence was removed from the revised version, since most of the detailed BASFOR model description has moved to the Part II paper, as described above.

'…*reveals a potential cross correlation*…' We assume Referee #2 means we should provide statistical measures (eg R^2) of the correlations we mention between N deposition and climate. We agree this is a not a straightforward message to convey, or correlation to demonstrate, for two reasons: i) the relationships of $N_{dep}$ to MAT or MAP are circumstantial rather than strictly causal, as we explain in the first paragraph of Section 4.3.1 (and as also pointed out by Referee #2 of the Part II paper); and ii) the relationships are not linear but bell-shaped, as evident in the figures shown below. We do not mean to imply that a change in MAT or MAP automatically induces a change in $N_{dep}$, but that $N_{dep}$ cannot be considered to be completely independent of, or unrelated to, climate.

We have added the following figure to the Supplement (Fig. S4), as suggested by the Editor, to show the relationship of $N_{dep}$ to MAT and MAP:

[Figure]

Figure S4. Spatial variations in measurement-based nitrogen deposition ($N_{dep}$), plotted as a function of (A) mean annual temperature (MAT) and (B) mean annual precipitation (MAP). Temperature and precipitation are not direct determinants of $N_{dep}$, but the geographical occurrence of peak $N_{dep}$ levels in mid-range for both MAT and MAP means that the relationship of forest productivity to $N_{dep}$ cannot be considered independently of climate at the European scale.

'*…reasonable overall model performance…*' We have provided in each sub-plot of Fig.6 (to which this sentence refers on line 675-676) the R^2, MAE and NRMSE statistics of modelled vs measured. It would be cumbersome and repetitive to add to the text all these numbers for each variable.

Finally, there seems to be some parts of the paper that are outdated, such as the start of the discussion about the Magnani paper. These can be omitted.

The publication of the Magnani et al. (2007) paper actually served the purpose of triggering much critical discussion, constructive debate and further studies and reviews on the complicated issue of the impact of nitrogen pollution on the carbon cycle. It is correct that there is now a consensus that the dC/dNdep values calculated by Magnani et al. were over-estimated, but the present paper has a different focus. We believe it is still necessary to stress the importance of two aspects, which were overlooked in the Magnani et al. (2007) paper, but not only there:

1- The size (and often dominance) of dry deposition relative to wet deposition, as well as the uncertainty in dry deposition. Other studies have also downplayed the role of dry deposition, probably because it is more difficult to quantify, and less visible, than wet deposition.
2- The linear relationship of NEP to $N_{dep}$ derived from the Magnani et al. (2007) dataset did not make it apparent that N saturation in polluted areas may be counter-productive to forest growth. Later studies have indicated non-linear relationships, while accounting for other drivers, and we now refer more specifically to them in the text.

We believe that contrasting findings from this study with those from Magnani et al. (2007), and others before and after them, benefits the reader in providing a more balanced view. In this, we are supported by Referee #1 of the Part II paper, who writes: '…*Magnani et al. (2007) reported very large responses of forest carbon sequestration to nitrogen deposition. Several authors rapidly pointed out that the response proposed was way above previous estimates and direct observations in N addition studies. This apparent discrepancy has been discussed at length for more than a decade now, but there is still a need for a more stringent analysis of how dC responds to dN.*'

To highlight the contrast between the different studies, we have added the following two sentences:

Section 4, start of Discussion, line 721: '…*Sutton et al., 2008).* Other attempts have subsequently been made to assess impacts of N deposition on forest growth and carbon sequestration, while accounting for other drivers, at more than 350 long-term monitoring plots in Europe (Solberg et al., 2009; Laubhann et al., 2009; De Vries et al., 2008). *A special feature of the present study is that it aims to assemble N deposition rates …*'

Section 4.1, line 739: '… *especially not with wet N deposition only.* Based on a review of experimental N addition studies (e.g. Högberg et al., 2006; Pregitzer et al., 2008) and monitoring based field studies along N deposition gradients (e.g., Solberg et al., 2009; Laubhann et al., 2009; Thomas et al., 2010), De Vries et al. (2014) suggested that the C response reaches a plateau near 1.5-2.0 g N $m^{-2}$ $yr^{-1}$ and then starts to decrease. *The linear relationship between C sequestration and wet Nr deposition …*'

Additional references:

Högberg P., Fan, H., Quist, M., Binkley, D. and Tamm, C.O.: Tree growth and soil acidification in response to 30 years of experimental nitrogen loading on boreal forest, Glob. Change Biol., 12, 489–499, https://doi.org/10.1111/j.1365-2486.2006.01102.x, 2006.

Pregitzer, K.S., Burton, A.J., Zak, D.R. and Talhelm, A.F.: Simulated chronic nitrogen deposition increases carbon storage in Northern Temperate forests, Glob. Change Biol., 14, 142–153, https://doi.org/10.1111/j.1365-2486.2007.01465.x, 2008.

Thomas, R.Q., Canham, C.D., Weathers, K.C. and Goodale, C.L.: Increased tree carbon storage in response to nitrogen deposition in the US, Nat. Geosci., 3(1), 13–17, https://doi.org/10.1038/ngeo721, 2010.

We have in addition delete an unnecessary reference to Magnani et al. (2007) (line 604-605).

It raises the question to me about what the purpose of the paper is and how it relates to the second part. If it is a description of the database, including reference to the data sources, models and including an uncertainty estimate it can be a much shorter well focused paper with detailed information as supplementary information. The second part can then focus on the use and interpretation of the data. It would be my recommendation to split it in such a way. I therefore recommend major revision of the paper.

We hope we have convinced Referee #2 that this paper is about much more than the description of the database (i.e. a data paper) underpinning Part II of the study. In our response to the Referee's opening general comments (see above), we have argued that constructing realistic C and N budgets at many observation sites, and closing and connecting the C and N cycles (our main scientific objective), on the basis of measurement-based data and complemented where required by models, requires a thorough critical examination of the methods employed. We do not wish to show that our data are flawless (which they are not), but that they are as good as can be under the constraints we are facing, and that it is important to recognize, identify and rank important gaps in data, knowledge and models, whatever they are (e.g. N2 loss by denitrification; DOC leaching; organic N deposition; etc). We believe that the space taken up by method descriptions and uncertainty assessments is justified by these objectives.

To address the Referee's comment that it is not clear '…*what the purpose of the paper is and how it relates to the second part…*', we have made the following changes:

- The current abstract mentions uncertainty in the beginning but then does not mention it anymore. We have summarized our main findings on uncertainty in one sentence added towards the end of the abstract : 'Uncertainties in elemental budgets were much larger for nitrogen than carbon, especially at sites with elevated $N_{dep}$ where $N_r$ leaching losses were also very large, and compounded by the lack of reliable data on organic nitrogen and $N_2$ losses by denitrification.'
- The formulation of the goals of the paper, as defined in the last paragraph of the Introduction, may have been slightly misleading. We wrote in a single sentence: "*The main aim of this paper is to build tentative C, N and GHG budgets … prior to an assessment in the companion paper …*", which could be misinterpreted as meaning that Part 1 is just the Materials and Methods for Part 2. We have rephrased the opening sentence in the following way: 'A main objective of this paper is to build tentative C, N and GHG budgets, and analyse C/N interactions empirically, for a wide range of European monitoring sites, by using measurements or observation-based data as far as possible, complemented by modelling. Important methodological goals are to critically examine uncertainties in measurement methods and elemental budgets, to identify knowledge and data gaps, and to assess the current state of process understanding as encoded in models. To this end, we compiled …'. We have postponed the mentioning of the Part 2 paper until the very last sentence of the Introduction.

**Specific comments**:

Line 127-130– The information in the introduction about tropical forest could be shortened, as the paper focusses on temperate/boreal forest.

The three lines about tropical forests are provided in the introduction as background and contrast to temperate/boreal forests, which are the focus of this paper. It is relevant to briefly contrast the different regions of the globe at the start of the introduction, which discusses the likely magnitudes of global and hemispheric CO2 sinks.

Line 159 – 1 by 1 degree does not correspond to 10 by 10 km, it covers a larger area.

We do not mean that 1 x 1 degree corresponds to 10 x 10 km. We mean that the resolutions of models are typically 10 x10 km for regional CTMs, or 1° x 1° for global CTMs, as indicated by the adverb '*respectively*' at the end of the sentence.

Line 195 – How about vegetation changes at all sited, e.g. due to growth and/or composition change?

We have assumed (because we have no data to suggest otherwise) that vegetation species composition has not evolved since the beginning of measurements, or (in the case of BASFOR modelling) since forests were planted. But we have collected as much data as possible on tree heights, diameters, density, etc at various dates in the past, from publications and databases, as shown for example in Table S2 and Figure S5 of the Supplement. All these data were used in the Bayesian calibration of the BASFOR model (Cameron et al., 2018).

Line 220 – Not all FLUXNET sites seem to be included in this analysis, what are the criteria used to in- or exclude individual sites?

We did not conduct a meta-analysis of all FLUXNET sites, and there was no selection process. We stated in Introduction (lines 205-210) and in Materials and Methods (lines 237-240) that the sites were those included in the CarboEurope IP and NitroEurope IP projects, at which we installed $N_r$ deposition monitoring equipment (and measured other variables). We could not use all FLUXNET sites for this study, since our aim was to quantify as many components of the C and N cycles on the basis of measurements as far as possible, and such data are not available for most FLUXNET sites.

Line 262 – In this section it is sometimes unclear to what "models" are referred. Specify what type of models, e.g. dry deposition models. What are the key differences between these deposition models? It would also be good to specify the EMEP model domain and resolution in this section.

The dry deposition models were described in some detail in Flechard et al. (2011), where four models were compared and their uncertainties analyzed. The main principle of such models is described in the present paper on lines 258-261; i.e., the dry deposition flux is obtained by the product of measured ambient concentration and a deposition velocity ($V_d$), that depends on meteorology, surface roughness, leaf area index, chemical species, etc. We recognize that it is difficult for a non-specialist to see clearly how this works on the basis of such a short description, but we believe that a full description of the method is outside the scope of this paper, especially if we do not wish to make the paper longer than it already is. The most effective way is to refer the reader to specialized papers (see line 261), as we do for all other methods.

The EMEP model resolution is given in this section (line 288), and the modelling domain is shown in Fig. 5.

Line 284 – leaving out the organic N data leads to a systematic bias.

We acknowledge that failing to account for organic $N_r$ deposition in our total $N_r$ deposition estimate leads to a small bias. We were unfortunately unable to estimate the missing organic terms (both wet and dry) by modelling. But as we state on line 287, the underestimation is a small one (<5% of total $N_r$ deposition), and much smaller than the overall uncertainty in $N_{dep}$ (as discussed in Section 4.1.1).

Line 290 – There are probably new data available?

We copy below the reply we made to Referee #1 on the same topic:

Some EMEP model results were indeed used for the calculation of total N deposition in this paper, but only for i) wet deposition of inorganic N and ii) the NO2 concentration used in dry deposition. The comment about different versions of the model leading to different results is really only relevant for dry deposition of inorganic N, but this output of the model was not used in our total N deposition estimate; it is only used as a comparison in Fig. S2 of the supplement. Regarding wet deposition and NO2 concentrations from EMEP, we have stated at the end of section 2.2.1 that '…*Evaluation of the model against measurements over this period has shown quite consistent results for the wet-deposited components and NO2 concentrations…*', and therefore the data obtained in 2013 are not significantly different from the most up-to-date version of the model.

Line 308 – Could you elaborate on why you decided to scale up the N2O and NO fluxes using linear interpolation?

This a common but admittedly not ideal procedure in such studies, where flux measurements are not continuous and gaps need to be filled somehow to yield annual flux estimates. We acknowledge that linear interpolation can lead to large uncertainties in annual flux estimates, e.g. by missing emission peaks that can occur (unnoticed) between two consecutive flux measurement dates, or by over-representing over time large fluxes measured on certain days. The issue is related to whether the statistical NO or $N_2O$ flux distribution is normal (Gaussian) or quasi log-normal. The latter is frequent in managed, fertilized agro-systems, where large emission peaks occur during just a few days or weeks per year, in response to fertilization, and emissions are otherwise very small. In forests and non-fertilized semi-natural systems, fluxes tend to be more normally distributed, and we believe that the uncertainty associated with linear interpolation is much smaller.

Ideally a more mechanistic gap-filling procedure for NO or $N_2O$ fluxes, based on soil moisture, temperature, and other indicators of soil microbial activity and SOM turnover, should be used. However, in practise this is often very hard to achieve using field measurements, either due to the low temporal resolution of measurements, or because no significant patterns can be derived between fluxes and environmental macro-drivers. Linear interpolation then represents the default alternative. We have added the following text in this section, to briefly summarize the issue:

*'…scaled up to yearly values by linear interpolation or using the arithmetic mean of all flux measurements*. There may be considerable uncertainty in the annual flux if gap-filling is based on linear interpolation between discrete values, when flux measurements are made manually and therefore discontinuous and infrequent (Parkin, 2008). This is due to the episodic nature and lognormal distribution of NO and $N_2O$ emissions, observed particularly in fertilized croplands and grasslands. However, this 'episodicity' is less pronounced in semi-natural ecosystems, or at least the magnitude of the episodic fluxes is generally much smaller than in fertilized agro-systems (Barton et al., 2015). The uncertainty in annual emissions estimated in our study from manual chamber measurements is related to the observation frequency (bi-weekly or monthly), and larger than in the case of automatic (continuous) chamber measurements.'

Added References:

Barton, L., Wolf, B., Rowlings, D., Scheer, C., Kiese, R., Grace, P., Stefanova, K. and Butterbach-Bahl, K.: Sampling frequency affects estimates of annual nitrous oxide fluxes, Sci. Rep.-UK, 5:15912, https://doi.org/10.1038/srep15912, 2015.

Parkin, T.B.: Effect of sampling frequency on estimates of cumulative nitrous oxide emissions, J. Environ. Qual., 37, 1390–1395, https://doi.org/10.2134/jeq2007.0333, 2008.

Line 326 – DINTF is not included in the abbreviations table.

$DIN_{TF}$ was added to Table 2.

Line 334 – there is an enormous range, why not used in some way?

We did not find a pattern to the ratio of throughfall to total $N_{dep}$ for these five sites. More observations would be needed to link this ratio to forest and climatic characteristics.

Line 336 – This section, about the EC processing, is lengthy. Consider shortening it.

We have touched upon this specific issue as part of our response to Referee #2's general comments (see above). The discussion of uncertainties in the $CO_2$ budgets is dependent on methods being described in some detail, because the assumptions made in EC data post-processing, gap-filling and partitioning are critical for the C budgets derived from the raw data. Ecosystem carbon budgets and sequestration are the main motivations for this work, and collective experience from 25 years of flux monitoring networks around the world shows that there are many ways and options to measure and analyze eddy covariance data, with the end results (inter-annual mean $CO_2$ budgets) being heavily influenced by methodological choices. One illustration is the discussion on night-time fluxes and advection issues (lines 843-909).

Line 389 – Missing comma between "some sites" and "such as"

Done

Line 427 – Add a reference to the table in the supplement that includes the major publications per site.

Done

Line 480 – It would make more sense to move the primary purpose of the BASFOR model to the beginning of the section.

Most of the BASFOR description (Section 2.6) was moved to the companion paper, as suggested above. But we have kept a very brief summary (a few lines) describing the purpose and basic principles of BASFOR.

Line 538 – 550 – The discussion on the model uncertainties would fit better in the discussion section.

We agree that lines 538-542 consitute discussion material for dry deposition; they were moved to Section 4.1.1.

Lines 543-550 describe wet deposition results by different methods without further discussion.

Line 580 – 594 – This section discusses the computation of the denitrification losses and the uncertainty associated with it. Consider moving a part of this section to the methods/discussion.

We have very little data on denitrification $N_2$ losses. This paragraph provides a couple of very tentative numbers as part of results, as a background to the more informed and detailed inorganic N fluxes; but it is really too small to split into Methods and Discussion as suggested by the Referee.

Line 603– 612 – It seems a bit sudden to address the ultimate objective of the whole project here. It would be better to discuss this earlier on in the paper.

The objectives of the paper were laid out in the final paragraph of Introduction, and were strengthened in the revised version by the addition of a couple of sentences, as described above in response to the Referee's request for a clearer description of the aims of the paper.

However, the sentence on line 603-604 did not actually refer solely to this paper's objectives, but to the '*ultimate objective of the project*' as a whole, including the companion paper (Part II of the study).

Line 640 – You state that there is a "broad negative correlation between MAP and MAT", could you support this with a R2 value? Moreover, the MAP alone does not seem appropriate to address site-specific water-availability, without considering other parameters such as soil water holding capacity etc.

We have added the $R^2$ value to the text:

'…for sites with MAT > 7 °C there was a broad negative correlation between MAT and MAP ($R^2$ = 0.24, p = 0.01) i.e. the warmest sites in southern Europe…'

We agree entirely with the Referee's comment that MAP is not '*appropriate to address site-specific water availability, without considering other parameters such as soil water holding capacity etc*'. As we state in introduction to 3.2.1, the first stage of the study is a '*descriptive approach*', which however '*…is done with the strong reservation that a simple empirical relationship does not necessarily prove causality, as other confounding and co-varying factors, e.g., climate, soil, age, etc, may exist.*' (lines 606-608). We further emphasized, later on (lines 969-976), that '*…In addition, a range of other potential explanatory variables such as soil type, especially the water holding capacity ($\Phi_{FC}$ - $\Phi_{WP}$), soil fertility (Vicca et al., 2012; Legout et al., 2014), tree species, stand age (Besnard et al., 2018), are potentially needed to explain the observed variability (Flechard et al., 2019).*'

Thus our intention in this paper was first to show the apparent inter-relationships between carbon sequestration, N deposition and climate, in a similar way to previous studies, but at the same time pointing to the difficulties and risks of single factor interpretations. Multiple factors are clearly involved (climate, soil, fertility, age, species, etc), which, owing to the limited size of our database and the non-independence of controlling factors, could not be untangled using statistical methods (see lines 959-965 and 972-976). Having established this methodological challenge, we therefore proposed to investigate the issue in the companion paper (Part II) using mechanistic modelling.

Line 647 – Missing "were"

The verb for the sentence was on the next line (648)

Line 659 – 674 – In this section you are discussing results related to Figure 4 again. Consider moving this section up.

Fig.4 contains the data described in both Sections 3.2.1 and 3.2.2. We think it is more logical to start describing the results with '*Spatial variability of the carbon sink in relation to climate and nitrogen deposition*' (3.2.1) and then move on to '*Differences between plant functional types*' (3.2.2).

Line 737 – Add a reference for this number.

We have added the reference to Figure 3:

'…when Ndep exceeds a critical load of approximately 2-2.5 g (N) m-2 yr-1 (Fig. 3),…'

Line 754 – how can you demonstrate that it 'provides a better estimate' and that it is more realistic than an transport model?

We agree that the wording of this sentence is perhaps too strong. Indeed, we can't prove from our data and with 100% certainty that '…*measuring gas-phase and aerosol Nr concentrations locally did provide a better estimate of total ecosystem Nr inputs…*'. The only way to prove this would be to measure all dry deposition fluxes for all gaseous and particulate $N_r$ species by micrometeorological methods, for several years, which is practically impossible.

We nonetheless believe that having a record of actual measured gas-phase and aerosol $N_r$ concentrations at all sites, for several years, at least reduces the uncertainty in $N_r$ concentrations, compared with a large-scale chemical transport model.

We have changed the verb of the sentence from '*did provide*' to '*should provide*'.

Line 804 – It would be better to either add the definitions for N saturation here or refer to this later.

This section discusses the uncertainties in N losses and budgets, rather than the concepts of saturation, whose definitions come later (4.3.2); however it makes sense to at least mention N saturation at this stage.

Line 930 – Why is the uncertainty in the non-CO2 fluxes possibly >100% larger?

'…*the uncertainty in non-CO2 GHG fluxes is much larger (possibly > 100%) than for multi-annual EC-based CO2 datasets…*' for several reasons, which we have explained in the text:

- because, unlike $CO_2$, in our datasets the non-$CO_2$ GHG fluxes were not measured in situ continuously (manual soil chamber measurements made periodically, except at a few sites equipped with auto-chambers);
- because non-$CO_2$ GHG flux measurements were mostly made by chambers and thus not at the ecosystem scale, unlike eddy covariance, which integrates a large footprint;
- because for some sites, there were no in-situ flux measurements, and fluxes were estimated from the bioassay experiments, or from BASFOR modelling.

Line 972–975 – You state here that your attempts with more advanced forms of regression analysis were not successful. This is a bit on the vague side. You should either elaborate on the attempts that were made and/or for instance add some references, or exclude this statement from this section.

We have rephrased this sentence in the following way:

'…*are potentially needed to explain the observed variability*. In order to account for, and untangle, the multiple inter-relationships, we chose a mechanistic model (BASFOR) based approach, described in Flechard et al. (2019), whereby most of the known interactions of plant, soil, climate, age, species, are encoded and parameterised to the best of our current knowledge. Given the limited size and very large diversity of the dataset, such an approach appears to be preferable to regression-based statistical analyses, since a simple pattern to explain the coupling of carbon and nitrogen budgets with the available data and knowledge is unlikely.'

Line 1016 – It is unclear why considering the uncertainties would amount to a confident threshold of 2-2.5 g N, rather than just widening the range of the given estimate, could you elaborate on this?

We agree that the wording was confusing. What we meant is that there may be less uncertainty in the threshold for the advanced stage of saturation (2-2.5 g N m$^{-2}$ yr$^{-1}$) than in the thresholds for early or moderate saturation. We changed the sentence starting line 1016:

'…*Following definition ii) of N saturation given above, the sum of inorganic Nr losses, heavily dominated by DIN leaching at the upper end of the Ndep range in our datasets (Fig. 3), may indicate various stages of N saturation in all forests with Ndep > 1-1.5 g (N) m-2 yr-1.* A threshold for a more advanced saturation stage could be placed at 2-2.5 g (N) m-2 yr-1, where inorganic N$_r$ losses are consistently larger than 50% of N$_{dep}$. *Such numbers are entirely consistent…*'

Line 1052 –1055 – This sentence is a bit confusing. It refers to the large uncertainty in dry deposition modelling, but both deposition estimates from CTM and denuders make use of dry deposition models in the end. So I would say that it is not so much the deposition models that improved substantially, but more the estimates of the concentrations of these compounds.

We did not state that '…*the deposition models … improved substantially…*', but that '…*the low-cost network to monitor atmospheric gas-phase and aerosol N$_r$ contributed to substantially reducing the large uncertainty in total N$_{dep}$ rates at individual sites*'. We agree however that the sentence was a little long and confusing, and therefore we split it into two:

'Nevertheless, the low-cost network to monitor atmospheric gas-phase and aerosol N$_r$ contributed to substantially reducing the large uncertainty in total N$_{dep}$ rates at individual sites (compared with gridded outputs of a regional chemical transport model). This was because dry deposition almost systematically heavily dominates over wet deposition in forests, except at very remote sites (away from sources of atmospheric pollution), and directly measured N$_r$ concentrations reduced the uncertainty in dry deposition fluxes'

Figure 4 – What type of LAI data is used to determine LAImax?

Leaf are index is defined both in the text (line 629) and in Table 1 (footnote) as '*1-sided for broad-leaf, or half of total for needle-leaf*' .

Figure 8 – Some of the labels seem to be missing.

For each site in each plot, there are two values (observed and modelled). We provided only one label per site (attached to the 'observed' series) in order not to clutter the plots. The site name for the 'modelled' series can be easily deduced by following a vertical path, since both 'observed' and 'modelled' data points have the same x-value.

Figure 9 – "to each tohet" – to each other OK

Consider splitting up / reformulating these sentences for clarity/readability:

For each sentence we provide the revised text.

Line 119-121 – "The global terrestrial . . . Pg (C ) yr-1"

'The global terrestrial net sink for atmospheric carbon dioxide (CO2) is approximately 1.7 Pg (C) yr-1, i.e. roughly one fifth of global CO2-C emissions by fossil fuel combustion and industry (9.4 ± 0.5 Pg (C) yr-1). This corresponds to the land based carbon (C) uptake of 3.2 ± 0.8 Pg (C) yr-1 minus emissions from deforestation and other land-use changes of 1.5 ± 0.7 Pg (C) yr-1.'

Line 127-130 – "Tropical forest areas . . . generally believed."

'Tropical forest areas are believed to be closer to carbon neutral (Pan et al., 2011), or even a net C source globally (Baccini et al., 2017), due to emissions from deforestation, forest degradation and land use change offsetting their

sink potential. However, others (Stephens et al., 2007) have argued that the tropical land CO2 sink may be stronger – and the Northern hemispheric land CO2 sink weaker – than was generally believed.'

Line 297-299 – "Nitrogen losses to . . . empirical methods."

'Nitrogen losses to the atmosphere (gaseous emissions) and to groundwater (N leaching) are especially hard to quantify and thus typically cause large uncertainties in ecosystem N budgets. These $N_r$ losses were estimated by direct flux measurements or by indirect empirical methods.'

Line 308 -309 – "To address . . . responses of soils."

'Direct in situ Nr and non-CO2 GHG gas flux measurements were unavailable at many sites. These soil N2O, NO (and also CH4) fluxes were therefore also estimated, as part of NEU, from the temperature and moisture responses of soils.'

Line 525-528 – "Total Nr deposition . . . Schwede et al., 2018)."

'Total Nr deposition was around 25% smaller on average at short semi-natural vegetation sites compared with forests (Fig. S2), even though the mean total atmospheric Nr concentrations (reduced and oxidized, N-containing gas and aerosol compounds) were quite similar between the two data sets (Flechard et al., 2011). The difference was driven by higher dry deposition rates over forests due to higher aerodynamic roughness and deposition velocities (Fig. S3; see also Schwede et al., 2018).'

Line 548-550 – "Wet deposition . . . most forest sites."

This sentence is fairly straightforward.

Line 754-756 – "Despite these . . . cycling processes."

'Despite these uncertainties, measuring gas-phase and aerosol Nr concentrations locally should provide a better estimate of total ecosystem Nr inputs than the outputs of a large-scale chemical transport model. In addition, the partitioning of wet *vs* dry deposition, reduced *vs* oxidized N, and canopy absorption *vs* soil deposition, should also be improved, all of which are useful in interpreting ecosystem N cycling processes.'

Line 790-793 – "In addition . . . as beech or oak."

'In addition, it is noteworthy that the two sites with the largest $N_{dep}$ and DIN leaching rates (EN15, EN16) were dominated by pine or Douglas fir (Table S2). These species have been shown in a common garden experiment (Legout et al., 2016) to cause larger nitrification, NO3- leaching and acidification rates (as well as larger losses of calcium, magnesium and aluminium), compared with other tree species such as beech or oak.'

Line 824-828 – "By analogy . . . observation periods."

'Previous studies have normalised data through the carbon use efficiency (CUE, commonly defined from a plant's perspective as the NPP/GPP ratio), or the biomass production efficiency (BPE = BP/GPP; Vicca et al., 2012), which is a CUE proxy. By analogy, we define here an ecosystem-scale, medium-term indicator of carbon sequestration efficiency (CSE) as the NEP/GPP ratio, calculated from measurable fluxes over the CEIP/NEU project observation periods.'

Line 959-963 – "Through the continent-wide. . . continental scale."

'Nitrogen deposition patterns at the European scale result from the continent-wide geographical distribution of population, human, industrial and agricultural activities, and of precursor emissions, combined with mesoscale patterns of meteorology-driven atmospheric circulation and chemistry. Through the interplay of these factors, the elevated Ndep levels in this study happened to co-occur geographically with temperate climatic zones of Central-Western Europe (Fig. 5 C-D) that are the most conducive to vegetation growth at the continental scale.'

Line 972-975 – "Our attempts . . . the dataset."

This has been addressed and rephrased as part of a previous comment (see above).

Line 1052-1055 – "Nevertheless, . . . is much larger."

This has been addressed and rephrased as part of a previous comment (see above).

[revised manuscript text omitted]

**Commentaire [c4]:** Clarified the main goals of the paper

**Commentaire [c5]:** Reference to the companion paper for the analysis of the dC/dN response

*{Insert Table 2 here}*

The forest sites of the study ranged from very young (< 10 years old) to mature (> 150 years old), and can be broadly classified into four plant functional types (PFT) or five dominant tree categories (Table 1): deciduous broadleaf (DB), evergreen needle-leaf (EN, comprising mostly spruce and pine species), mixed deciduous/coniferous (MF), and Mediterranean evergreen broadleaf (EB). Forest species composition, stand characteristics, C and N contents of different ecosystem compartments (leaves, wood, soil), soil physical properties and micro-climatological characteristics are described in Tables S2-S5. Semi-natural short vegetation ecosystems included unimproved (mountainous and semi-arid) grasslands, wetlands and peatlands; they are included in the study as unfertilised, C-rich soil systems, providing a contrast with forests where storage also occurs above ground (thus with different C/N ratios). Among the 40 EC-$CO_2$ flux measurement stations, most sites (36) were part of the CEIP $CO_2$ flux network. A further three $CO_2$ flux sites were operated as part of the NEU network (EN2, EN16, and SN3), and one site (DB4) was included from the French F-ORE-T observation network (F-ORE-T, 2012). Table S6 provides an overview of the available C, N and GHG flux measurements, detailed hereafter.

**2.2 Nitrogen fluxes**

Input and output fluxes of the ecosystem nitrogen and carbon budgets are represented schematically in Fig. 1. The following sections describe the methods used to quantify the different terms, summarized in Table 3.

*{Insert Table 3 here}*

**Commentaire [c6]:** Added new Table 3 to provide overview of methods used in compiling C and N budget 
[revised manuscript text omitted]

[Figure]

**Commentaire [c27]:**
Improved site label readability:
  -Increased font size for site labels
  -Moved labels to avoid overlapping

**Figure 3. Comparison of measured and estimated ecosystem inorganic $N_r$ losses and their relationships to total atmospheric $N_r$ deposition (x-axis) at the forest sites. NO fluxes (A) and $N_2O$ fluxes (B) were either i) measured in situ using static or dynamic flux chambers, ii) scaled up from laboratory bioassay-derived T/WFPS relationships, or iii) simulated using the BASFOR ecosystem model (see text for details). DIN leaching (C) was either measured (lysimeter or suction cups), or predicted from the Dise et al. (2009) empirical algorithm. The sum of inorganic $N_r$ losses (DINleach + NO+ $N_2O$) was computed as the mean of measured values and modelled estimates. In panels A-C, site names are indicated for sites where in situ measurements were available.**

[Figure]

**Figure 4. Overview of inter-annual mean EC-derived C flux estimates ( GPP, $R_{eco}$ and NEP), ecosystem LAI and leaf N content, in relation to total (dry + wet) atmospheric $N_r$ deposition (A-E), and relationship of $R_{eco}$ to GPP (F), for forests (filled circles, black labels) and short semi-natural vegetation (filled stars, magenta labels). In all plots, the colour scale indicates mean annual temperature (MAT), while the symbol size is proportional to mean annual precipitation (MAP, scale provided in panel A).**

1670

[Figure]

**Figure 5. Distribution of observation-based nitrogen deposition (N$_{dep}$) (A) and gross primary productivity (GPP) (B) for the forest sites of this study, within the European climate space represented by mean annual temperature (MAT) and precipitation (MAP). In plot A the symbol color indicates N$_{dep}$ while the symbol size is proportional to GPP; in plot B the symbol color indicates GPP, while the symbol size is proportional to N$_{dep}$. Plots C shows modelled N$_{dep}$ from the EMEP model over coniferous forests (year 2010), represented in climate space (1 data point for each grid square of the EMEP domain containing coniferous forests), also shown as a map (D). The MAT axis can be seen as a proxy for latitude and/or elevation, while the MAP axis expresses to some extent longitude (distance to the ocean) and/or orographic precipitation enhancement.**

[Figure]

**Commentaire [c28]:**
Improved site label readability:
  -Increased font size for site labels
  -Moved labels to avoid overlapping

[revised manuscript text omitted]

---

## Referee Report (RR1)

3 February 2020

Re-review for the paper submitted by Flechard et al. MS No.: bg-2019-333

This is my second review to this paper, the present one after the authors had included their responses to the two reviewers' reports.

I'm satisfied with the point-by-point responses provided to the reviewers comments and with the changes introduced in the text. I think that the reading has improved after rearranging some sentences. I think that the authors have succeed to convey the message of the importance of their task of gathering the maximum number of measurements and observations, and modeling, to discuss the uncertainties of measurements and models, and how these uncertainties translate to the interpretation of the biogeochemical interactions of the C and N cycles. Then, an assessment can be made of the main gaps and future directions.

 As suggested by rev. 2, the text has benefitted from the transfer of the detailed BASEFOR model description to the companion article in Part II.

I have only very minor remarks.  I suggest to include them to the final version as a final touch.

Minor remarks:
Fig. 6  In the text for the figure, besides MAE and NRMS, add R2, coeficient of determination.

Fig. 8. Explain in the text for the figure that only dots corresponding to observations are labelled.

Fig. S4. Include a legend for discontinuous and continuous lines in the MAP and MAT plots.

Line 605. In response to rev 2 to better definí the "broad negative correlation between MAP and MAT" the autors have included the R2 value=0,24, line 606). However, if one wants to highlight the "negative relationship" it is best to use the correlation coeficient (r) which in this case will have a negative sign, so that the inverse relationship is better displayed.